# Modeling of mRNA deadenylation rates reveal a complex relationship between mRNA deadenylation and decay

Agnieszka Czarnocka-Cieciura [ID] [1,6], Jarosław Poznański [ID] [2,6], Matti Turtola [ID] [3], Rafał Tomecki[2,4], Paweł S Krawczyk[1], Seweryn Mroczek[1,4], Wiktoria Orzeł [ID] [1], Upasana Saha[5], Torben Heick Jensen [ID] [5], Andrzej Dziembowski [ID] [1,4✉] & Agnieszka Tudek [ID] [2✉]

## Abstract

**Complete cytoplasmic polyadenosine tail (polyA-tail) deadenylation is thought to be essential for initiating mRNA decapping and subsequent degradation. To investigate this prevalent model, we conducted direct RNA sequencing of *S. cerevisiae* mRNAs derived from chase experiments under steady-state and stress condition. Subsequently, we developed a numerical model based on a modified gamma distribution function, which estimated the transcriptomic deadenylation rate at 10 A/min. A simplified independent method, based on the delineation of quantile polyA-tail values, showed a correlation between the decay and deadenylation rates of individual mRNAs, which appeared consistent within functional transcript groups and associated with codon optimality. Notably, these rates varied during the stress response. Detailed analysis of ribosomal protein-coding mRNAs (RPG mRNAs), constituting 40% of the transcriptome, singled out this transcript group. While deadenylation and decay of RPG mRNAs accelerated under heat stress, their degradation could proceed even when deadenylation was blocked, depending entirely on ongoing nuclear export. Our findings support the general primary function of deadenylation in dictating the onset of decapping, while also demonstrating complex relations between these processes.**

**Keywords** mRNA Deadenylation and Degradation; ONT Nanopore Direct RNA Sequencing (DRS); Ccr4-NOT and Pan2/3 Deadenylases; Pab1; Dcp2 Decapping and Xrn1 Degradation
**Subject Categories** RNA Biology; Translation & Protein Quality

## Introduction

The 3' end polyadenosine tail (pA-tail) is an essential modification for the mRNA biogenesis in the nucleus and its cytoplasmic lifetime. In budding yeast, Pap1 polyA-polymerase synthesizes de novo 60 adenosine-long pA-tails, in a process governed by the Cleavage and Polyadenylation Factor (CPF) and the nuclear polyA-binding protein Nab2 (Turtola et al, 2021; Aibara et al, 2017; Rodríguez-Molina and Turtola 2022). A pA-tail of at least 40 adenosines is required to safeguard the mRNA from nuclear decay and facilitate its export to the cytoplasm (Dower et al, 2004). The protective function of the pA-tail is mediated by Nab2 (Schmid et al, 2015). The conserved Mex67-Mtr2 hetero-dimer mediates mRNA export to the cytoplasm (De Magistris, 2021). The inhibition of export by depleting Mex67 (Haruki et al, 2008) leads to a rapid decay of newly formed mRNAs, as Nab2 becomes sequestered on non-exported mRNAs that accumulate in the nucleus (Tudek et al, 2018).

Two main complexes, Pan2/3 and Ccr4-NOT, mediate cytoplasmic pA-tail deadenylation. The mechanism of deadenylation has long been a scientific question. In mammals, a two-phase deadenylation model posits that Pan2/3 initiates pA-tail shortening, with Ccr4-NOT completing the process (Chen and Shyu, 2010; Yi et al, 2018). However, in budding yeast, each complex has distinct preferred substrates, showcasing redundancy; yeast Pan2/3 predominantly targets mRNAs of high abundance, whereas Ccr4-NOT is recruited more efficiently to low-abundant mRNAs (Tudek et al, 2021).

Essential to mRNA metabolism, pA-tails in the cytoplasm are coated by the Pab1 polyA-binding protein (Brambilla et al, 2019). In vitro experiments have demonstrated that Pab1 significantly contributes to regulating pA-tail length by stimulating Ccr4-NOT-mediated deadenylation of a 60 A substrate and temporarily inhibiting deadenylation on a 30A RNA (Webster et al, 2018). Other in vitro assays revealed that Pan2/3 more efficiently deadenylates longer pA-tails than the shorter (20–25) ones.

[1]International Institute of Molecular and Cell Biology, Księcia Trojdena 4, 02-109 Warsaw, Poland. [2]Institute of Biochemistry and Biophysics, Polish Academy of Sciences, Adolfa Pawińskiego 5A, 02-106 Warsaw, Poland. [3]Department of Life Technologies, University of Turku, Biocity, Tykistökatu 6, 205240 Turku, Finland. [4]University of Warsaw, Faculty of Biology, Miecznikowa 1, 02-089 Warsaw, Poland. [5]Aarhus University, Department of Molecular Biology and Genetics—Universitetsbyen 81, 8000 Aarhus, Denmark. [6]These authors contributed equally: Agnieszka Czarnocka-Cieciura, Jarosław Poznański. ✉E-mail: adziembowski@iimcb.gov.pl; atudek@ibb.waw.pl

Importantly, Pab1 can stimulate Pan2/3 activity without being required for deadenylation (Wolf et al, 2014). Despite multiple efforts, the precise in vivo role of Pab1 in mRNA pA-tail metabolism has not been fully established, although it has been implicated in deadenylation based on reporter mRNA assays (Caponigro and Parker, 1995). The coexistence of Pan2/3 and Ccr4-NOT, potentially influenced by Pab1, complicates generalizations based solely on in vitro studies, substantiating the need for a comprehensive in vivo approach. In addition to Pab1, translation was proposed as a major regulator of the deadenylation rate. The translation initiation factor complex is thought to bind to Pab1, forming a loop that may influence translation initiation and mRNA degradation regulation (Tarun et al, 1997; Otero et al, 1999; Archer et al, 2015). Moreover, codon optimality, which is dictated by the relative concentration of specific tRNAs, was shown to additionally regulate the deadenylation and decay rates of reporter transcripts (Presnyak et al, 2015).

Current models of mRNA decay, often derived from reporter systems, suggest that significant pA-tail shortening triggers decapping by Dcp1/2, followed by Xrn1 5'-3' exonuclease-mediated decay (De Magistris, 2021; Chen and Shyu (2010); Yi et al, 2018; Decker and Parker, 1993). Here, our objective was to validate these models using transcriptome-wide in vivo data. To achieve this, we experimentally modeled deadenylation and decay rates under steady-state and stress conditions using data derived from Nanopore Direct RNA Sequencing (DRS). Our approach focused on describing the evolution of pA-tail length distribution across the entire coding transcriptome or specific transcript groups. To this end, we established two analysis methods. The first utilized a modified gamma distribution model combined with Mean Field Theory (Gupta and Groll, 1961; Réka and Barabási; 2002) and determined that the transcriptomic enzymatic deadenylation rate is constant and equal to 10 A/min. The second, a simplified method, calculated deadenylation rates based on the evolution of quantile pA-tail length values. We found that the apparent adenosine half-life of individual transcripts varied from seconds to tens of seconds and strongly correlated with mRNA decay rates, indicating a functional link. Within this correlation, long-lived, mostly abundant mRNAs are deadenylated slowly, whereas short-lived transcripts experienced rapid tail shortening. Our study showed that mRNA decay and deadenylation rates are consistent across large functional groups of transcripts, such as those derived from ribosomal protein genes (RPGs), and are correlated with codon optimality, as previously established (Presnyak et al, 2015). Further analysis of RPG mRNA group metabolism under heat stress conditions showed that although both deadenylation and decay rates increased, deadenylation is not a prerequisite for mRNA decapping and decay but is instead a major stimulating factor. RPG mRNA decay proceeds through complete deadenylation inhibition in a double *ccr4Δ pan2Δ* mutant and is strongly dependent on ongoing mRNA export. Therefore, we conclude that while deadenylation primarily dictates decapping onset, other factors can dominate in significant functional mRNA groups, such as RPGs, which constitute 40% of the coding transcriptome.

# Results

## mRNA nuclear export block reveals cytoplasmic deadenylation and decay

Previous studies have estimated bulk yeast mRNA half-lives of 12 min or even lower (Miller et al, 2011; Sun et al, 2012; Neymotin

et al, 2014; Presnyak et al, 2015; Chan et al, 2018). The efficient uncoupling of mRNA synthesis and degradation within the shortest possible time frame was critical for simultaneously modeling transcriptome-wide deadenylation and decay. Drawing from our previous work (Tudek et al, 2018, Schmid et al, 2018), we employed the Anchor-Away system (Haruki et al, 2008), to deplete the main cellular export factor, Mex67, thus rapidly inducing massive nuclear degradation of newly synthesized mRNAs. Concurrently, already-exported mRNAs continued their metabolism in the cytoplasm, allowing the specific monitoring of deadenylation and degradation kinetics of this subset of transcripts (Fig. 1A). We generated three biological replicates, each containing one control sample and five test time points to increase modeling power (Fig. 1B). To measure both mRNA abundance and pA-tail lengths, we utilized the Direct RNA Sequencing (DRS) method (Brouze et al, 2023; Tudek et al, 2021).

Transcript degradation was evidenced by reductions in mRNA abundance, as shown for bulk mRNA distributions (Fig. EV1A) and single gene examples (Figs. 1C and EV1B). mRNA half-lives were calculated using a standard continuous exponential function (Dataset EV1; Miller et al, 2011; Sun et al, 2012; Neymotin et al, 2014; Chan et al, 2018). The median transcript half-life was 9.7 min for mRNAs and 2.6 min for ncRNAs (Fig. 1D). These calculated half-lives strongly correlated with those from two previous studies that utilized metabolic labeling (Miller et al, 2011; Chan et al, 2018; Spearman rho 0.74 and 0.62, respectively; Figs. 1E,F and EV1C,D), despite each dataset yielding a different median half-life, likely due to a systemic bias as suggested by Chan et al, 2018. We concluded that Mex67 depletion provides a suitable experimental setting for studying cytoplasmic mRNA decay.

## Deadenylation rate differs by transcript abundance group

To describe global changes in mRNA pA-tail length over time, we generated whole-transcriptome distributions derived from the DRS datasets (as sum of all reads indiscriminately of transcript type; Figs. 2A and EV1E). Mex67 depletion resulted in a widespread shortening of pA-tails over time, with the most significant alterations observed in mRNA fractions with long pA-tails, as depicted in a density plot (Fig. 2A). Time-dependent violin plots further illustrated varied dynamics of pA-tail length change across quantiles, with values within the top quantiles decreasing more rapidly (Fig. 2A). Electrophoresis analysis of global pA-tail length distribution showed analogous dynamics following Mex67 depletion, thereby validating our methodology (Fig. 2B).

Subsequently, we separated the transcriptome into classes, which we hypothesized would exhibit varying rates of deadenylation; the main criterion for this classification was mRNA abundance. Consistent with our previous observations (Tudek et al, 2021), we confirmed that highly abundant mRNAs are characterized by shorter mean pA-tails and longer half-lives (Figs. 2C and EV1F,G). Strikingly, the 187 most enriched mRNAs in the DRS datasets (out of ~5000 total) accounted for about 60% of the reads. Within this group, mRNAs produced from ribosomal protein genes (RPG mRNAs) formed the largest functional subgroup, summing to 40% of all mRNA reads (Fig. 2C). These abundant mRNAs largely shape the distribution of the entire transcriptome (Figs. 2A and EV1H). In contrast, low-abundance mRNA clusters

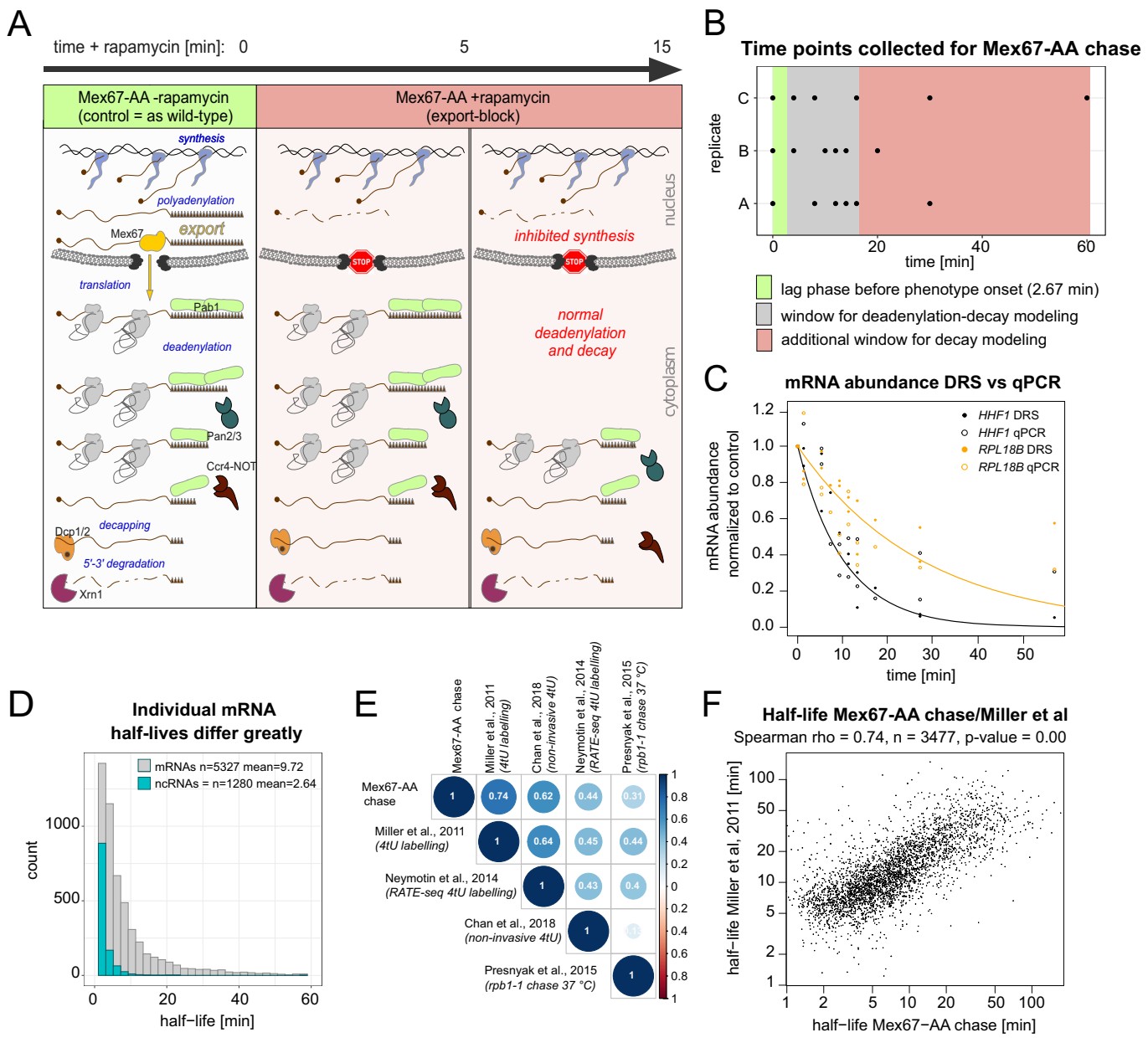

**Figure 1. mRNA nuclear export reveals cytoplasmic deadenylation and decay dynamics.**

(**A**) Schematic showing the theoretical experimental set-up. (**B**) Overview of collected time points and their relevance for decay or deadenylation modeling. (**C**) Time-dependent abundance change of *HHF1* or *RPL18B* mRNAs following Mex67 depletion in DRS datasets (dot) or by reverse transcription coupled with qPCR (ring). A fitted line represents the decay factor calculated from DRS data. (**D**) mRNA and ncRNA half-life distribution calculated from the Mex67-depletion time course. (**E**) Comparison of mRNA half-life estimations from various studies visualized as a matrix of Spearman rho coefficients (Miller et al, 2011; Neymotin et al, 2014; Presnyak et al, 2015; Chan et al, 2018). (**F**) Correlation between half-life values in Mex67-depleted sample and half-life values from Miller et al, 2011. Source data are available online for this figure.

exhibited more dynamic alterations in pA-tail lengths (Figs. 2D and EV1I,J), especially pronounced for mRNAs with long pA-tails at the initial time points. Indeed, low-abundance or non-RPGs mRNAs underwent deadenylation more rapidly than did highly abundant transcripts, including RPGs (Fig. 2E). This observation was supported by pA-tail distribution profiles of several individual genes, which exhibited notably diverse half-lives (illustrated in Fig. EV1K). We concluded that deadenylation rates are transcript-specific, with low-abundance mRNAs being deadenylated more

rapidly and consequently exhibiting shorter half-lives compared to high-abundance transcripts.

## Decapping dominates over deadenylation for mRNAs with pA-tails shorter than 20 As

One notable observation regarding the pA-tail length distribution in Mex67-depleted cells is the absence of tails shorter than 20 adenosines. This can be illustrated by a rapid decline in the

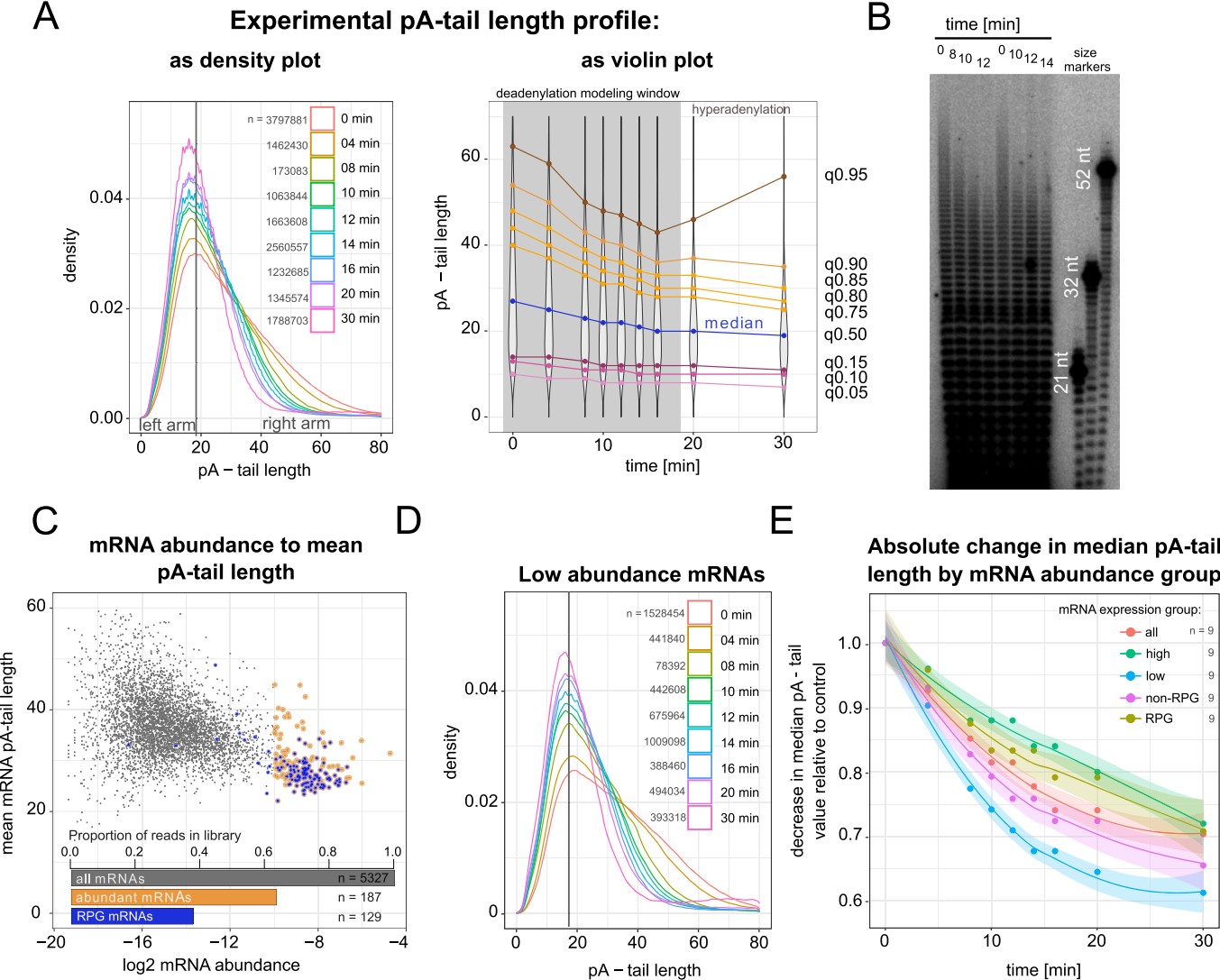

**Figure 2. mRNA groups exhibit various deadenylation rates.**

(**A**) Global distribution of mRNA pA-tail lengths during the Mex67-chase experiment. DRS data are presented in a density plot (left) or violin plot (right). Replicates, shown separately in Fig. EV1E, were merged. The number of transcripts (pA-tail estimates) in each density plot is given on the panel. As in Fig. 1B, the violin plot also highlights in gray the timepoints used for deadenylation rate modeling. The latter timepoints were discarded due to the occurrence of mRNA hyperadenylation; a phenotype specific to very few newly made mRNAs in nuclear export-block conditions (Jensen et al, 2001). These scarce species only significantly impact the overall pA-tail distribution after most cytoplasmic mRNA have been deadenylated and degraded. (**B**) Autoradiogram depicting the global distribution of pA-tail lengths in selected Mex67-chase samples used to construct the sequencing library. (**C**) Log2 mRNA abundance compared to mean pA-tail length for control Mex67-AA cells, with highly abundant and RPG mRNAs as gold and blue dots, respectively. A bar plot summarizes the read fraction for each transcript category. (**D**) Global distribution of pA-tail lengths of low-abundance mRNAs. The number of transcripts in each density plot is given on the panel. (**E**) Time-dependent changes in median pA-tail length across the entire coding transcriptome and four large mRNA groups: high-low abundance, and RPG-non-RPG. Local regression trend lines are shown with a 95% confidence interval. The number of pA-tail estimates used to calculate the median for each point is given in (**A, D**) and Fig. EV1E,H,J. Replicates were merged to produce nine median estimates for various time points. Source data are available online for this figure.

distribution curve at 20 adenosines and a complete absence of tails shorter than 15 for individual mRNAs (Figs. 2A,D and EV1E,H,J,K). Moreover, even at the later time points of Mex67 depletion, the left side of the distribution curve remained almost unchanged.

To investigate the biological relevance of this observation, we depleted the decapping enzyme Dcp2 or the Xrn1 5'-3' exonuclease for 2 h using the auxin-inducible degron (AID) system (Appendix Fig. S1A,B; Nishimura et al, 2009; Morawska and Ulrich, 2013). Depletion of 5' decapping and decay factors shifted the pA

distribution of all mRNAs and individual transcripts towards shorter-tailed values (Fig. 3A,B; Appendix Fig. S1C–E). The DRS method has reduced efficiency in detecting mRNAs with short pA-tails (Fig. 3A; Tudek et al, 2021), potentially underestimating the global pA-tail shortening in Dcp2- or Xrn1-depleted cells. Nonetheless, the observed accumulation of short-tailed mRNAs was significant under Dcp2-/Xrn1-depletion. Since this was not observed in the Mex67-depletion datasets, we can safely conclude that in yeast, deadenylation continues until the pA-tail reaches 20

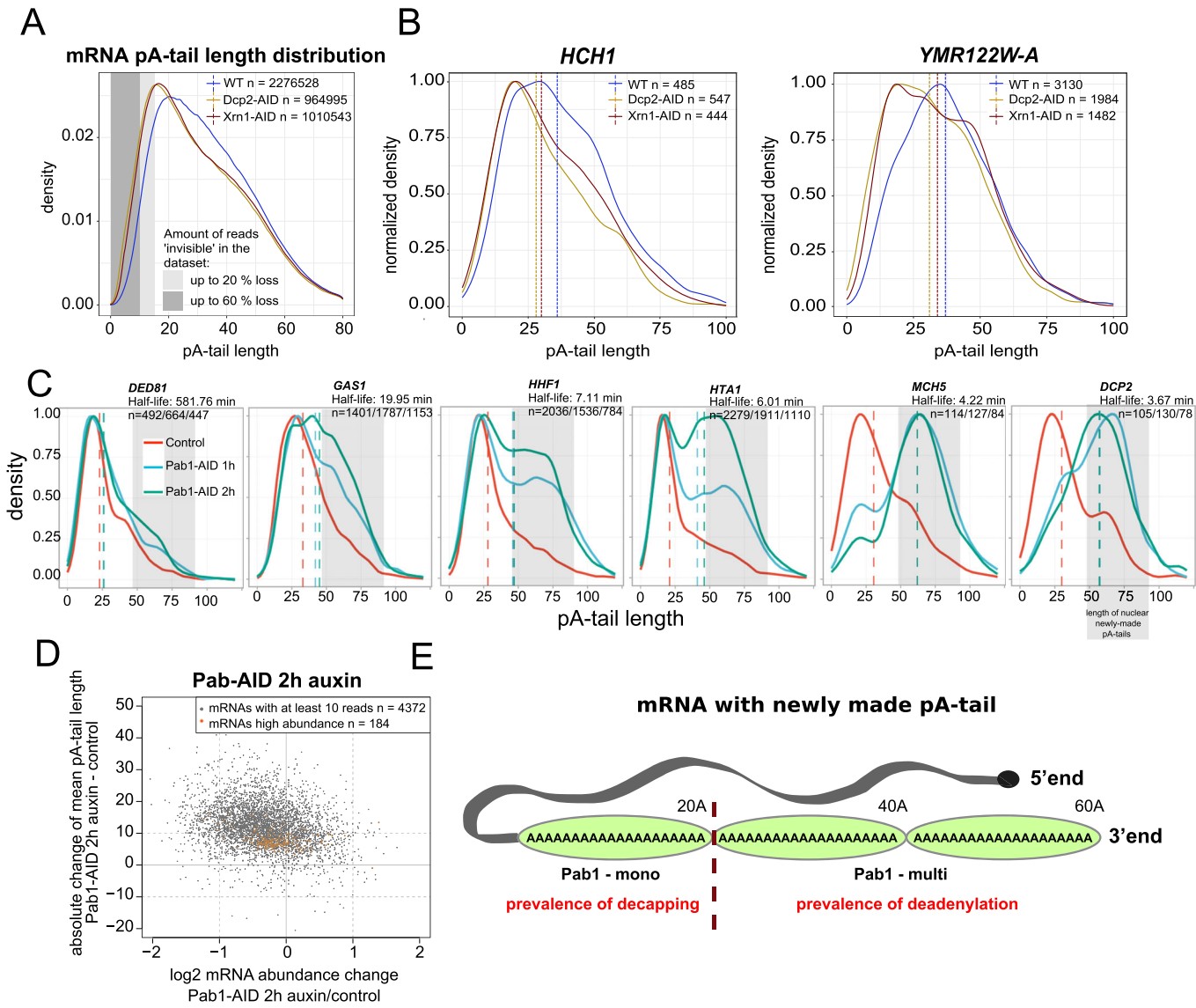

**Figure 3. Pab1 controls deadenylation and decapping.**

(A) Global distribution of pA-tail lengths of RNAs isolated from control, Dcp2- or Xrn1-depleted cells. Gray areas delineate the pA-tail lengths underestimated in the DRS library, as previously reported (Tudek et al, 2021). (B) pA-tail length distribution of *HCH1* and *YMR122W-A* mRNAs in control and Dcp2- or Xrn1-depleted cells. (C) pA-tail length distribution of *DED81*, *GAS1*, *HHF1*, *HTA1*, *MCH5*, and *DCP2* mRNAs in control cells compared to strains depleted of Pab1 using the AID system for 1 or 2 h. The mRNAs are ranked by half-life time. The number of reads contributing to each distribution is indicated in the panel with n = 'control'/'1 h depletion'/'2 h depletion'. The gray boxes show the range of adenosines added de novo in the nucleus by the polyA-polymerase Pap1 on pre-mRNAs (Turtola et al, 2021). (D) Comparison of log2 fold change in mRNA abundance to absolute change in mean pA-tail length for 2 h Pab1-depleted cells compared to control. (E) Schematic illustrating that one Pab1 can bind from 20 to 30 adenosines, therefore newly synthesized mRNA pA-tail bear di- or trimers of Pab1. Depending on the number of Pab1 attached to the pA-tail, the mRNA is either susceptible to deadenylation of decapping. Source data are available online for this figure.

adenosines in length, whereafter it triggers decapping, consistent with previous findings in mammalian cells (Eisen et al, 2020). To validate our observation, we performed in vitro digestion of a wild-type RNA sample with Xrn1, predicting the removal of uncapped transcripts to reveal the preferred pA-tail length at which decapping occurs (Appendix Fig. S1F). However, the whole mRNA pA-tail distribution did not change, indicating that in a wild-type context, uncapped transcripts are rare (Appendix Fig. S1G,H), which aligns with the belief that exonucleolysis promptly follows decapping, consistent with physical associations between Dcp1/2 and Xrn1 (Braun et al, 2012; Charenton et al, 2017).

However, upon closer inspection of individual mRNA p-tail profiles, we observed digestion of a small fraction of short-tailed mRNAs (Appendix Fig. S1I), indicating potential decapping following pA-tail shortening.

## Pab1 stimulates deadenylation when bound as a multimer

We examined the role of Pab1 in deadenylation based on several observations. First, previously published in vitro studies suggested that Pab1 regulates deadenylation by physically interacting with Ccr4-NOT and Pan2/3 deadenylases (Webster et al, 2018; Schäfer et al, 2019).

Second, we observed that the peak of the pA-tail distribution for both whole-transcriptome and single mRNAs fell between 20–30 As; a length which was previously shown to be bound by a single Pab1 molecule (Webster et al, 2018; Baer and Kornberg, 1980; Schäfer et al, 2019). Therefore, we reasoned that Pab1 may regulate the balance between decay and deadenylation depending on whether it binds the pA-tail as a monomer or a multimer. To test this hypothesis in vivo, we depleted Pab1 using the AID system for 1 and 2 h, attaining a 40% and 20% decrease in protein, respectively (Appendix Fig. S1J; Nishimura et al, 2009; Morawska and Ulrich, 2013). Inspection of the whole-transcriptome distribution revealed a slight and gradual accumulation of long pA-tailed mRNAs (50–80As) during Pab1 depletion (Appendix Fig. S1K). Next, we inspected single mRNA examples representing a wide range of mRNA half-lives (Fig. 3C; Appendix Fig. S1L). Polyadenosine distributions of mRNAs with long half-lives mimicked the whole-transcriptome distribution. In contrast, mRNAs with short half-lives showed a marked increase in long pA-tailed mRNAs, at times completely replacing the control pA-tail distribution. The length of accumulated pA-tails corresponded to the estimated length of newly synthesized pA-tails (Turtola et al, 2021; Tudek et al, 2021). Since budding yeast lack cytoplasmic adenylases, these accumulated mRNAs must originate from de novo transcription following Pab1 depletion. They gradually replaced mRNAs produced ulteriorly, which was clearly more pronounced for short-lived mRNAs. This suggests a global effect, as all mRNAs displayed an increase in mean pA-tail length (Fig. 3D; Appendix Fig. S1M). Since the long-tailed mRNAs tended to accumulate, we reasoned that they did not undergo deadenylation (Fig. 3E). This highlights the in vivo role of Pab1 in promoting deadenylation, consistent with previous in vitro and in vivo analyses performed on a much smaller scale (Schäfer et al, 2019; Webster et al, 2018; Sachs and Davis, 1989; Caponigro and Parker, 1995). Combining these findings with our previous analyses (Fig. 3A,B), which showed that decapping likely occurs around the most prevalent pA-tail length (20–30 A), we reasoned that mRNAs with a single Pab1 are more susceptible to decapping rather than deadenylation. Conversely, mRNAs with more than one Pab1 are preferentially deadenylated (Fig. 3E). These biological insights were crucial for refining our deadenylation modeling strategies.

## A modified gamma distribution describes mRNA pA-tail profiles in yeast

To develop a numerical model for deadenylation, our initial step involved fitting experimental distributions of pA-tails to a single function. This function required adaptable parameters that, when systematically adjusted, would accurately reflect the entire spectrum of experimental distributions. We selected the widely used two-parameter gamma distribution (Gupta and Groll, 1961):

$$p(x) \sim x^{\gamma shape} \cdot e^{(-\gamma rate \cdot x)}$$

Importantly, a variant of the gamma function known as the Erlang distribution can be applied to discrete distributions such pA-tails. However, our model needed to account for specific characteristics of the experimental polyadenosine distribution, which sharply declines at 20 adenosines for all mRNAs, an effect of domination of decapping/decay in this range. This decline aligns with the fact that

Pab1 binds minimally to 20 adenosines, and mRNAs lacking Pab1 are highly susceptible to decapping. Therefore, we modified the standard gamma function to consider the protective role of Pab1 against RNA decapping using Mean Field Theory (Réka and Barabási, 2002). To account for the effective impact of each Pab1 RRM binding to the pA-tail, we assumed that each adenosine binds to Pab1 with the same strength and that the binding force is additive, ultimately leading to a strong association of the entire Pab1 molecule with the pA-tail. Thus, interactions involving $N_A$ indistinguishable objects (adenosine residues) were described using the following equation:

$$tanh(\beta \cdot N_A),$$

The $\beta$ factor, affecting the shape of the distribution, was estimated as 0.096 through iterative adjustment of function fitting to the Mex67-AA depletion dataset. Gratifyingly, this function closely follows the left arm of experimental distributions, rapidly increasing from 0 and saturating at 20 adenosines (Fig. 4A, red line).

In the end, the combination of the classical gamma distribution with the saturable pA-tail:Pab1 interaction led to the final equation:

$$p(N_A) = tanh(0.096 \cdot N_A)^{\gamma shape} \cdot e^{(-\gamma rate \cdot N_A)}$$

This equation produced distributions that matched experimental data (Fig. EV2A). The form of the modified gamma distribution is governed by the two standard parameters, $\gamma\_rate$ and $\gamma\_shape$ (Fig. 4B).

The $\gamma\_rate$ can be conceptually linked to the balance between rates of nuclear adenylation and cytoplasmic deadenylation, as it primarily dictates the position and shape of the distributions' right arm, which represents the quantity of long pA-tailed mRNAs (Figs. 4B and EV2A). The model implies that the relative abundance of two RNA populations differing by a single adenosine in the pA-tail is constant ($e^{-\gamma\_rate}$), independent of the pA-tail length. The latter is supported by the asymptotic log-linear dependence observed in the experimental p($N_A$) distribution.

The $\gamma\_shape$ parameter lacks a direct biological interpretation, but within our experimental conditions, the interplay between $\gamma\_shape$ and $\gamma\_rate$ dictates the position of the distribution maximum. However, the Erlang distribution interprets the *shape + 1* value as the number of critical events following the Poisson distribution required to complete a process. In the biological system we studied, these critical events correspond to the removal of three Pab1 molecules (each binding to 20 As of a 60 A newly made pA-tail) during deadenylation, followed by decapping and rapid mRNA decay mediated by Xrn1. Therefore, the total number of known events leading to mRNA disintegration sums to five. Accordingly, the experimental $\gamma\_shape$ values of the control pA-tail distributions ranged between 3.5 and 4, reflecting a number of critical events ($\gamma\_shape + 1$) close to five (Figs. 4B and EV2A).

Having established a function to describe pA-tail distributions, we fitted it to experimental data from each time point of the Mex67-depletion experiment (Figs. 4C and EV2B). The distributions observed after Mex67 depletion aligned well with the model. However, we observed noticeable deviations from the modified

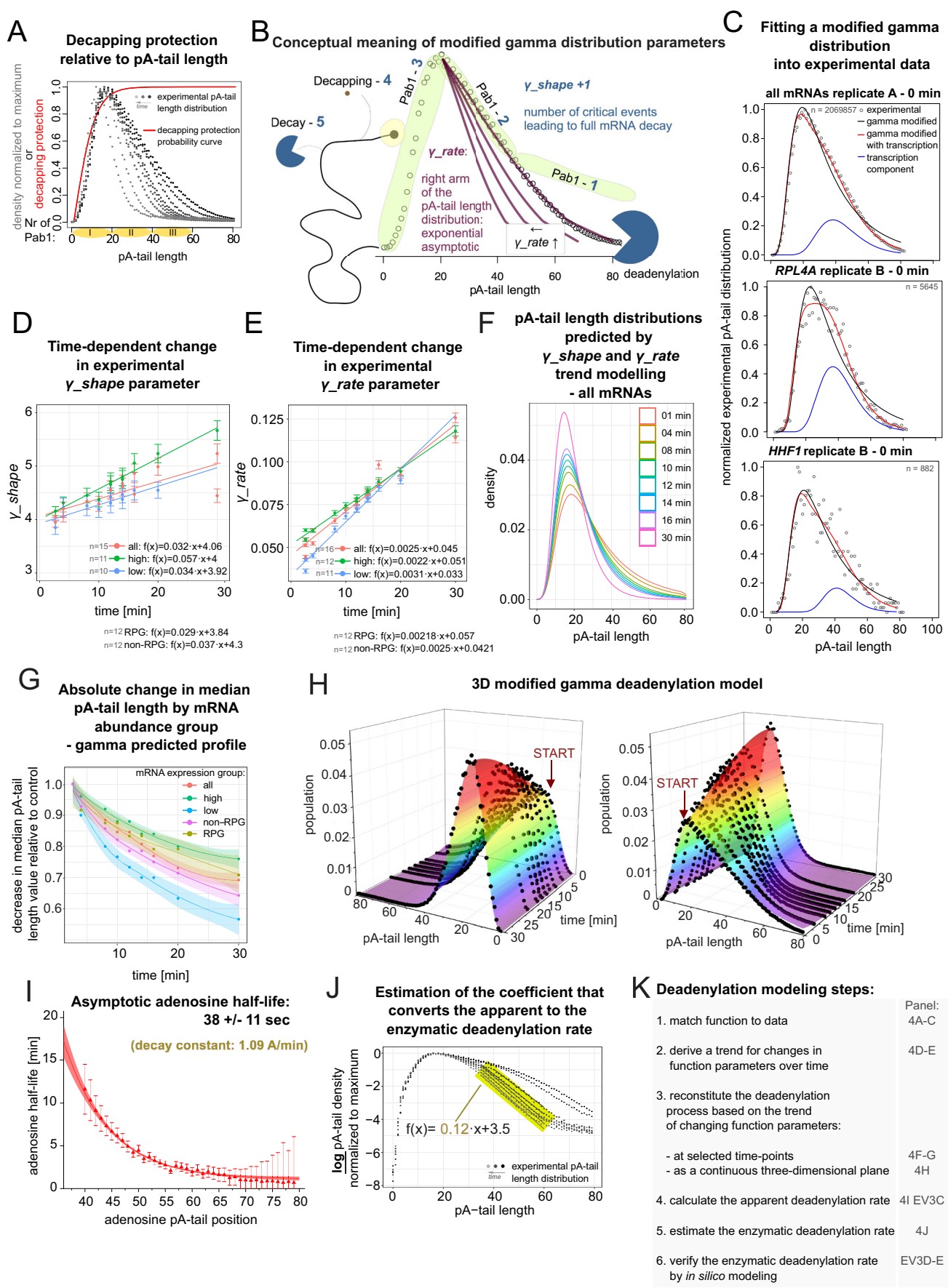

◀ **Figure 4.  The modified gamma model predicts mRNA pA-tail length distributions.**

(A) The probability of protection against decapping in function of pA-tail length is represented by the red line. Gray dots show experimental pA-tail length distributions from all Mex67-chase time points. The number of reads summing up to form the distribution is indicated in Fig. 2A. The presumed number of Pab1 subunits on a pA-tail of a given length is displayed. (B) Conceptual scheme illustrating the modified gamma distribution parameters (see main text). $\gamma\_rate$ corresponds to the exponential coefficient of the right arm of the pA-tail distribution, whereas $\gamma\_shape + 1$ represents the number of critical events leading to complete mRNA decay. (C) Using gray dots, three density plots show experimental pA-tail length distribution for whole mRNA transcriptome in control cells for replicate A and in control samples for *RPL4A* and *HHF1* mRNAs. Black lines represent fitted modified gamma distributions. Estimation of new mRNA production (by log-fitting) is shown using blue lines. The red line represents the sum of the modified gamma and transcription estimate distributions. the *n* number indicates the number od pA-tail estimations (reads) used to produce the experimental distribution. (D, E). Changes in the value of $\gamma\_shape$ (D) and $\gamma\_rate$ (E) parameters of the modified gamma probability distribution over time for all mRNAs or those of low and high abundance separately (RPG and non-RPG are omitted for clarity). The parameters are given as full-colored dots supplemented with vertical standard error bars. Each estimate was derived from distributions shown in Figs. 2A, D and EV1A, H–J (refer to those panels for the number of reads). Continuous functions, which were fitted are shown with the equations given. The number of dataset points considered for the estimation of the functions' equation is indicated on each panel. Those functions were used in subsequent modeling (F, G and Fig. EV2F–J). (F, G) Time-dependent evolution of whole-transcriptome pA-tail length distribution (E) or median (G) based on functions described in (D, E). (H) Modified gamma model of whole-transcriptome deadenylation over time in three-dimension shown as a continuous surface colored according to population density. Black dots show experimental pA-tail distributions. The number of reads summing up to form the distribution is indicated in Fig. 2A. (I) As red triangles with vertical error bars are plotted terminal adenosine half-lives calculated independently for each position relative to their rank in the pA-tail, calculated from distributions adjusted for mRNA decapping (Fig. EV3B). A continuous exponential function is fitted with a 0.95 confidence interval. (J) Experimental pA-tail distributions (same as in (A)) displayed on a normal logarithmic scale (gray dots). The linear distribution between pA-tail lengths 25–60 (yellow box) is highlighted. The slope of the function represents the correction factor used to calculate microscopic enzymatic deadenylation speed from the apparent one. (K) Table summarizing the steps of the deadenylation modeling performed using the modified gamma distribution. Source data are available online for this figure.

gamma profile specifically in control samples, particularly within segments containing longer pA-tails. We hypothesized that ongoing transcription may distort the modified gamma profile. To determine the form of the pA-tail fraction responsible for distorting the control samples, we subtracted the modified gamma distribution from the experimental data, obtaining distributions clustering around 40–60 As. These pA-tail lengths corresponded perfectly with the previously published estimates of newly synthesized mRNA pA-tail lengths, which were experimentally determined using various orthogonal methods (Tudek et al, 2021; Turtola et al, 2021). Thus, we conclude that the modified gamma profile effectively describes the experimental yeast pA-tail length distributions, and that any major deviations from the modified gamma distribution are primarily due to increased synthesis of new mRNA. Accordingly, the mean and variance of the fitted modified gamma distributions correlated well with the experimental data from Mex67 depletion (Fig. EV2C,D).

To ultimately validate our approach, we examined the time-dependent change in the $\gamma\_shape$ and the $\gamma\_rate$ parameters of the fitted modified gamma distributions across the entire transcriptome, its sub-groups (Figs. 4D,E and EV2E), and individual transcript examples (Fig. EV2F,G). Notably, these parameters exhibited coordinated changes over the course of Mex67 depletion. This allowed us to fit predictive functions for their evolution. Consequently, we generated theoretical pA-tail distributions for all mRNAs, as well as those with high and low abundance, which strongly resembled experimental distributions (compare Figs. 4F to 2A; compare EV2H to 2D and EV1H). Specifically, the model accurately replicated the time-dependent changes observed in experimental pA-tail distributions, including their mean, variance (Fig. EV2I) and, most importantly, the median (compare Figs. 4G to 2E). As expected, theoretical pA-tail distributions for individual gene examples (*HHF1*, *RPL4A*, *GAS1*, and *RPL36A*; Fig. EV2J) deviated more significantly from experimental data (likely due to lower read counts, affecting accurate estimation of pA-tail distributions; compare with Fig. EV1K). Nonetheless, these deviations clearly highlighted differences in deadenylation dynamics for each transcript.

We conclude that the modified gamma distribution accurately describes experimental mRNA pA-tail profiles, particularly for highly expressed mRNA groups, thereby providing a reliable numerical model to predict the evolution of pA-tail distributions in budding yeast.

## Estimation of the whole-transcriptome microscopic (enzymatic) deadenylation rate from the modified gamma model

To visualize dynamic changes in the deadenylation process over time, we developed a three-dimensional model (Fig. 4H) based on the modified gamma functions fitted to experimental whole-transcriptome data data (Fig. EV3A). It is important to note that the model does not directly provide a precise deadenylation-decapping rate because distributions are internally normalized in each snapshot. To incorporate decapping-decay into our calculations, we instead used a coefficient derived from the decreasing amount of pA$^+$ RNA recovered from Mex67-depleted cells in each chase time point (Fig. EV3B). To calculate the deadenylation rate, we analyzed changes in the levels of each adenosine relative to the preceding nucleotide (N in relation to N + 1) using a series of 80 coupled differential equations (see "Methods" section). Within the range of 40–80 adenosines, where the contribution of decapping is minimal, the levels of each pA-tail length followed an exponential decay pattern (Fig. EV3C). Importantly, control samples matched this trend when shifted 160 s forward on the time scale (or 2.67 min; Fig. EV3C). We interpret this discrepancy as the time required for Mex67 export from the nucleus and the establishment of the export-block phenotype, marking the biological starting point of our depletion experiment. Therefore, our previous mRNA half-life calculations were adjusted to reflect this time difference (highlighted in Fig. 1B).

Then we plotted the half-lives of individual adenosines against their positions in the pA-tail, revealing an exponential relationship (Fig. 4I). Initially, this suggests that the rate of adenosine removal slows down as the pA-tail shortens. However, the decrease in

adenosine half-life at position N is simply a consequence of the same chain length ($N$) being formed from the decay of a longer tail ($N + 1$). The half-lives of individual adenosines at positions 40–80, where the $N + 1$ contribution was minimal, converged to an asymptote. Importantly, this indicated that the deadenylation rate at the transcriptome level remains constant along the length of the pA-tail. The transcriptomic adenosine half-life equaled the value of the asymptote and was estimated at $38 +/- 11$ s, translating to a disintegration constant of 1.09 (confidence limits 0.84–1.54) adenosine per minute (Fig. 4I). It is important to note that this rate represents an apparent deadenylation rate, describing changes across the entire pA-tail distribution. As stated above, in a mixed population of pA-tail lengths ($X_{NA}$), the microscopic (enzymatic) rate of removing the terminal adenosine from a pA-tail of a given length $N_A$ is seemingly diminished by the deadenylation of a polyadenosine chain that is one adenosine longer ($X_{NA+1}$ of $N_A + 1$, respectively). In other words, the loss of the total number of pA-tails of 39 adenosines (due to their deadenylation to 38 As) is apparently reduced by the products of decay in pA-tails of 40 adenosines to 39 As. The difference between the enzymatic ($R$) and apparent deadenylation rate ($R_{app}$) is encapsulated by the first-order reaction formula widely used in molecular biology:

$$-R_{app} \cdot X_{NA} = dX_{NA}/dt = -R \cdot X_{NA} + R \cdot X_{NA+1}$$
$$= -R \cdot X_{NA} \cdot (1 - X_{NA+1}/X_{NA}).$$

We demonstrated that the distribution of pA-tail lengths ($X_{NA}$) is described by the continuous modified gamma distribution (Fig. 4C–H). Within the range of pA-tail lengths long enough to saturate interaction with at least one Pab1 molecule (i.e., longer than 20 As, or else the length of pA-tail protected by one Pab1 in footprint experiments, as reported in Webster et al, 2018, or shown to bind Pab1 in structural studies by Schäfer et al, 2019) the ratio $X_{NA+1}/X_{NA}$ equals $e^{-\gamma\_rate}$, and the apparent deadenylation rate can be simply expressed as:

$$R_{app} = R \cdot (1 - e^{-\gamma rate}),$$

where $\gamma\_rate$ is a parameter of the modified gamma distribution estimated for the combined RNA population (Fig. EV3A). For short experimental times, $\gamma\_rate$ values were estimated to be ~0.1 (Fig. EV3A), implying a microscopic deadenylation rate of ~10 times higher, ~10.9 A/min (confidence interval: 8.4–15.4). Alternatively, the rate adjustment coefficient ($1 - e^{-\gamma\_rate}$) can also be roughly estimated as the average slope value of the linear functions fitted to the right arm of the experimental distributions, where normalized density is displayed on a log-normal scale (Fig. 4J; "Methods"; note that this factor is actually estimated from the global three-dimensional model).

## Validation of the calculated microscopic deadenylation rate using in silico simulation of whole-transcriptome deadenylation

To further validate our model, we conducted deadenylation simulations (Fig. EV3D,E). We assumed that deadenylation is distributive and occurs at a constant speed (i.e., microscopic

deadenylation rate). These simulations were based on a key factor, $\alpha$, which is a dimensionless scaling factor that frames one round of deadenylation. Starting with control distribution data for the entire transcriptome (simulation in Fig. EV3D uses one of the control samples in EV1E), we iteratively applied multiple rounds of virtual deadenylation procedures (Fig. EV3D). Next, we compared these simulated results to actual experimental data collected at various time points after Mex67 depletion, aiming to identify the simulated distribution that best matched experimental observations. This comparison revealed a strong Pearson correlation ranging from 0.87 to 0.96 between the experimental time and the number of virtual deadenylation rounds applied in the simulation. (Fig. EV3E). This supports the notion that the microscopic deadenylation rate is constant and does not depend on the pA-tail length, at least within the range of pA-tails longer than 20 As (see "Discussion").

In sum, our modeling strategy yielded a common modified gamma equation that can describe deadenylation-related changes of yeast pA-tail distributions. This allowed us to reconstitute the deadenylation process and led to the calculation of the constant transcriptomic enzymatic deadenylation rate, which we have cross-verified using in silico simulation (Fig. 4K).

## An alternative method to estimate deadenylation rates with small datasets

Estimating deadenylation rates using the modified gamma distribution model is effective only for pA-tail profiles with high read coverage because it relies on terminal adenosine half-life estimates. However, for most mRNA profiles, quantitative representation is insufficient. Therefore, to calculate the deadenylation rate of individual transcripts, we developed a method that provides a good approximation of the previous model. In this method, we describe the distribution of polyadenosine tail lengths by measuring specific quantile values, which significantly decrease over time during Mex67 depletion. This decrease is particularly noticeable in the upper quantiles (from the 75th to the 95th percentile). For more details, see the combined distribution in Fig. 2A and the individual data points in Fig. 5A. We fitted exponential functions to quantile pA-tail lengths (Figs. 5A and EV3F), obtaining deadenylation coefficients and calculating adenosine half-lives for each pA-tail length. Adenosine half-lives relative to quantile pA-tail lengths followed an exponential function (Fig. 5B), similar to the findings of the modified gamma model. Within the terminal adenosines, the average adenosine half-life was $39 -/+ 7$ s (Fig. 5B), and the associated disintegration constant to 1.3 adenosine/min (Fig. EV3G), consistent with the apparent deadenylation rate derived from the gamma model.

In conclusion, we developed two separate methods to estimate the apparent deadenylation rate. The first method, using the modified gamma distribution, is precise and enables calculation of the enzymatic (microscopic) deadenylation rate. However, it requires large datasets and is unsuitable for analyzing individual transcripts. The second method, based on quantiles, works with a smaller number of reads (at least 10) but offers less precision. This quantile-based method does not directly measure the enzymatic deadenylation rate, but instead provides a numerical value useful for comparing the deadenylation rates among individual mRNAs, as shown in Fig. 5C.

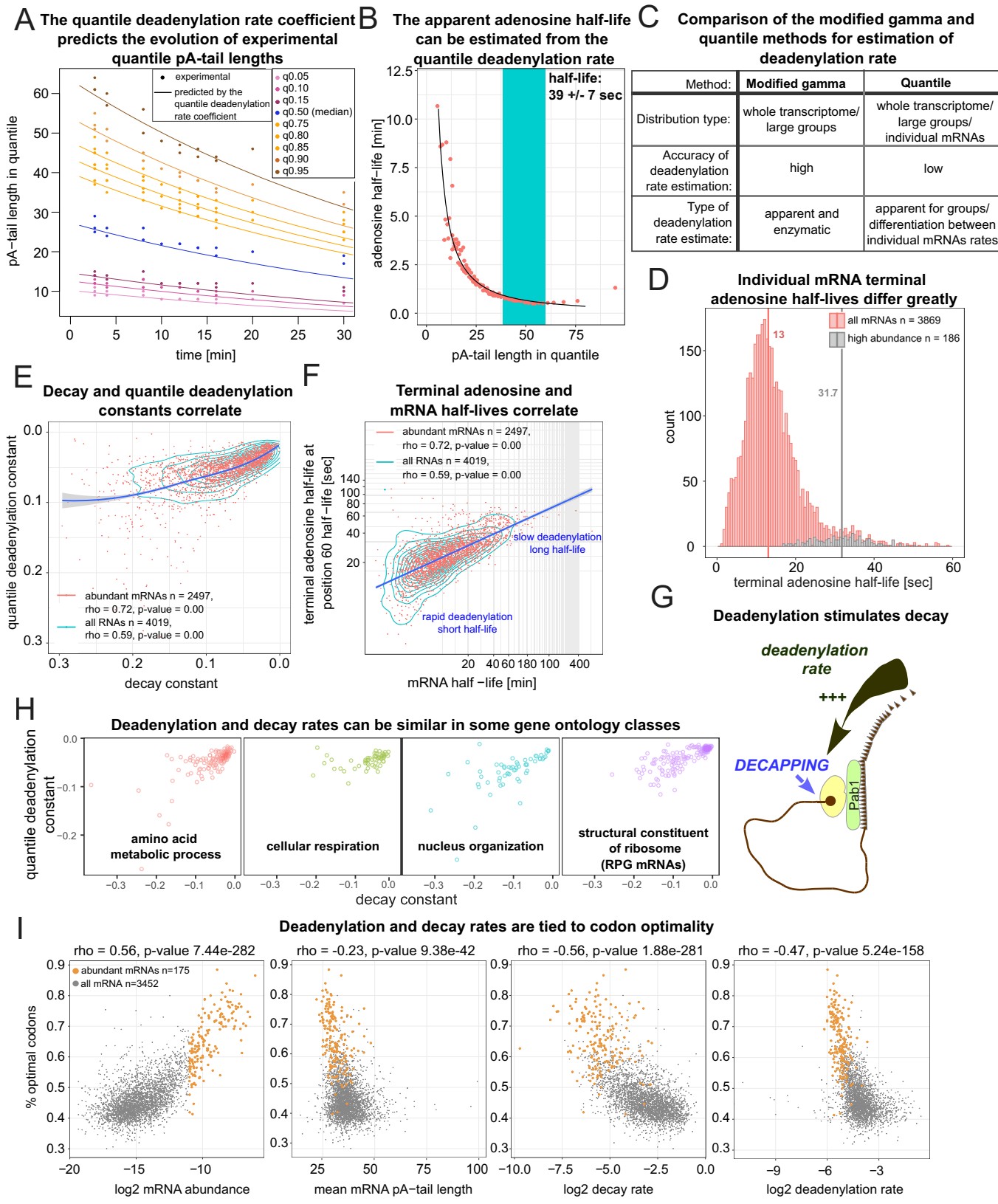

**Figure 5.  At steady-state, deadenylation and decay rates correlate.**

(**A**) Time-dependent change in quantile pA-tail lengths (as color-coded by quantile dots: upper—75-80-85-90-95[th]; median—50[th]; and lower—15-10-5[th]) for the entire coding transcriptome distribution in each replicate. Fitted continuous lines represent the quantile deadenylation coefficient calculated based on the upper quantiles. (**B**) Distribution of adenosine half-lives relative to pA-tail length in each quantile. The terminal adenosine half-life of $39 +/- 7$ s was calculated by averaging the half-lives obtained from quantiles 75-95[th] (blue area). (**C**) Table summarizes key differences between the modified gamma and quantile methods for calculating deadenylation rate. (**D**) Histogram displaying the distribution of single mRNA terminal adenosine half-lives for the entire coding transcriptome (light-red bars) or mRNAs of high abundance (gray bars). Vertical lines indicate the group median. (**E, F**) Comparison of decay and quantile deadenylation rates (**E**) or log2-scaled mRNA half-life to terminal adenosine half-life (**F**). Density plot represents all transcripts, whereas mRNAs with at least 70 reads in replicate A are shown as individual dots with a blue regression line. Spearman's rho correlations and p-values were calculated separately for each set using the rstatix package in R. (**G**) Schematic highlighting the strong, potentially causal link between deadenylation and decapping inferred from data presented in (**E, F**). (**H**) Comparison of decay and quantile deadenylation rates for various gene ontology groups (see also Fig. EV4D). (**I**) Percentage of optimal codons compared to: log2 mRNA abundance, mean mRNA pA-tail length, log2 decay, or quantile deadenylation rates. Source data are available online for this figure.

## Deadenylation and decay rates are positively correlated, and are linked to transcript functions and codon optimality

Our next step was to compare deadenylation rates among individual mRNAs. We averaged the deadenylation coefficients derived from the upper quantiles (75–80–85–90–95th), as these provided the highest estimates across the whole transcriptome (Fig. EV3F). This method accurately predicted changes in pA-tail length across all quantiles for single mRNA species (Fig. EV4A). We calculated the quantile half-lives of terminal adenosines for single mRNAs (Dataset EV1) and found a median of ~13 s (Fig. 5D). Remarkably, this median half-life was 32 s for the 186 most abundant mRNAs (Fig. 5D). We concluded that the transcriptomic deadenylation rate of 38 s is primarily dictated by a small subset of transcripts (see bar plot in Fig. 2C). Indeed, there was a strong correlation between quantile deadenylation rate and mRNA abundance (Fig. EV4B), though not with the mRNA mean pA-tail length (Fig. EV4C). Finally, we observed a strong correlation between decay and quantile deadenylation rates, with a Spearman rho coefficient as high as 0.72 (Fig. 5E,F), suggesting a causal relationship between both processes (Fig. 5G). In summary, we found that deadenylation rates, similar to mRNA half-lives, varied greatly between transcripts, with highly abundant mRNAs generally undergoing slower deadenylation and decay compared to less abundant transcripts, which were predominantly short-lived and underwent rapid tail removal.

Subsequently, we explored how the distributions of decay and quantile deadenylation rates were shaped across different functional gene ontology (GO) clusters. mRNAs coding for housekeeping genes such as RPGs, factors responsible for amino-acid metabolic process, nucleus organization, and cellular respiration exhibited notably low decay and quantile deadenylation rates (Figs. 5H and EV4D). In contrast, most other GO term groups displayed more varied and generally rapid deadenylation and decay rates (Fig. EV4D), particularly mRNAs involved in regulating mitotic and meiotic events. This illustrates that deadenylation and decay processes in distinct functional mRNA groups can be coordinated.

Prior research postulated that translation rates govern the speed of mRNA degradation. In particular, mRNAs containing rare codons are deadenylated rapidly. In contrast, those with optimal codons display slower deadenylation rates (Presnyak et al, 2015; Harigaya and Parker, 2016; Radhakrishnan et al, 2016). Indeed, both decay and quantile deadenylation rates showed nonlinear correlations with codon optimality, with Spearman rho coefficients of 0.56 and 0.47, respectively (Fig. 5I). Subsequently, we compared our findings with the translation model developed by Siwiak and Zielenkiewicz (2010). We observed that quantile deadenylation and decay rates were less correlated with ribosome density than with the time required to translate a single mRNA (Fig. EV4E). This supports the hypotheses that slower translation, due to the presence of rare codons, can accelerate decay (Presnyak et al, 2015; Radhakrishnan et al, 2016; Webster et al, 2018), extending this effect to the deadenylation step. Off note, codon optimality is also positively correlated with mRNA level (Fig. 5I). Thus, abundant, mainly housekeeping transcripts, are easily translated, which slows their deadenylation, extends their productive lifespan, and ultimately increases protein production.

## Deadenylation rate is not the dominant factor dictating decay of RPG mRNAs

Under optimal growth conditions, decay and deadenylation rates were strongly correlated (Fig. 5E–G). To explore potential functional and causative relationships further, we examined yeast cells under stress conditions (Fig. 6A), wherein RNA degradation plays a prominent role (Bresson et al, 2017; Marguerat et al, 2014). We conducted two experiments: a heat stress chase and a thiolutin chase. During the heat stress experiment, certain housekeeping genes, including RPGs, are transcriptionally silenced, while others, like those encoding chaperone mRNAs, become active (Vinaya-chandran et al, 2018). Thiolutin is a transcription inhibitor, which, at low doses, is known to activate the transcription of genes related to stress response (Adams and Gross, 1991). In the heat stress experiment, we quickly raised the temperature from 25 °C to 37 °C and monitored mRNA for up to 20 min (Fig. 6A). We confirmed the expected increase in chaperone mRNAs under both stress conditions by reverse transcription coupled to qPCR and in the DRS datasets (Appendix Fig. S2A,B).

In the Mex67-AA chase experiment, we analyzed approximately 125 RPG mRNA transcripts and observed similar deadenylation and decay rates among them (Fig. 5H). By aggregating the data for all RPG transcripts, we generated common pA-tail density profiles under various stress conditions (Appendix Fig. S2C–E). Significant differences in the overall RPG mRNA deadenylation rates prompted us to calculate decay and deadenylation rates for both RPG and non-RPG mRNAs in two new chase experiments. We presented those as density plots showing the half-lives of mRNA and terminal adenosines (Fig. 6B,C). In addition, we plotted the

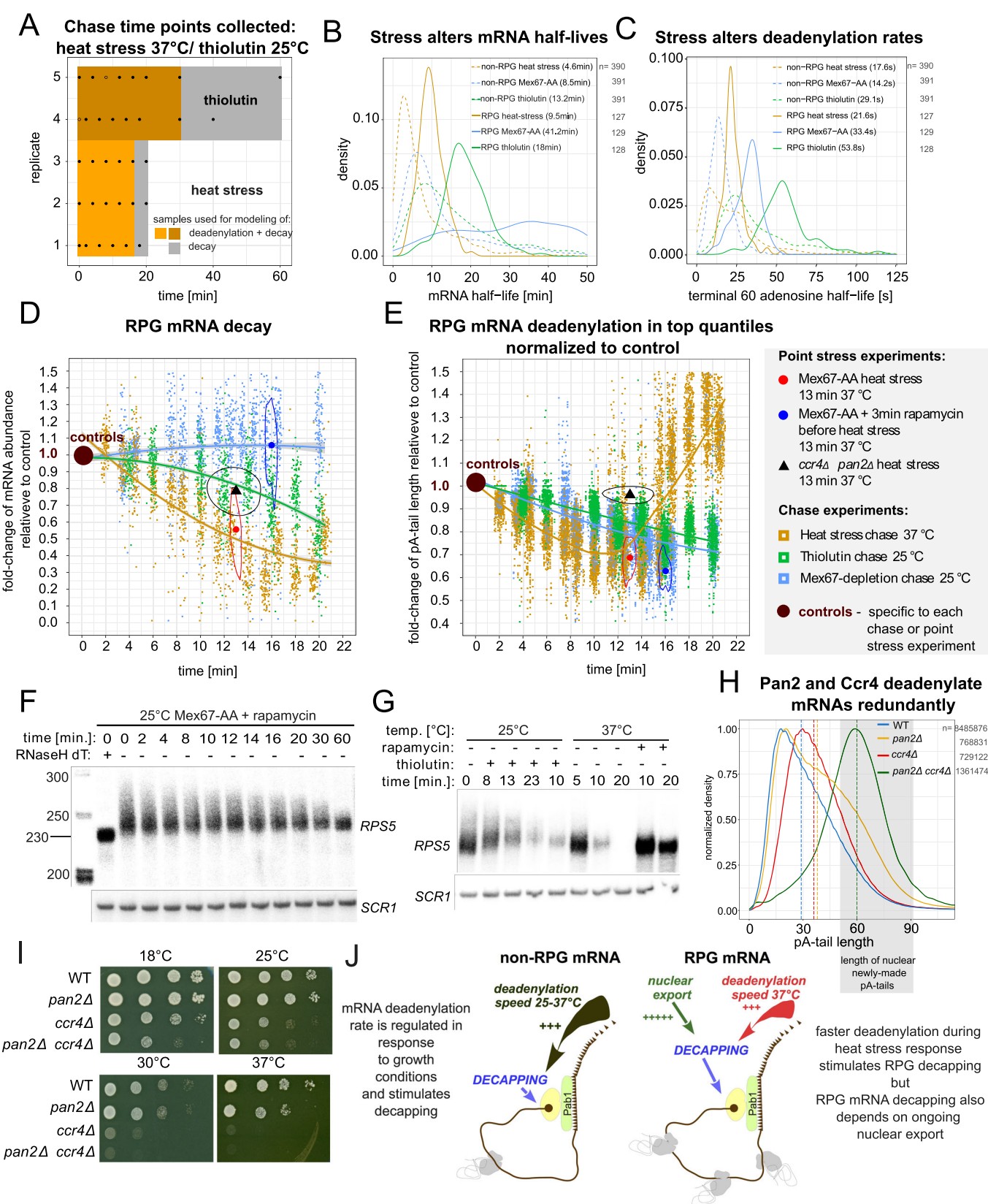

Figure 6. Deadenylation stimulates decay but is not a prerequisite for RPG mRNA decapping under stress conditions.

(A) Time points collected for the thiolutin 25 °C and heat stress 37 °C chase experiments. Colored surfaces designate the utility of each time frame. Dot contours represent data points excluded from the analysis. (B, C) Distribution of mRNA half-lives (B) or terminal adenosine half-lives (C) in Mex67 depletion, thiolutin, and heat stress chase datasets for RPG and non-RPG mRNAs transcriptionally down-regulated during heat stress according to Vinayachandran et al (2018). The number of estimates in each group is indicated next to the figure legend. (D) Time-dependent change in levels of RPG mRNAs in Mex67 depletion, thiolutin, and heat stress time course normalized to control. Fitted lines show the group trend with a 95% confidence interval in gray. Data points normalized to corresponding controls obtained for 13 min heat stress of a double *ccr4Δ pan2Δ* mutant (black triangle) or Mex67-AA strain treated or not with rapamycin 3 min prior to heat stress for 13 min at 37 °C. For these data, the median is shown along with the data-point density contour. The experiment legend is listed next to panel 6E. (E) Time-dependent changes in pA-tail length values of the 95–90–85–80–75$^{th}$ quantiles normalized to the control sample for the same samples as in (D). Normalization allows direct comparison of all changes within the group without grouping transcripts by control pA-tail length. (F, G) Northern blot showing *RPS5* levels during a Mex67-depletion time course (F) or heat stress at 37 °C and thiolutin treatment at 25 °C (G). (H) Normalized pA-tail density plot showing the whole-transcriptome adenylation profiles in wild-type cells compared to single *ccr4Δ* or *pan2Δ* mutants and a double *ccr4Δ pan2Δ* strain. The gray box shows the range of adenylation produced de novo in the nucleus by the polyA-polymerase Pap1 on pre-mRNAs (Turtola et al, 2021). (I) Spot tests comparing wild-type cells to *ccr4Δ*, *pan2Δ*, and double *ccr4Δ pan2Δ* mutant at indicated temperatures. (J) Scheme illustrating the role of deadenylation in mRNA decapping and decay. For most mRNAs, deadenylation is a rate-limiting factor that dictates the onset of decapping. RPG transcripts are a special group for which deadenylation can accelerate decay in conditions such as heat stress, but ultimately, decapping and decay are activated by an unknown factor linked to nuclear export. Source data are available online for this figure.

raw data underpinning our calculations, including the time-dependent changes in RPG abundance (Fig. 6D) and changes in upper quantile pA-tail lengths, both normalized to control conditions (Fig. 6E). These analyses revealed accelerated deadenylation and decay of RPG and non-RPG mRNAs under heat stress compared to other conditions, as shown by specific examples (Appendix Fig. S2F; Fig. EV5A). Validation of these findings for selected RPG mRNAs was performed using Northern blotting and quantitative PCR (Fig. 6F,G; Appendix Fig. S2F), confirming that the rates of mRNA deadenylation and decay can be modulated in response to changing growth conditions. However, these chase experiments did not conclusively determine whether deadenylation is the primary factor defining the decay rate.

To delve deeper into the functional relationship between deadenylation and decapping/decay, we investigated the relative catalytic contributions of the three deadenylase catalytic subunits expressed in budding yeast cells. The Pan2/3 complex harbors one enzyme, Pan2, whereas the multisubunit Ccr4-NOT complex contains two deadenylases, Ccr4 and Pop2. We examined adenylation changes across the entire transcriptome profile and found that pA-tail lengths exhibited greater alterations in *pan2Δ* or *ccr4Δ* cells (on average by more than seven As for both; Tudek et al, 2021) than in the *pop2Δ* mutant (1.9 As; Fig. EV5B). Importantly, pA-tail length changes in *pop2Δ* and *ccr4Δ*, rather than *pan2Δ* and *ccr4Δ*, were highly correlated (Fig. EV5C), clearly indicating that Pop2 and Ccr4 both target similar transcripts within the same complex, even though Pop2's contribution to deadenylation was minor. Therefore, we conclude that Ccr4 and Pan2 are the primary cellular deadenylases in budding yeast (Tudek et al, 2021).

Next, we investigated the roles of Ccr4 and Pan2 in RPG mRNA deadenylation and decay during heat stress. Under steady-state conditions, RPG mRNA pA-tails were longer in *pan2Δ* cells compared to *ccr4Δ* (Fig. 6H; Tudek et al, 2021), suggesting these mRNAs are often targeted by the Pan2/3 complex. However, in a double *pan2Δ ccr4Δ* strain, mean pA-tail lengths were ~60.73 A, with the 5$^{th}$ and 95$^{th}$ quantiles at 40.08 and 90.89 adenosines, respectively (Fig. 6H). We concluded that very little, if any, deadenylation occurs in the cytoplasm of a double *pan2Δ ccr4Δ* strain because previous research indicated that the newly produced pA-tail in the nucleus are 60 As long with fail-safe mechanisms that limit adenylation by Pap1 to 90 adenosines (Turtola et al, 2021). We further tested this conclusion by comparing pA-tail lengths of a

*ccr4Δ pan2Δ* mutants grown under steady-state conditions to those after a 13 min exposure to heat stress. While wild-type RPG pA-tails were significantly shortened compared to controls (Fig. 6E: yellow chase curve), minimal deadenylation levels were observed in the double *pan2Δ ccr4Δ* strain (Fig. 6E: compare black triangle to control; some underlying raw data Fig. EV5D). This confirmed that both deadenylases are required to shorten RPG mRNAs pA-tails during heat stress. Strikingly, RPG mRNAs in heat-shocked *pan2Δ ccr4Δ* cells were still degraded. The decay rate was similar to conditions in which wild-type cells were treated with thiolutin at 25 °C (Fig. 6D; compare the black triangle designating the *ccr4Δ pan2Δ* mutant to the green thiolutin chase curve). This observation suggested that decapping and subsequent decay can occur on long-tailed mRNAs, challenging the strictly causal link between mRNA deadenylation and decay. We also found that RPG mRNA decay was delayed in both single *pan2Δ* or *ccr4Δ* mutant strains, heat-stressed for 13 min (Fig. EV5E), further confirming that both deadenylase complexes contribute independently to the decay process. In contrast, the loss of *POP2* had no notable effects on RPG mRNA decay dynamics, underscoring the minor role of this deadenylase in decapping/decay (Fig. EV5E). Thus, we concluded that in a wild-type strain, accelerated deadenylation by either Pan2 or Ccr4 stimulates RPG mRNA decay under heat stress, however it is not critical for decapping and degradation. Notably, the observation that decay is not completely inhibited when deadenylation is blocked explains the viability of the double *ccr4Δ pan2Δ* mutant (Fig. 6I).

Finally, we explored additional factors potentially triggering RPG mRNA decay beyond deadenylation. Initially, we considered the role of exosome-mediated 3'–5' decay that could be triggered by changes in the translation machinery at high temperatures. However, we ruled this out because the deletion of the exosome-associated helicase Ski2 did not inhibit RPG mRNA decay (Fig. EV5E; Tomecki et al, 2023). Instead, we observed that RPG mRNA decay under heat stress was strongly inhibited by the depletion of Dcp2 (Fig. EV5F) as well as the deletion of the decapping co-factors *DHH1* and *LSM1* (Fig. EV5E). Next, we investigated whether mRNA export affects decay by blocking export 3 min before applying heat stress. We observed that the decay of most RPG mRNAs was completely inhibited under these conditions (Fig. 6D: density around the median blue dot clustered at the level of its control, similarly to the entire Mex67-AA chase

marked with a blue line; Figs. 6D and EV5F). This was surprising, as only a few RPG mRNAs decayed efficiently without export (e.g., *RPL4A*; Fig. EV5G). However, deadenylation of RPG mRNAs was accelerated under heat stress regardless of whether cells were export-blocked or not (Fig. 6E: compare density clusters around the blue and red dot representing export-blocked and wild-type cells to the heat stress chase curve marked in yellow).

In sum, we showed that during the heat response for RPG mRNA, both deadenylation and degradation are enhanced. However, under these specific conditions, deadenylation is not essential for decay. This deadenylation-independent pathway is strictly dependent on ongoing mRNA export. Altogether, our results suggest that whereas at the transcriptome-wide level, deadenylation and decay are strongly linked, functionally they can be uncoupled under specific stress conditions (Figs. 5E,F and 6J).

## Discussion

The causal link between deadenylation and decapping is axiomatic to the cytoplasmic mRNA decay model. There are many transcriptomic estimates of mRNA decapping and degradation rates in yeast (Miller et al, 2011; Neymotin et al, 2014; Chan et al, 2018; Presnyak et al, 2015), however, attempts to model deadenylation are rare and limited to higher eukaryotes (Eisen et al, 2020). Here, we provide the first numerical model for deadenylation in budding yeast, which enabled high-precision estimation of the transcriptomic deadenylation rate of 10 A/min.

Our numerical model for deadenylation-dependent decay is based on a modified gamma distribution, used to describe experimental pA-tail length profiles. The $tanh(0.096·N_A)$ formula inserted into the classical gamma distribution accounted for both the biological effects of the Pab1 function in protecting mRNA from decapping and in stimulating deadenylation. Specifically, stimulation of deadenylation occurs when multimeric Pab1 binds to long polyadenosine tails, whereas the protective function associated with monomeric Pab1 is lost when pA-tails are shorter than 20 As. This model is supported by previous in vitro and structural data, along with early in vivo studies performed on reporter mRNAs (Schäfer et al, 2019; Yi et al, 2018; Webster et al, 2018; Caponigro and Parker, 1995). In our study, we confirmed the essential role of Pab1 in mRNA deadenylation in vivo. The decapping onset set at a pA-tail length of 20 As is consistent with previous estimates in higher eukaryotes derived from PAL-seq2 data (Eisen et al, 2020), which we independently validated using Dcp2 or Xrn1 degron strains. Importantly, this does not negate the occurrence of deadenylation-independent decapping on specific mRNAs, particularly those that are translationally compromised (Tomecki et al, 2023). However, such aberrant transcripts are scarce compared to the functional coding transcriptome and do not constitute a mass large enough to distort the whole-transcriptome deadenylation profile. Therefore, the model does not account for those mRNA quality control events but does not exclude their existence.

Our model reveals a constant transcriptomic deadenylation rate of ten adenines per minute. This rate represents the average across the transcriptome, although individual transcripts can exhibit varying deadenylation rates. This variability is best highlighted by our in silico deadenylation simulation using a narrow pA-tail distribution (akin to a short transcription burst), which shows a gradual increase in distribution variance over time (Fig. EV5H). Although our model does not explicitly differentiate between individual mRNAs, this simulation highlights the potential for transcript-specific deadenylation rates. Notably, the precise role of Pab1 in deadenylation, particularly the function of each RNA recognition motif, cannot be fully modeled due to these complexities. The deadenylation rate holds significant implications for mRNA decapping-decay rates. For the first time we show that the in vivo time required to sufficiently shorten the pA-tail to remove the last Pab1 can explain mRNA half-life times (decapping onset) in budding yeast. Newly synthesized pA-tails typically range from 60 adenosines to a maximum of 200 (Tudek et al, 2021; Turtola et al, 2021). Decapping is likely induced on tails shorter than 20 adenosines, implying that a minimum of 50–40 adenosines must be removed before decapping, a process that likely occurs within minutes. This time frame aligns with the median mRNA half-life of 9 min, suggesting that deadenylation may be slightly faster than decay. Any differences may stem from the time required to export the newly synthesized mRNA out of the nucleus, which effectively extends its lifetime (Dargemont and Kühn, 1992; Mor et al, 2010; Bahar Halpern et al, 2015), although technical considerations cannot be ruled out.

In our study, we also present a simple method for calculating mRNA deadenylation rates based on quantiles of the pA-tail distribution. This approach allows for the estimation of deadenylation rates from small datasets (representing the pA-tail length distributions of individual transcripts) with an accuracy comparable to that of the modified gamma numerical model. This method was instrumental in unveiling several important biological observations. We established the link between codon optimality and deadenylation at the transcriptomic level, which has previously only been observed in reporter mRNAs (Presnyak et al, 2015). We also explored the longstanding hypothesis regarding a causal link between deadenylation and decapping using a model group of RPG mRNAs. We showed that under heat stress, deadenylation can accelerate, thereby shortening mRNA half-lives, which supports the classic deadenylation-decapping model. However, we also showed that deadenylation is not a prerequisite for decapping/decay, but rather serves as a major stimulatory factor. This is supported by our comprehensive analyses under stress conditions, revealing that inhibition of cytoplasmic deadenylation can lead to decapping on mRNAs with long pA-tails, suggesting the existence of additional regulatory mechanisms that can induce decapping independent of deadenylation. Specifically, for RPG mRNAs, one such mechanism appears to be linked to ongoing nuclear export. Clearly, additional research is necessary to unveil the molecular mechanisms governing the regulation of deadenylation and decapping rates. Although some mechanisms may be temperature-sensitive, where a rise in temperature boosts any enzymes' activity, others are presumably specific to transcript groups. In particular, RPG mRNAs and many other housekeeping transcripts exhibit longer lifespans compared to the transcriptomic median (Miller et al, 2011; Neymotin et al, 2014; Chan et al, 2018; Presnyak et al, 2015). Preserving the lifespan of mRNAs encoding essential cellular components, regardless of growth conditions, is likely advantageous to reduce the energy cost associated with mRNA synthesis.

# Methods

### Reagents and tools table

| Reagent/resource | Reference or source | Identifier or catalog number |
| --- | --- | --- |
| **Experimental models** | | |
| *Saccharomyces cerevisiae* wild-type W303 and its derivatives | Appendix Table S1 page 10 | Appendix Table S1 page 10 |
| **Recombinant DNA** | | |
| pUC57 | Addgene | https://www.addgene.org/vector-database/4509/ |
| pET-28 b (+) | Addgene | https://www.addgene.org/vector-database/2566/ |
| **Antibodies** | | |
| Anti-FLAG | Sigma | F1804 (RRID:AB_262044) |
| Anti-PGK1 | Novex LifeTechnology | discontinued |
| Anti-Pab1 | Santa Cruz | sc57953 (RRID:AB_672248) |
| Anti-Rpb3: 1Y26 | Abcam | ab81859 |
| Goat Anti-Mouse IgG, H&L Chain Specific Peroxidase Conjugate | Merck | 401215-2 ML |
| Goat Anti-Rabbit IgG, H&L Chain Specific Peroxidase Conjugate | Merck | 401393-2 ML |
| **Oligonucleotides and other sequence-based reagents** | | |
| Oligonucleotide list | Appendix Table S2 page 11 | Appendix Table S2 page 11 |
| **Chemicals, enzymes, and other reagents** | | |
| Auxin sodium salt | Merck | I5148-2G |
| Chloroform was added | Sigma | C2432 |
| Dynabeads™ Oligo(dT)$_{25}$ | LifeTechnology | 61005 |
| 1-Naphthaleneacetic acid (1-NAA) | Merck | N0640-25G |
| Phenol solution saturated with 0.1 M citrate at pH 4.3 | Sigma | P4682 |
| Platinum SYBR™ Green qPCR SuperMix-UDG | LifeTechnology | 11733046 |
| Rapamycin | Cayman Chemicals | 13346 |
| RNase A | LifeTechnology | R1253 |
| RNase H | New England Biolabs | M0523 |
| RNase T1 | LifeTechnology | EN0542 |
| RT qPCR Mix SYBR | A&A Biotechnology | 2008-1000 |
| RiboLock RNase inhibitor | LifeTechnology | EO 0382 |
| SuperScript IV reverse transcriptase | LifeTechnologies | 18090050 |
| Thiolutin | Sigma | T3450 |
| T4 RNA Ligase | LifeTechnologies | EL0021 |

| Reagent/resource | Reference or source | Identifier or catalog number |
| --- | --- | --- |
| ULTRA-Hyb Oligo Hybridization buffer | Invitrogen | AM8663 |
| [5'-32P]pCp (3000 Ci/mmol, 10 mCi/ml) | Hartmann Analytic | https://www.hartmann-analytic.de/gamma-p32-atp-3000-ci-mmol-10-mci-ml-fp-301.html |
| **Software** | | |
| Code to estimate the γ_shape and γ_rate parameters of the modified gamma distribution based on experimental distributions | Mendeley data and Appendix section 5 page 27 | https://doi.org/10.17632/2j3hh37zzs.1 and Appendix section 5 page 27 |
| Guppy | Oxford Nanopore Technology (ONT) | versions listed in Dataset EV2 (4.4.1; 5.0.11; 6.0.0) |
| ImageJ software | https://imagej.net/ | versin 1.50i version 1.54j |
| Inkscape | https://inkscape.org/ | version 1.3.2 |
| Minimap2 | https://github.com/lh3/minmap2 | version 2.17 |
| Nanopolish polya | https://github.com/jts/nanopolish | versions listed in Dataset EV2 |
| Origin | www.originlab.com | version 10.0 |
| R | The R Foundation for Statistical Computing https://www.r-project.org/ | version 3.6.2 (2019-12-12) version R-4.2.1 (2022-06-23) version R-4.4.1 (2024-06-14) |
| RStudio | https://posit.co/products/open-source/rstudio/ | version 1.2.5033 for Mac, 4.2.3, RStudio Team, 2019 version 2024.042 build 764 for Windows, Posit team, 2024 |
| **Other** | | |
| Direct RNA Sequencing kit | Oxford Nanopore Technology (ONT) | SQK-RNA002 |
| FLA 9000 scanner | N/A | N/A |
| Hybond-N+ membrane | GE Healthcare | RPN303B |
| LightCycler LC480 Roche | Roche | N/A |
| Nitrocellulose membrane | Bio-Rad | 1620112 |
| PhosphorImager screen | FujiFilm | N/A |
| Whatman paper | Bio-Rad | 1703967 |

## Yeast culture conditions

Yeast cultures were prepared in YPDA media. The Mex67-AA and thiolutin chase experiments were conducted at 25 °C. To deplete Mex67 using the Anchor-Away tag (Haruki et al, 2008) rapamycin (Cayman Chemicals cat. no. 13346) was added to a final concentration of 1 µg/ml. The heat stress chase experiment was

performed by pre-culturing cells at 25 °C and adding an equal volume of media pre-heated to 51 °C, resulting in a final temperature of 37 °C, which was maintained by culture incubation in a waterbath. Treatment with thiolutin (Sigma; T3450) was performed by adding the compound to a final concentration of 4 ug/ml. Samples for all chase experiments were collected by mixing the cell culture with an equal volume of ice-cold ethanol, which inactivates cellular metabolism. To deplete Xrn1 and Dcp2 using AID tags (Nishimura et al, 2009; Morawska and Ulrich, 2013), 1-Naphthaleneacetic acid (1-NAA; N0640-25G, Merck) was added to a final concentration of 1 mM. Pab1 depletion using the AID system involved adding auxin sodium salt (I5148-2G; Merck) to a final concentration of 1–3 mM. Yeast strains used in this study are listed in Appendix Table S1.

## RNA extraction

RNA extraction was performed using the hot acid phenol method. Cell pellets were resuspended in 400 µl phenol solution saturated with 0.1 M citrate at pH 4.3 (Sigma; P4682), followed by the addition of 400 µl of TES buffer (10 mM Tris pH 7.5, 5 mM EDTA, 1% SDS). Samples were vortexed for 45 min at 65 °C, then centrifuged at 4 °C for 10 min. The supernatant was transferred to a fresh tube, and 400 µl phenol solution saturated with 0.1 M citrate at pH 4.3 was added. The samples were again vortexed for 20 min at 65 °C and then centrifuged at 4 °C for 10 min. The supernatant was transferred to a fresh tube and 400 µl of chloroform was added (C2432; Sigma). The samples were briefly vortexed at room temperature to remove phenol, and centrifuged at 4 °C for 10 min. The supernatant was transferred to a fresh tube; 45 µl of 2 M LiCl was added and 1 ml of 95% ethanol. Samples were precipitated at −80 °C for at least 30 min, then centrifuged at 4 °C for 25 min, washed with 400 µl of 80% ethanol, and dried at 37 °C after removing the supernatant. Pellets were resuspended in nuclease-free water, and RNA concentration was measured using a Nanodrop apparatus.

## Enrichment of the pA$^+$ fraction for sequencing library preparation and qPCR analyses

The pA$^+$ fraction was prepared using magnetic beads coupled to oligo-dT from LifeTechnologies (61005). 35 µg of total RNA was resuspended in 50 µl of nuclease-free water. The RNA was mixed with 50 µl of binding buffer (20 mM Tris-HCl, ph 7.5, 1 M LiCl, 2 mM EDTA) and denatured for 2 min at 65 °C before cooling on ice. In total, 100 µl of slurry beads per 35 µg of total RNA was pre-washed three times in 1 ml of binding buffer and resuspended in 50 µl of binding buffer per sample. The beads were added to the denatured RNA and incubated at room temperature with occasional shaking for 20 min. The supernatant was removed, and beads were washed twice with 100 µl of wash buffer (10 mM Tris-HCl pH 7.5, 150 mM LiCl, 1 mM EDTA) and after removing any remnants of wash buffer, resuspended in 12 µl of nuclease-free water. The beads were then incubated for 2 min at 80 °C, and the supernatant removed from the beads was utilized for sequencing library preparation and/or qPCR analyses as the pA$^+$ fraction. Oligonucleotides used in this study are listed in Appendix Table S2.

## Nanopore sequencing

DRS libraries were prepared using a Direct RNA Sequencing kit (ONT—Oxford Nanopore Technology, SQK-RNA002) according to the manufacturer's instructions, using 50–200 ng oligo-dT$_{(25)}$-enriched mRNA from *Saccharomyces cerevisiae* yeast mixed with cap-enriched or total RNA from other organisms (human, mouse, *A. thaliana*, or *C. elegans*), as described in Bilska et al (2020). Raw data were basecalled using Guppy (ONT). The raw sequencing data files (fast5) were deposited at the European Nucleotide Archive (ENA; for a list of accession numbers, see Dataset EV2).

## Bioinformatic analysis

### pA-tail length determination

DRS reads were mapped to the custom *S. cerevisiae* transcriptome described in Tudek et al (2021) using Minimap2 2.17 with options -k 14 -ax map-ont –secondary = no. These alignments were processed with samtools 1.9 to filter out supplementary alignments and reads mapping to the reverse strand (samtools view -b -F 2320). Any unmapped reads were filtered out and discarded from analysis. Poly(A) tail lengths for each read were estimated using the Nanopolish 0.13.2 polya function and only length estimates with the QC tag reported as PASS were considered in subsequent analyses (see "Quality Control details" in Appendix). Since the replicates strongly correlated with one another, unless otherwise indicated, samples from the same condition were analyzed together. Changes in mean/median pA-tail length and mRNA abundance were analyzed using R. Tables with the number of counts, mean, median, geometric mean pA-tail lengths, and quantiles are deposited at Mendeley data (https://doi.org/10.17632/2j3hh37zzs.1).

## Modeling of the transcriptomic pA-tail distribution using the modified gamma distribution

### Modification to the standard gamma distribution

The numeric model is based on the standard two-parameter gamma distribution, which has been widely used in life-testing for decades (Gupta and Groll, 1961):

$$p(x) \sim x^a \cdot e^{-b \cdot x} \tag{1}$$

where $b$ reflects the rate of asymptotic exponential decay of the analyzed process, and $a$ characterizes the shape of the distribution. The mean of the distribution is $(a + 1)/b$, and its maximum is located at $a/b$ (note that $x$ does not necessarily denote time flow).

The pre-exponent factor in the model was further modified to quantitatively describe the limited contribution of RNA protection by the polyA-binding protein Pab1. We assumed that Pab1 recognizes n succeeding adenosine residues, with the free energy of interaction being identical and additive for each residue. This assumption implies an interaction with indistinguishable independent objects. Such a model has already been developed for the analogous problem of the n-spin system in the Mean Field Theory (Albert and Barabási, 2002; Kadanoff, 2009) in the form of:

$$p(n) \sim tanh(\beta \cdot n) \tag{2}$$

where $n$ is the number of objects, and $\beta$ characterizes the strength of individual interaction with the external field (here Pab1); formally, $\beta = \Delta G/RT$, where $\Delta G$ is the free energy of interaction, $T$ is temperature, and $R$ is the gas constant (8.13 J·K/mol). Combining the two above equations leads to the three-parameter modified gamma distribution expressed as:

$$p(i) \sim (tanh(\beta \cdot i))^{\gamma\_shape} \cdot e^{-\gamma rate \cdot i} \qquad (3)$$

where $i$ is the number of adenosine residues in the pA-tail, and $\beta$ was roughly estimated at 0.096. The latter number implies that ~20 succeeding adenosine residues saturate the interaction ($tanh(20^*\ 0.96) = 0.96$). As such, Pab1 recognizes approximately 20 residues of the pA-tail (see Fig. 4A), consistent with previous in vitro footprint experiments (Webster et al, 2018; Baer and Kornberg, 1980; Schäfer et al, 2019). The $\gamma\_rate$ parameter should be interpreted as the apparent asymptotic proportion within the pA-tail, where $p = e^{\gamma\_rate}$ represents the ratio of abundance between two pA-tails differing in length by a single adenosine residue. There is no direct interpretation for the $\gamma\_shape$ parameter. However, comparing the gamma distribution to the Erlang one (in which parameter $a$ in Eq. (1) is an integer), $\gamma\_shape + 1$ in Eq. (3) roughly reflects the number of critical events leading to RNA degradation (e.g., removal of individual Pab1, a relatively slow process, and rapid decapping coupled to degradation).

### Estimation of γ_shape and γ_rate parameters from experimental distributions

For each experimental pA-tail distribution from the Mex67-depletion chase, the parameters of Eq. (3) (i.e., $\gamma\_shape$ and $\gamma\_rate$) were fitted using the Nonlinear Least Squares procedure (nls) implemented in R (version 4.2.3; www.r-project.org). The procedure code in R and the table containing the list of modified gamma parameters are deposited at Mendeley data (https://doi.org/10.17632/2j3hh37zzs.1) and also included in the publication's Appendix file (Appendix Table S3). The graphical output generated as a pdf file is exclusively deposited at Mendeley. Please refer to the instruction in the Appendix section 1.8 for further code modifications necessary to adapt the code to other datasets. The time dependence of the $\gamma\_shape$ and $\gamma\_rate$ parameters was further analyzed in R using a linear model. In addition, the time evolution of the pA-tail distributions was illustrated as a 3D graph using Origin (version 10.0; www.originlab.com), assuming a linear time dependence of the $\gamma\_shape$ and $\gamma\_rate$ parameters.

### Estimation of newly made mRNA pA-tail profiles

New mRNA pA-tail profiles, depicted as blue lines in Figs. 4C and EV2B, were generated by subtracting the modified gamma distribution from the experimental pA-tail profile and subsequent log2-fitting. This procedure was applied to all Mex67-AA distributions but yielded significant estimates of new mRNA production only in control samples (compare control distributions in Fig. 4C to profiles obtained after Mex67 depletion in EV2B and refer to graphs deposited at Mendeley data: https://doi.org/10.17632/2j3hh37zzs.1). This confirms the previously observed decrease in transcript synthesis determined using 4tU-labeling (Tudek et al, 2018; Schmid et al, 2018).

### Modeling the pA-tail degradation kinetics

Let us consider a population of RNAs with specific of pA-tail lengths of $i$ equal to $x_i$, and assume that the microscopic deadenylation rate, $k$, is universal, i.e., does not substantially depend on either the length of a pA-tail or on time. The time evolution of the pA-tails' population can therefore be described by a system of $N$ differential equations, with initial conditions $x_i(t = 0) = n(i)$.

$$dx_i/dt = -k \cdot x_i + k \cdot x_{i+1} \qquad (4)$$

In the above Eq. (4), the factor '$-k \cdot x_i$' describes the deadenylation of a particular pA-tail of length $i$. In addition, the deadenylation of the pA-tail of the length $i + 1$ contributes to the apparent recovery of the pA-tail of the length $i$ (factor '$+k \cdot x_{i+1}$').

This system of differential equations can be solvable analytically, leading to the Eq. (5):

$$x_i(t) = e^{-\alpha} \cdot \sum_{j=i}^{N} (n(j) \cdot \alpha^{(j-i)}/(j - i)!) \qquad (5)$$

where the parameter $\alpha$, equaling $k \cdot t$, is the dimensionless scaling factor, and $N$ represents the maximal length of a pA-tail.

Equation (5) enabled the in silico evolution of any initial pA-tail distribution (Figs. EV3D and EV5H). However, according to Eq. (5), the accumulation of short-tail RNAs is expected, which is not observed experimentally due to an accelerated degradation of non-protected RNAs. The latter phenomenon is incorporated into the model as the consequence of the removal of the last Pab1, arbitrarily taken into account following Eq. (2):

$$x_i(t) = (tanh(\beta \cdot i))^{\alpha} \cdot e^{-\alpha} \cdot \sum_{j=i}^{N} (n_k \cdot \alpha^{(j-i)}/(j - i)!) \qquad (6)$$

The final model demonstrating the evolution of the pA-tail distribution is depicted in Figs. EV3D and EV5H. In Fig. EV3D, the initial distribution is the same as the one in Fig. EV1E replicate A. In Fig. EV5H, the model is tested on a theoretically sharp distribution akin to a short transcription burst. In Fig. EV3D, subsequent distributions resemble properties observed experimentally for replicate A. Comparing a series of uniformly sampled distributions (e.g., for $\alpha = 0, 1, 2, 3, 4,...$) evolved from the initial experimental distribution (at $t = 0/\alpha = 0$) with subsequent experimental ones (Fig. EV3E) facilitates the estimation of the microscopic rate $k$, denotes by the ratio $\alpha_{best}/t$, i.e., the slope of a line illustrated in Fig. EV3E, where $\alpha_{best}$ identifies the in silico-derived distribution closest to the experimental one.

### Relation between enzymatic microscopic and apparent deadenylation rates

Since the experimental pA-tail distribution can be sufficiently described by the modified gamma distribution (Eq. (3)), for long enough tails (i.e., where $tanh(\beta \cdot i) \approx 1$), the ratio of populations of two "subsequent" pA-tails, $n_{i+1}/n_i$, remains constant, equaling $p = e^{-\gamma\_rate}$. Under these assumptions, Eq. (6) can be simplified to:

$$x_i(t) = e^{-k \cdot t} \cdot \sum_{j=i}^{N} (n_i \cdot (p \cdot k \cdot t)^{(j-i)}/(j - i)!) \approx n_i \cdot e^{-k \cdot t \cdot (1-p)} \qquad (7)$$

Therefore, the exponential decay of the RNA population carrying a specific pA-tail length is expected with the apparent

decay rate, $k_{app}$, substantially lower than the microscopic rate of enzymatic deadenylation, k:

$$k_{app} = k \cdot (1 - p) = k \cdot (1 - e^{-\gamma rate}) \quad (8)$$

## Modeling of decay and quantile deadenylation rates

### Normalization of RNA abundance

Initially, sequencing data were normalized to mitigate the impact of unequal library sizes. *ENO2* transcript counts were removed from the raw datasets as in some datasets they were sourced by libraries unrelated to the project (used as artificially synthesized spike-in). The abundance of this transcript often introduced unwanted bias to RNA level quantification. For decay modeling, the transcript levels were normalized across chase datasets. First, absent raw counts (NA values) were substituted with 0.01. Second, count levels were adjusted based on library size. These data were used to calculate decay rates.

### Data quality control and filtering for decay and deadenylation modeling

The Appendix file, along with Appendix Figs. S3 and S4 and Appendix Table S4, details the quality control procedures and outlines the strategy for decay and quantile deadenylation. Briefly, data were filtered to calculate decay and quantile deadenylation rates. Initially, all quantile estimates derived from less than 10 counts were discarded due to low estimation quality for quantiles and medians. Next, all values representing mRNA abundance or quantile pA-tail length greater than the reference were removed. This approach is justified for two primary reasons. First, budding yeast do not code for cytoplasmic adenylases, rendering any increase in pA-tail length biologically irrelevant and likely arising for other processes. Under Mex67-depletion conditions, newly synthesized mRNAs predominantly undergo decay prior to Pap1-mediated adenylation. However, a very small fraction of de novo produced mRNAs may escape nuclear decay and acquire a long pA-tail in a process called hyperadenylation (Jensen et al, 2001). This is due to mis-regulation of Pap1 polyA-polymerase activity (Turtola et al, 2021). Although the share of these reads is negligible in short Mex67-depletion times, this phenomenon increases over time as the cytoplasmic mRNA pool gradually degrades. Similarly, during heat stress chase, mRNA levels and pA-tail lengths decrease following transcriptional shut-down, however beyond 10 min, a large pool of newly synthesized transcripts with long pA-tails can be observed following transcriptional resumption. These processes independently lead to a bi-modal distribution of pA-tail lengths, ultimately justifying the exclusion of certain samples from modeling (Figs. 1B, 2A and 6A; see Appendix).

To calculate decay and deadenylation rates under heat stress at 37 °C or thiolutin treatment at 25 °C, only mRNAs previously identified as transcriptionally down-regulated by RNAPII ChIP were selected. To this end, a reference list of such mRNAs was directly sourced from Vinayachandran et al (2018) and is deposited at Mendeley (https://doi.org/10.17632/2j3hh37zzs.1).

### Exponential modeling

Exponential modeling for decay rate estimation was previously used by other authors (Miller et al, 2011; Sun et al, 2012; Neymotin et al, 2014; Presnyak et al, 2015; Chan et al, 2018). The decay constant of the standard exponential decay formula was estimated from experimental data as the fraction degraded over a given time ($ln(L\_time\_x/L\_time\_x-1)/time$), where $L$ represents the pA-tail length or mRNA abundance. Decay and quantile deadenylation constants were independently calculated for each quantile in each chase replicate and subsequently averaged. Details concerning the modeling process are provided in the Appendix, including the rationale for merging replicates. When calculating decay rates for Mex67-AA chase datasets, control times were adjusted by 2.67 min (the time estimated by the modified gamma model required for manifestation of the Mex67-depletion phenotype). Dataset EV1 presents all decay and deadenylation constants for the three chase experiments. Throughout the publication, the "mean_mean" values were used. In addition, the tables used for these calculations can be accessed in Mendeley data (https://doi.org/10.17632/2j3hh37zzs.1).

## Data availability

The datasets produced in this study are available in the following databases: Basecalled nanopore sequencing data: Gene Expression Omnibus GSE272785, consult the Dataset EV2 for sample accession numbers); Raw nanopore sequencing data: European Nucleotide Archive (consult the Dataset EV2 for sample accession numbers); Source data underlying each figure panel, R code to calculate $\gamma\_rate$ and $\gamma\_shape$, are deposited at Mendeley Data, https://doi.org/10.17632/2j3hh37zzs.1. In addition, the R code is also listed in the Appendix section 5.

The source data of this paper are collected in the following database record: biostudies:S-SCDT-10_1038-S44318-024-00258-3.

## Peer review information

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

## Acknowledgements

This work was mainly supported by the National Science Centre Poland, the SONATA grant (2020/39/D/NZ2/02174 to AT), the TEAM/2016-1/3 Foundation for Polish Science grant (to AD), and HORIZON Europa ERC AdG (101097317 to AD). Work by RT was supported by National Science Centre Poland, SONATA BIS grant (2017/26/E/NZ1/00724), and by the National Centre for Research and Development (LIDER/35/46/L-3/11/NCBR/2012). SM's work was supported by the National Science Centre Poland grant (2020/38/E/NZ2/00372 to SM). MT was supported by the Academy Research Fellow grant (nos. 349698 and 353682) awarded by the Research Council of Finland. Work by MT in the laboratory of THJ was supported by a Federation of European Biochemical Societies long-term fellowship and an EMBO long-term fellowship (ALTF 328-2019). This research was performed thanks to the IIMCB IN-MOL-CELL Infrastructure funded by the European Union, co-financed under the European Funds for Smart Economy 2021-2027 (FENG) and the European Union— NextGenerationEU.

## Author contributions

**Agnieszka Czarnocka-Cieciura**: Data curation; Validation; Investigation; Visualization; Writing—review and editing. **Jarosław Poznański**: Conceptualization; Data curation; Software; Formal analysis; Visualization; Methodology. **Matti Turtola**: Resources; Funding acquisition; Validation; Investigation. **Rafał Tomecki**: Funding acquisition; Validation; Investigation. **Paweł S Krawczyk**: Data curation; Software; Supervision. **Seweryn Mroczek**: Supervision; Funding acquisition; Methodology. **Wiktoria Orzeł**: Investigation. **Upasana Saha**: Resources. **Torben Heick Jensen**: Supervision. **Andrzej Dziembowski**: Conceptualization; Supervision; Funding acquisition; Investigation; Methodology; Writing—review and editing. **Agnieszka Tudek**: Conceptualization; Resources; Formal analysis; Supervision; Funding acquisition; Validation; Investigation; Visualization; Methodology; Writing—original draft; Writing—review and editing.

Source data underlying figure panels in this paper may have individual authorship assigned. Where available, figure panel/source data authorship is listed in the following database record: biostudies:S-SCDT-10_1038-S44318-024-00258-3.

## Disclosure and competing interests statement

The authors declare no competing interests.

# Expanded View Figures

**Figure EV1. Following nuclear decay of pre-mRNAs in export-block conditions the cytoplasmic deadenylation and decay of mRNAs is revealed.**

(A) Time-dependent change in mRNA abundance relative to control samples in Mex67-depleted cells shown in the form of a series of boxplots. The boxplot central line is the median. The box edges are the 25th and 75th percentiles. Whiskers extend to 1.5 times the IQR (Inter Quartile Range). All three replicates were combined into time point ranges. For each time-slot the number of transcripts assessed in each replicate is given (3875, 3989 and 3032). Sum of 10896 mRNAs is assessed in each time-slot. Note that those are largely overlapping). (B) Time-dependent abundance of selected mRNAs in DRS datasets from Mex67-depleted cells. Fitted lines demonstrate the decay factor calculated for each mRNA based on DRS data. (C) Scatterplot comparing half-life times in Mex67-depleted DRS datasets to those reported by Chan et al (2018). (D) Scatterplot comparing half-life times reported by Miller et al (2011) to those by Chan et al (2018). (E) Global distributions of pA-tail lengths of total mRNA in three replicates of Mex67-AA chase experiments used to model deadenylation. The number of transcripts (pA-tail estimates) in each density plot is given on the panel. (F) Scatterplot comparing single mRNA log2 abundance to half-life. Abundant mRNAs and RPG mRNAs are highlighted with gold and blue dots. (G) Scatterplot comparing mean pA-tail length to mRNA half-life. (H) Global distribution of pA-tail lengths of highly abundant mRNAs isolated from control samples and at various times of Mex67 depletion, presented as a density plot (left panel) and violin plot (right panel). Replicates were merged. The number of transcripts (pA-tail estimates) in each density plot is given on the panel. (I) Global distribution of pA-tail length of low-abundance mRNAs. Replicates were merged. (J) Global pA-tail distribution of RPG and non-RPG mRNAs. Replicates were merged. The number of transcripts (pA-tail estimates) in each density plot is given on the panel. (K) Global pA-tail distribution of individual *RPL36A*, *GAS1*, *RPL4A*, and *HHF1* mRNAs. Stars mark long pA-tailed mRNAs accumulating in later Mex67-depletion time points, indicative of hyperadenylation (Jensen et al, 2001). The number of transcripts (pA-tail estimates) in each density plot is given on the panel.

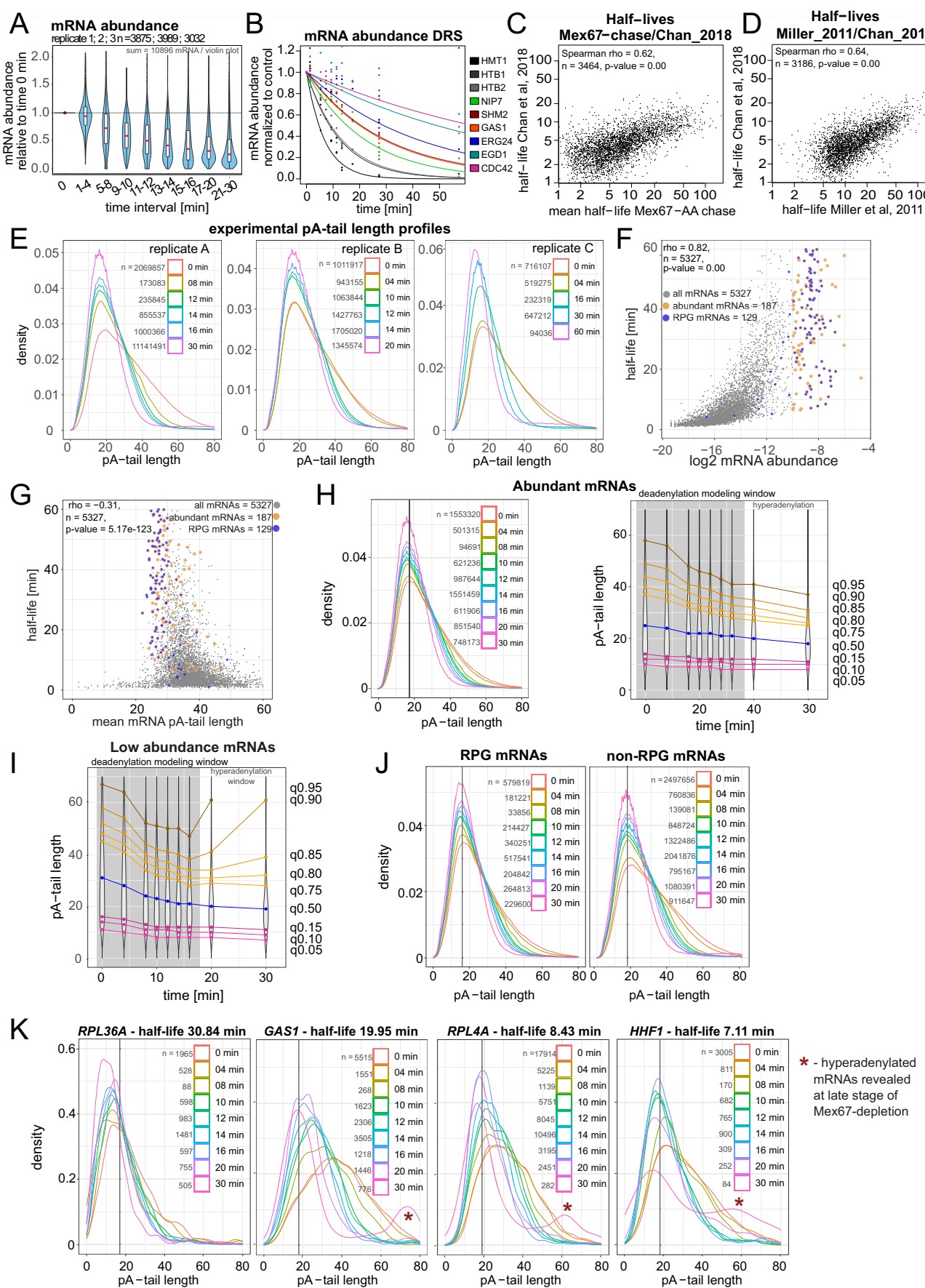

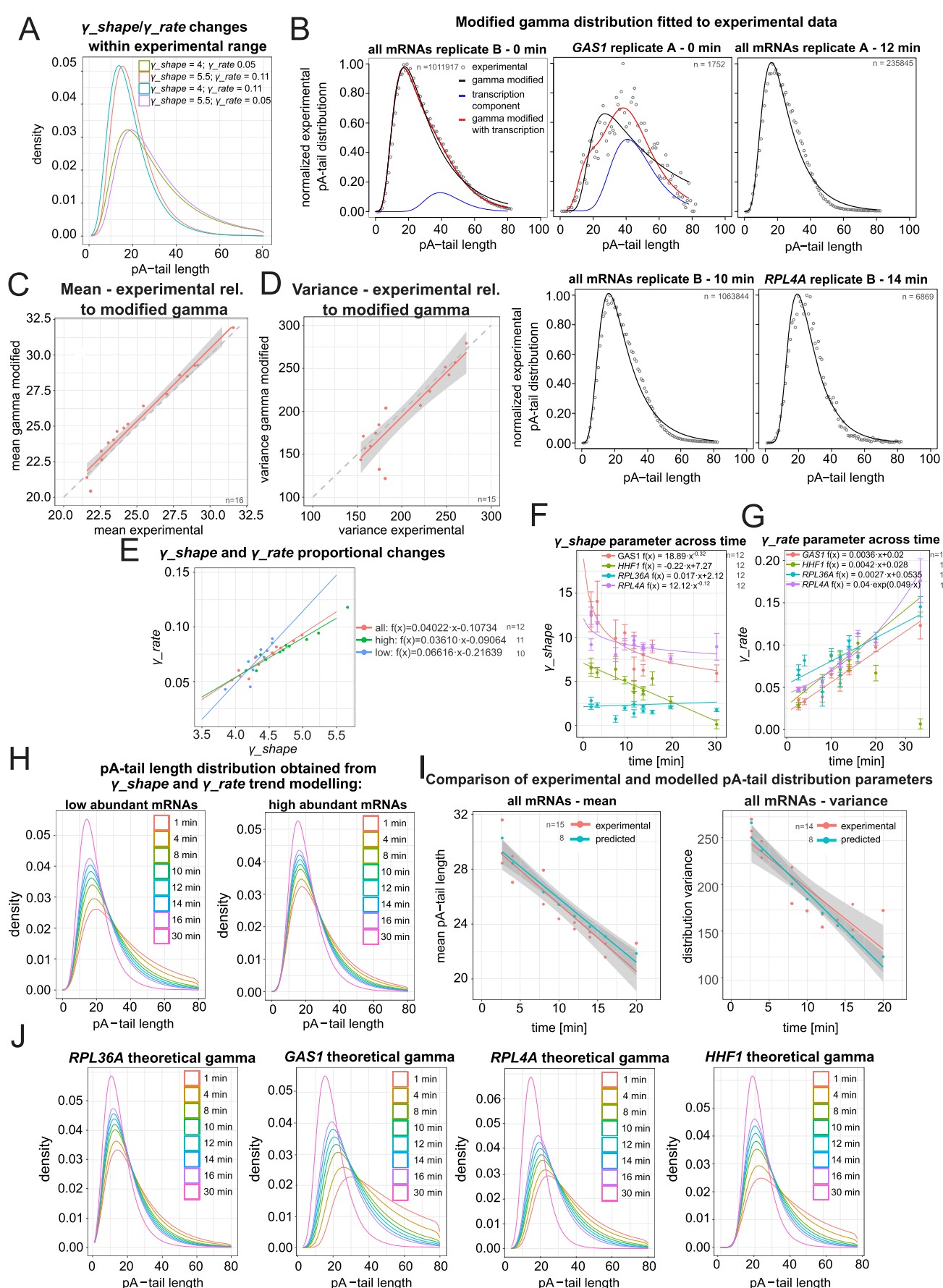

◀ **Figure EV2.   A modified gamma distribution accurately describes experimental yeast pA-tail distributions.**

(A) Plot displaying modified gamma distributions for $\gamma\_shape$ and $\gamma\_rate$ parameters within experimental values. The green line represents a function that effectively describes the pA-tail distribution of the entire transcriptome in control cells ($\gamma\_shape = 4$; $\gamma\_rate = 0.05$). (B) Fitting of the modified gamma distribution to experimental datasets. Figure complements examples shown in Fig. 4C. (C) Comparison of the experimental whole-transcriptome mean pA-tail length to the mean pA-tail length of the fitted modified gamma model. A regression line with a 95% confidence interval was fitted to the data points for comparison with the diagonal. The modified gamma distribution slightly overestimates the mean due to discrepancies in estimating very long pA-tails, as seen in Fig. EV2B for pA-tails of 50 and greater. (D) Graph comparing the experimental whole-transcriptome pA-tail length variance length to the variance of the fitted modified gamma model. A regression line with a 95% confidence interval was fitted. (E) Graph comparing changes in the $\gamma\_shape$ and $\gamma\_rate$ parameters of the modified gamma distribution fitted to Mex67-depletion chase time points for all mRNAs and those of high and low abundance. The graph complements plots shown in Fig. 4D, E. (F, G) Graphs showing time-dependent changes in the value of $\gamma\_shape$ (F) and $\gamma\_rate$ (G) parameters of the modified gamma probability distribution fitted into experimental data for selected mRNAs: *GAS1*, *HHF1*, *RPL36A*, and *RPL4A*. The parameters are given as full-colored dots supplemented with vertical standard error bars. Each estimate was derived from distributions shown in Fig. EV1K (refer to the panel for the number of reads). Equations and continuous lines describe functions fitted to modified gamma parameters to predict the evolution of theoretical distributions (Fig. EV2J). (H) Density plots showing time-dependent evolution of modeled pA-tail length distributions for mRNAs of low and high abundance. The evolution of the median of those distributions is shown in Fig. 4G. (I) Graphs comparing the time-dependent evolution of the modeled modified gamma distribution mean and variance to the experimental one for the whole coding transcriptome. Lines were fitted to both datasets and are displayed with a 95% confidence interval. (J) Density plots showing predicted *GAS1*, *HHF1*, *RPL36A*, and *RPL4A* pA-tail length distributions obtained by modeling changes in modified gamma parameters ($\gamma\_shape$ and $\gamma\_rate$), as shown in Fig. EV3F,G. These should be compared to experimental distributions shown in Fig. EV1K.

                                 

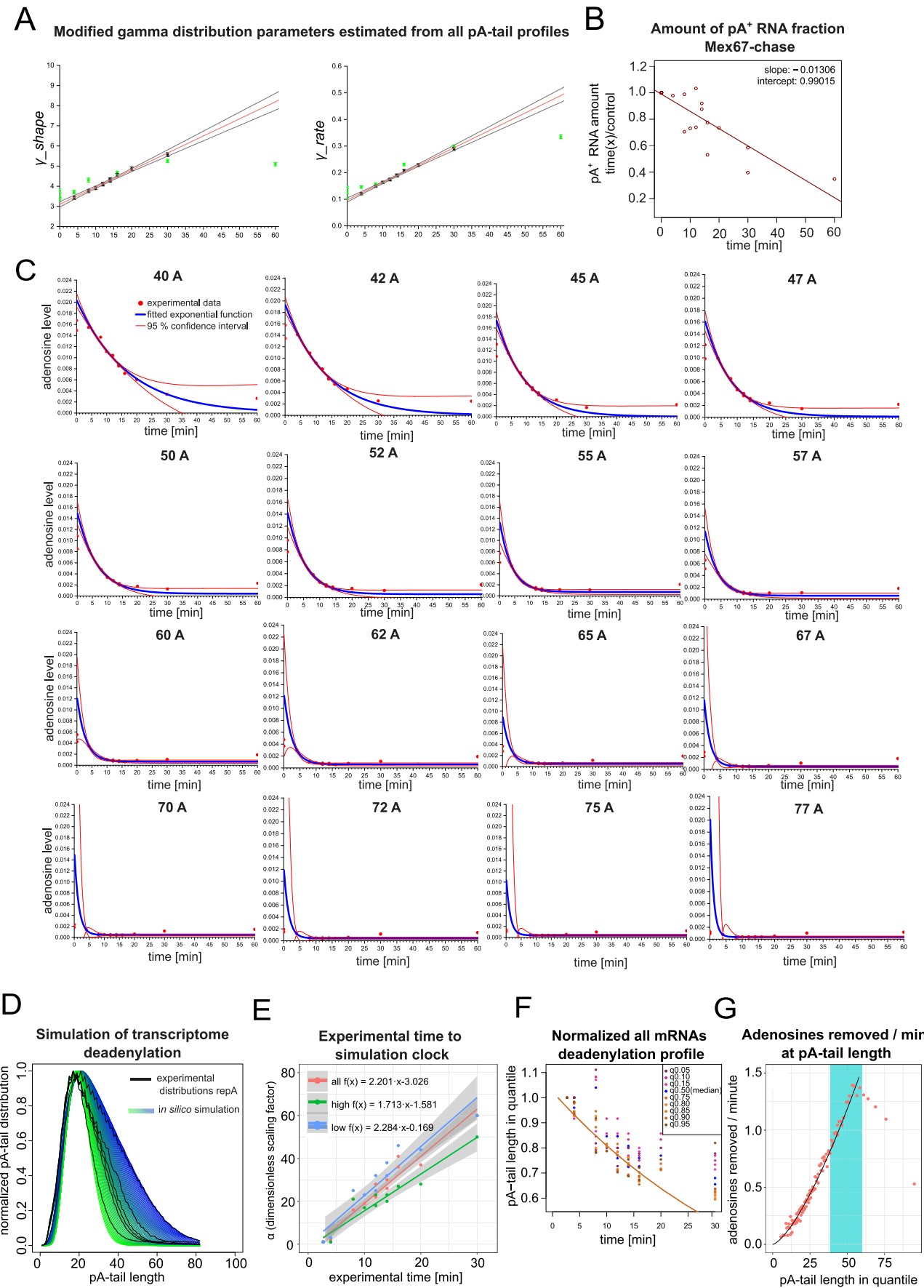

◀ **Figure EV3.** **The deadenylation process can be reconstituted in silico using the modified gamma distribution.**

(A) Graphs illustrating the time-dependent evolution of the $\gamma\_shape$ and $\gamma\_rate$ parameters of the 3-dimensional modified gamma distributions presented in Fig. 4H. (B) Graph depicting time-dependent change in pA$^+$ RNA fraction recovery from total using beads coated with oligo-dT$_{(25)}$. A linear function was fitted to predict changes in whole-transcriptome pA$^+$ RNA levels and was used to normalize pA-tail levels for adenosine half-life calculations shown in Fig. EV3C. (C) Series of graphs showing absolute change in levels of adenosines at specific pA-tail positions for the whole transcriptome. Half-lives of adenosines at each position were calculated and plotted in Fig. 4I to derive a transcriptomic apparent deadenylation rate. (D) Graph comparing experimental pA-tail density plots for the coding transcriptome in Mex67-depletion chase replicate A (black lines) to distributions resulting from consecutive deadenylation simulation cycles (series of blue-green lines). The constant dictating each deadenylation simulation interval was defined by the dimensionless parameter $a$ (see "Methods"). This was carried out to ensure marked differences between consecutive distributions. All distributions were normalized to the distribution peak (distribution maximum). The in silico distribution that best overlapped with the experimental distributions was selected, assigning a specific value of $a$. (E) Graph comparing experimental deadenylation times [min] with the ordinal number of the dimensionless $a$ parameter (dimensionless scaling factor) describing the number of in silico deadenylation steps. (F) Graph displaying the time-dependent decrease in pA-tail length values in each quantile normalized to control. The dots designating the upper quantiles (orange dots) cluster around a steeper slope than those designating lower quantiles (blue and maroon dots). (G) Graph showing the number of adenosines removed per minute as a function of pA-tail length. This graph complements the one shown in Fig. 5B.

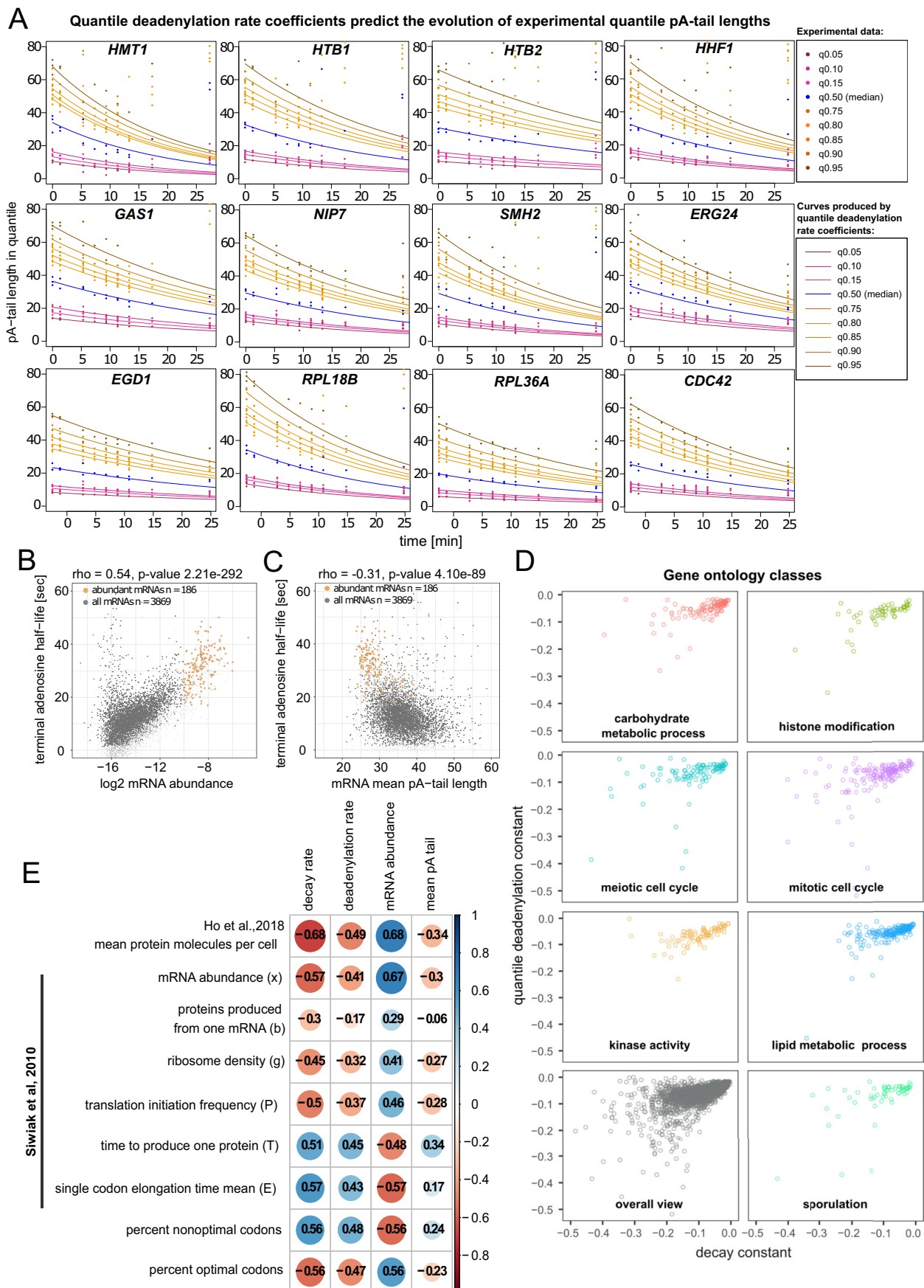

◀   **Figure EV4.  A simplified method based on changes to distribution quantile values can measure deadenylation rates.**

(**A**) Series of graphs depicting as color-coded dots the changes in pA-tail length in selected quantiles (upper—75–80–85–90–95th, median—50th, and lower—15–10–5th) over time for selected single mRNA examples. The continuous lines represent the quantile deadenylation coefficients calculated from the upper quantiles. (**B**, **C**) Scatterplots comparing terminal adenosine half-life to log2 mRNA abundance (**B**) or mean mRNA pA-tail length (**C**). Spearman's rho correlations and p-values were calculated separately for each set using the rstatix package in R. (**D**) Comparison of decay to deadenylation rates for various gene ontology groups, which are also shown in Fig. 5H. (**E**) Correlation matrix comparing decay, deadenylation rates, mRNA abundance, and mean pA-tail length derived from the Mex67-depletion time course to estimates of protein abundance from Ho et al (2018), various translation rate parameters (Siwiak and Zielenkiewicz (2010)), and percent optimal or non-optimal codons.

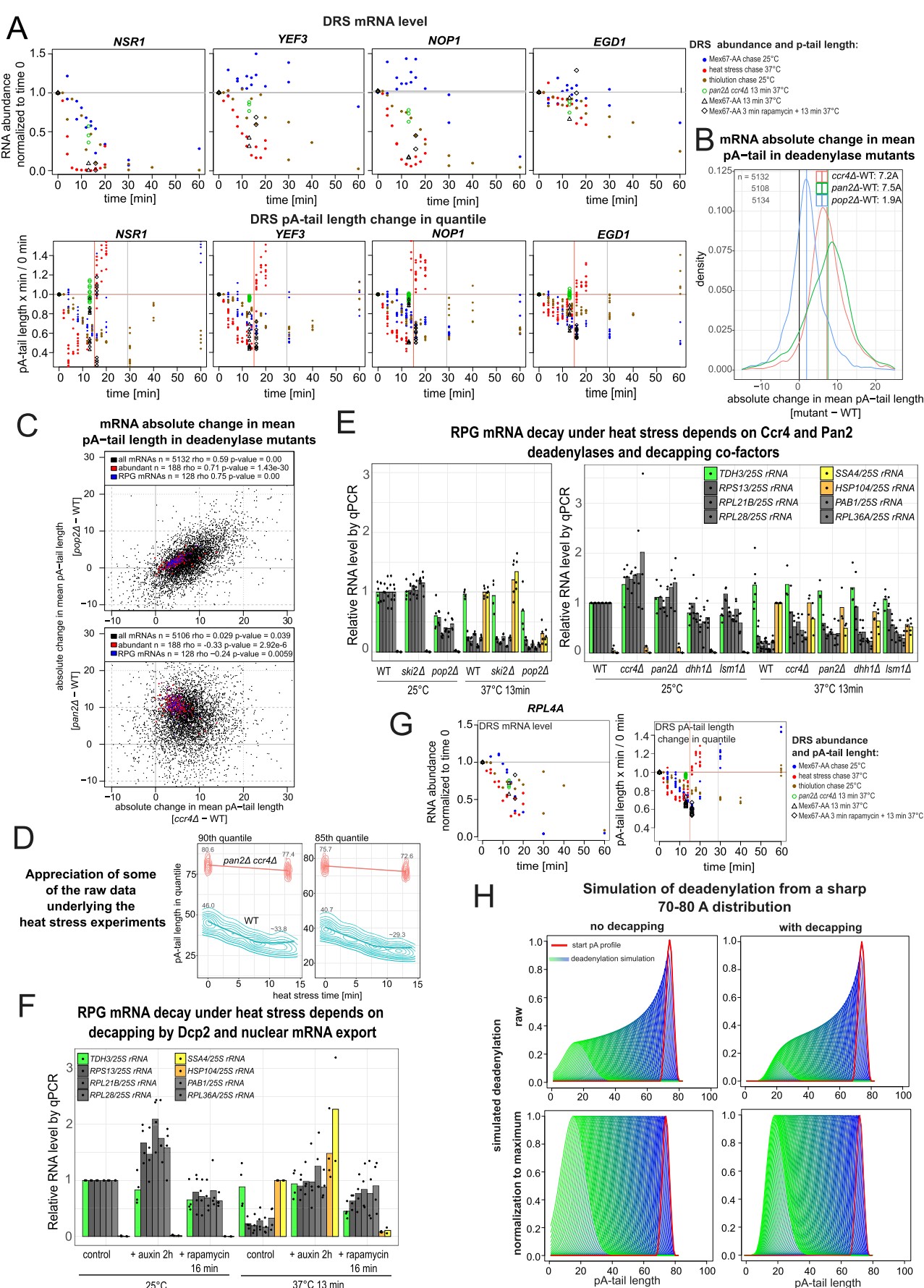

◀  **Figure EV5.   RPG and non-RPG mRNAs have altered deadenylation and decay rates during response to heat stress.**

(A) Series of graphs showing, in rows, the time-dependent and normalized to control changes in: (top) mRNA abundance by DRS, and (bottom) pA-tail length in upper quantiles (75–95th) for single RPG mRNAs: *NSR1, YEF 3, NOP3,* and *EGD1* in Mex67 depletion, heat stress 37 °C, and thiolutin 25 °C chase sequencing data. (B) Mean pA-tail lengths of individual mRNAs in wild-type cells were subtracted from the mean lengths observed in deadenylase mutants (*ccr4Δ* or *pan2Δ*, or *pop2Δ*) and displayed in the form of a density plot. The mean change in each strain is marked as a vertical line and specified in the figure legend. (C) Scatterplots showing the absolute change in mean pA-tail length in *ccr4Δ* - WT on the x-axis compared to *pop2Δ* – WT (top graph) or *pan2Δ* – WT (bottom graph) on the y-axis. Transcripts of high abundance or RPG mRNAs are highlighted in blue and red, respectively. Spearman's rho coefficients and the number of mRNAs compared (n) are listed in each panel legend. (D) Plots displaying the raw pA-tail length density distribution of two example upper quantiles in the wild-type strain heat stress chase compared to the control and 13 min heat shock of a double *pan2Δ ccr4Δ* mutant strain. The plots present raw data also shown in Fig. 6E. The fitted lines are for orientation purposes only and connect the local maxima either with local or linear regression. (E) Abundance of selected RPG mRNAs normalized to *25S rRNA* in control or *ski2Δ, pop2Δ, ccr4Δ, pan2Δ, lsm1Δ,* and *dhh1Δ* cells at 25 °C compared to 13 min heat shock at 37 °C determined using reverse transcription coupled to qPCR. Single dots show biological replicate values used to calculate the mean. (F) Abundance of selected RPG mRNAs normalized to *25S rRNA* in a double Dcp2-AID and Mex67-AA strain under steady-state and after 13 min heat stress determined using reverse transcription coupled to qPCR. Prior to heat stress the strains were either treated with auxin for 2 h to deplete Dcp2 or with rapamycin for 3 min to deplete Mex67 prior (25 °C). Heat stress was conducted at 37 °C for 13 min. Single dots represent biological replicates. (G) Graphs showing for *RPL4A* (top) mRNA abundance by DRS, and (bottom) change in pA-tail length in upper quantiles (75–95th) normalized to control for Mex67 depletion, heat stress 37 °C, and thiolutin 25 °C chase sequencing data, along with point heat stress of *ccr4Δ pan2Δ* double mutant and cells Mex67-depleted 3 min prior to heat stress. (H) A theoretical pA-tail distribution of median 75As and low variance (red line) subjected to deadenylation simulation (series of blue and green lines). Artificial deadenylation was performed with and without inducing decapping, displayed with or without normalization to each distribution maximum value. Normalization to the maximum is shown to better display the systematic change in distribution variance and location of the maximum.

