## [Peer Review File · The EMBO Journal]

Modeling of mRNA deadenylation rates reveal a complex relationship between mRNA deadenylation and decay

Agnieszka Czarnocka-Cieciura, Jarosław Poznański, Matti Turtola, Rafał Tomecki, Paweł Krawczyk, Seweryn Mroczek, Wiktoria Orzeł, Upasana Saha, Torben Jensen, Andrzej Dziembowski, and Agnieszka Tudek

Corresponding author(s): Agnieszka Tudek (atudek@ibb.waw.pl) , Andrzej Dziembowski (adziembowski@iimcb.gov.pl)

Review Timeline:

Submission Date:	28th Feb 24
Editorial Decision:	15th Apr 24
Revision Received:	12th Jul 24
Editorial Decision:	15th Aug 24
Revision Received:	11th Sep 24
Accepted:	19th Sep 24

Editor: Cornelius Schneider

Transaction Report:

Dear Dr. Tudek,

Thank you for submitting your manuscript for consideration by the EMBO Journal. We have now received comments from three reviewers, which are included below for your information.

As can be seen from the reports, all three referees found the results of importance and interest, and agreed that the experiments were performed competently. However, there are also several major concerns which would need to be addressed before publication at The EMBO Journal. We agree with referees #1 that the readability for non-expert readers could be improved by streamlining the manuscript. More importantly we also think that the additional experiments requested by referees #2 and #3 would substantially strengthen the manuscript as the inclusion of these additional factors (Caf1 and e.g. Puf3) would not only strengthen the main message of the manuscript but also improve the model. I should also add that it is The EMBO Journal policy to allow only a single major round of revision and that it is therefore important to resolve the main concerns at this stage.

I am happy to answer any additional questions or discuss possible revisions via e-mail or videoconferencing.

We generally allow three months as standard revision time, which can be extended to 6 months in case of major revisions, such as the experiments required here. As a matter of policy, competing manuscripts published during this period will not negatively impact on our assessment of the conceptual advance presented by your study. However, we request that you contact the editor as soon as possible upon publication of any related work, to discuss how to proceed. Should you foresee a problem in meeting the deadline, please let us know in advance and we may be able to grant an extension.

Thank you for the opportunity to consider your work for publication. I look forward to your revision.

Yours sincerely,

Cornelius Schneider

Cornelius Schneider, PhD
Editor
The EMBO Journal
c.schneider@embojournal.org

- a point-by-point response to the referees' comments, with a detailed description of the changes made (as a word file).
- a word file of the manuscript text.
- individual production quality figure files (one file per figure)
- a complete author checklist, which you can download from our author guidelines

(<https://www.embopress.org/page/journal/14602075/authorguide>).
- Expanded View files (replacing Supplementary Information)
Please see out instructions to authors
<https://www.embopress.org/page/journal/14602075/authorguide#expandedview>

We realize that it is difficult to revise to a specific deadline. In the interest of protecting the conceptual advance provided by the work, we recommend a revision within 3 months (14th Jul 2024). Please discuss the revision progress ahead of this time with the editor if you require more time to complete the revisions. Use the link below to submit your revision:

Referee #1:

In this manuscript, the authors develop a mathematical model to measure deadenylation rates based on nanopore direct RNA sequencing. The authors use as a base for their model data obtained after inhibition of nuclear mRNA export. The authors fit their model to the generated data to derived deadenylation rates and other parameters. Next, they combined the model with stress conditions and mutants for the mRNA degradation machinery to explore the relationship between mRNA stability and polyA length.

The relationship between mRNA stability and decay is a complex and interesting topic.

My main concern is that the readability of the paper decreases significantly from the description of the modeling strategy. And the follow up section where the derived model is combined with environmental and genetic perturbations (pages 8-9) is too compact and difficult to follow. To improve the readability and facilitate the evaluation of the authors work I have the following recommendations.

1. Add a conceptual interpretation of the parameters of the model. The authors describe it a bit better in the methods section, but not in the main text. I think they should devote the first lines of the section to explain what is X, what is the rate,... and state more clearly the assumptions. Then one can move some of the more technical aspects to the methods section where more detail can be provided.
2. I would recommend adding a pictogram depicting the biological phenomena that the model is trying to capture and how the parameters can be placed in that model. That would be a great panel for a main figure.
3. On many occasions the authors explain that they fit to "the data". However, it is not obvious which data they use to fit unless the reader goes deep in the supplementary figures.
4. It is not clear to me the differences between the normal and the quantile model. Are quantiles of polyA lengths? Are the 2 methodologies independent, or is a simplification where instead of each nt of the polyA the authors use quantiles of the lengths? Again, a pictogram would facilitate readability.
5. As the main deliverable from the paper is the models generated, the authors should provide a script (or at least a toy model) to be able to rerun the model. Ideally using as an input some of the tables the authors deposited in Mendeley.
6. I found very difficult to understand the logic of the last section. I believe the interpretation from the authors is correct, however it should be expanded. For example, it is not obvious for a reader how to one should interpret genetic perturbations at single time points (Fig 6D-E). It is also difficult to put in context the experiments described in 6F-G and S7B-C. As those are key for the biological conclusions, they should be better explained.

Non-essential suggestion:

I feel that this manuscript has an excess of figures. This makes difficult to identify the key supplementary figures bringing conceptual advancement from the more descriptive ones.

Referee #2:

General summary

Here, the authors have established that the anchor-away system applied to depletion of nuclear Mex67 is an effective tool to study cytoplasmic mRNA deadenylation and degradation that correlated well with previous metabolic labelling studies. The study reports quantitative modelling of cytoplasmic mRNA deadenylation and degradation in yeast. Two numerical models for mRNA deadenylation are developed that fit the experimental data very closely; one is precise but limited to transcripts with high read scores, while the other is more broadly applicable but less accurate. A transcriptome-wide deadenylation rate is determined to be ~ 10 As/min, with variation between individual transcripts that correlates with mRNA abundance, codon optimality and response to stress. As reported previously (Tudek et al., 2021), the data support the conclusion that high abundance mRNAs have short poly(A) tails and are deadenylated slowly, while low abundance transcripts have long poly(A) tails and are deadenylated more rapidly. The approach shows mRNAs to be sensitive to decapping when the poly(A) tail has been shortened to ~ 20 nucleotides. Pab1 depletion led to an increase in poly(A) tail length, with a greater effect on low abundance mRNAs. Deadenylation rates were shown to be constant for a given transcript, correlated inversely with mRNA abundance and were low for transcripts from house-keeping genes, and that these correlated with codon optimality. Deadenylation was shown to be accelerated upon heat stress and sensitive to depletion of Pan2 and Ccr4 in a synergistic manner.

The experimental data is of a very high technical quality. The development of accurate numerical models for mRNA deadenylation provide a basis for the future quantitative analyses of more detailed mechanistic aspects of mRNA degradation on a transcriptome-wide scale in yeast and other systems.

main points

The only main point that I would raise is that it is not clear to me why there is no data addressing the role of Caf1 in deadenylation. There are several reports supporting a contribution of Caf1 in mRNA deadenylation in yeast and mammalian cells (e.g. Tucker et al., 2001; Webster et al., 2018; Mostafa et al., 2020; Yi et al., 2018).

minor points

There is no Methods section in the submitted main text of the manuscript. The Methods section in the Supplemental data is overly long for the main text; a concise version should be included in the main text. This should include sufficient information for the reader to follow the mathematical modelling in particular.

Supplemental Tables 1 and 2 are not in the Supplemental data.

Is the increase in poly(A) tail length observed at the 30 minute time-point for the q0.95 fraction (Fig. 2A) due to hyperadenylation, as noted in Fig. S1K? This should be noted when describing this data, which comes before S1K is referenced.

Error bars should be included for data across the three triplicates for each time-point shown in Fig 2E

Similarly, error bars should be shown for the average (?) poly(A) tail length indicated for control Dcp2-AID and Xrn1-AID by the vertical lines in Fig. 3B. Perhaps the poly(A) tail length before a rapid decrease in density is observed would be more appropriate for comparison than the average poly(A) tail length.

The authors should comment on why the poly(A) tail length of individual transcripts shows a sensitivity to Xrn1 digestion but there is no effect at a global level (Fig. S2G)? Is this because the transcripts (other than RPS13?) are of low abundance?

The authors should describe how the estimates of transcription (plotted as blue lines in Fig. 4B) were determined - I couldn't see this described in the Methods section.

The section of text on p6 describing how the microscopic deadenylation rate is determined from the transcriptomic deadenylation rate could be clarified in simpler terms.

At the end of the manuscript there is some data addressing mRNA turnover in strains mutant for decapping factors. It's not clear why mutants of the SKI complex have not been addressed.

At the end of the Discussion section, it is stressed that the analyses reveal mechanisms that allow decapping in the absence of deadenylation under stress conditions. Does this relate to the degradation observed in the *ccr4 Δ pan2 Δ* mutant when there is little deadenylation? If that is the case, the degradation observed has not been shown to be dependent upon decapping.

Referee #3:

This manuscript from Czarnocka-Cieciura and colleagues explores the quantitative landscape of deadenylation and decay in budding yeast. With the advent of third-generation direct RNA long-read sequencing technology, the ability to measure poly(A) tail lengths (easily) has opened the world of deadenylation dynamics, making this manuscript timely and innovative. Another key innovation is the use of a Mex67-anchor allele, which allows extremely rapid shut-off of nuclear export and fast degradation of nuclear transcripts - together this tool enables accurate measurements of cytoplasmic RNA fates. There are many strengths to

this paper. First, the authors generate quantitative models for global and gene-specific deadenylation rates. Second, the results highlight the perils in using steady-state poly(A) tail lengths due to complex relationships between deadenylation, decapping, and export processes - this manuscript, as well as other recent papers, reconciles the surprising conundrum that more unstable transcripts have longer (not shorter) poly(A)-tail lengths at steady state. Third, the data provide strong evidence for the protective role of Pab1p in blocking decapping. Although I cannot support publication in its current form due to a few major weaknesses, once these have been dealt with, this manuscript will be an important contribution to the field.

Major comments:

1. It would be useful to discuss more quality-control metrics for the original datasets. E.g., how similar were the half-lives and tail length (dynamics) between replicates? How were replicates combined? What was the minimum read number for each gene? How many genes? Adding some of this information up-front will help orient the reader for the more sophisticated analyses.
2. The authors convincingly highlight the perils of relying purely on steady-state poly(A) tail lengths to inform an understanding of dynamics, and yet this is their strategy for a variety of experiments, including Xrn1, Dcp1, and Pab1 depletion. In the case of Pab1, early reports from Caponigro and Parker suggest that decapping occurs on long-tailed RNA species. Thus, there are alternative explanations that the authors do not address to explain the poly(A)-tail length changes upon Pab1 depletion. In particular, looking at deadenylation rates in this line would substantially strengthen this section and this paper.
3. It is unclear of the extent to which the authors directly looked at how much decapping rates vary between different mRNAs. Critically, in the mammalian paper from Eisen et al., one of the main conclusions is that more rapid deadenylation coincides with more rapid decapping. There is no discussion of this idea in the text - and it is unclear whether this possibility was explored in the modeling. If not, that is critical point. If the authors obtained similar decapping rates, then that is an important result.
4. Finally, this paper would be substantially strengthened by repeating the deadenylation dynamics experiments and modeling in a line lacking a trans-factor like Puf3. This experiment would provide quantitative information on how such a trans-factor (that has been studied in vitro) affects deadenylation and/or decapping rates.

Minor comments:

1. In displaying the modeling and other results, sometimes it is not clear from the text or figure whether all mRNAs are being modeled (and, if so, is this a weighted average of them or is just based on raw reads?), etc. More clear labeling within the figures and text would help with clarity.

REBUTTAL LETTER

mRNA deadenylation modeling at permissive and stress conditions reveals complex relations to decay

Agnieszka Czarnocka-Cieciura^{1,#}, Jarosław Poznański^{3,#}, Matti Turtola², Rafał Tomecki^{3,4}, Paweł S. Krawczyk¹, Seweryn Mroczek¹, Wiktoria Orzeł¹, Upasana Saha⁵, Torben Heick Jensen⁵, Andrzej Dziembowski^{1,4*}, Agnieszka Tudek^{3,*}

¹ International Institute of Molecular and Cell Biology, Księcia Trojdena 4, 02-109 Warsaw, Poland

² Department of Life Technologies, University of Turku, Biocity, Tykistökatu 6, 205240 Turku, Finland

³ Institute of Biochemistry and Biophysics, Polish Academy of Sciences, Adolfa Pawińskiego 5A, 02-106 Warsaw, Poland

⁴ University of Warsaw, Faculty of Biology, Miecznikowa 1, 02-089 Warsaw, Poland

⁵ Aarhus University, Department of Molecular Biology and Genetics – Universitetsbyen 81, 8000 Aarhus, Denmark

those authors contributed equally

* correspondence should be addressed to: atudek@ibb.waw.pl or adziembowski@iimcb.gov.pl

TABLE OF CONTENT

Response to Editors Comment – page 2

List of modifications of the text and figures that were required to conform to EMBO Journal formatting – page 4

Response to Reviewer 1 – page 5

Response to Reviewer 2 – page 10

Response to Reviewer 3 – page 19

Data availability statement – page 28

Response to Editor's Comment

As can be seen from the reports, all three referees found the results of importance and interest, and agreed that the experiments were performed competently. However, there are also several major concerns which would need to be addressed before publication at The EMBO Journal. We agree with referees #1 that the readability for non-expert readers could be improved by streamlining the manuscript. More importantly we also think that the additional experiments requested by referees #2 and #3 would substantially strengthen the manuscript as the inclusion of these additional factors (Caf1 and e.g. Puf3) would not only strengthen the main message of the manuscript but also improve the model. I should also add that it is The EMBO Journal policy to allow only a single major round of revision and that it is therefore important to resolve the main concerns at this stage.

ANSWER: *We acknowledge comments and questions from the Editor and Reviewers. In order to address all the constructive criticism, we have performed additional experiments, extensively rewritten the Result and Discussion sections (including with the advice of external language services provided by native speakers), and improved the data and model presentation. Additionally, the manuscript was tailored to fit the EMBO Journal formatting requirements. Thus, we hope that after this major revision, you will find our manuscript substantially improved and suitable for publication.*

All Reviewers raised multiple important questions regarding the presentation of our new deadenylation modeling methodologies. Therefore, we inserted three novel schematic panels that: (1 - 4B) explain the conceptual meaning of the pA-tail length modified gamma distribution parameters, (2 - 4K) sum up the modified gamma modeling strategy and (3 - 5C) compare the two methods we developed for calculation of the deadenylation rate. Additionally, we revised panel 5A and associated EV4A to better distinguish between experimental data and modeling results. The main manuscript text was also extensively rewritten to better convey our message, with additional numerical proof being provided. Importantly, the original R code used to estimate the modified gamma parameters, along with its main deliverables, is now deposited at Mendeley and presented in the Appendix together with guidelines for future users regarding its possible adaptation to other datasets (novel Appendix section 1.8). Importantly, in response to Reviewers 2 and 3 we now provide a large Appendix section 1.10 supported by novel Appendix Figures 3 and 4 and Appendix Table 6 that discusses our data quality control and decay and deadenylation modeling strategy.

On many occasions, the Reviewers pointed out that the presentation of experimental data was convoluted, which depreciated the validity of the conclusions stated in the main text. To remedy this, we inserted two models in (5G and 6J), revised existing experimental panels, and associated text. Specifically, in response to Reviewer 3, we now clearly state in the main text that individual mRNAs display varied deadenylation and decay rates and that rapid deadenylation is indeed associated with short mRNA half-lives. In contrast, slow deadenylation confers transcript stability (revised panel 5F). Reviewer 3 also asked for additional proof of the Pab1 role in deadenylation. Though, for technical reasons specified in the response, we were unable to use the strain for deadenylation-decay modeling, we did re-analyze the existing data (novel Fig. 3C and Appendix Fig. 1L). This showed that short-lived mRNAs especially accumulate a long tailed mRNA fraction over time. This RNA population's pA-tail lengths are equal to previous estimates of new mRNA adenylation (60-90A), which, in conjunction with the lack of cytoplasmic adenylases in budding yeast, supports our conclusion that Pab1 is required for deadenylation in vivo. We also note that the related outstanding question regarding the

possibility of deadenylation-independent decapping has been addressed in the last section of the manuscript using an orthogonal method (Fig. 6 and revised EV5). This section, for multiple other concerns raised by all Reviewers (notably 2), was extensively revised and supplemented with novel data.

Following all Reviewer's (particularly # 2 and 3) requests, we now provide novel data that dissect the relative in vivo importance of the three cellular deadenylases (Caf1/Pop2, Ccr4 and Pan2). The novel panels 6H and EV5B-D show that the two main cellular deadenylases in yeast are Ccr4 and Pan2, while Pop2's contribution is minor, at best. This underscores the divergence from humans where the role of hPop2(Caf1) is dominant. Importantly, this data shows that, though individual deletion of CCR4 or PAN2 genes leads to similar mean elongation of pA-tail lengths by around 7 adenosines, only removal of both genes leads to the creation of a pA-tail profile similar to that of newly made non-deadenylated mRNAs (novel Fig. 6H). This clearly supports the notion that both deadenylases act redundantly, and in their absence, deadenylation is virtually inhibited. Such new data better supports data ulteriorly shown on panels 6D-E, which indicates that despite very low deadenylation levels, decay of RPG mRNAs still proceeds. To substantiate this, some of the raw data underlying panel 6E is now shown in the revised EV5D.

Finally, Reviewer # 3 asked about the effects of deadenylase cofactors on deadenylation kinetics, particularly, Puf3. We followed those suggestions, but the initial dataset generated by us indicated that the effects of PUF3 gene deletion on its mRNA targets are pleiotropic and complicated. In vivo, PUF3 deletion did not lead to the elongation of polyA tails of its targets, which was expected for deadenylase cofactor, but rather to shortening. Therefore, we think that analysis of the role of Puf3 as well as other deadenylase cofactors requires more comprehensive and dedicated studies.

In sum, the manuscript has been substantially improved. We thus hope that it will be suitable for publication and would like to extend our gratitude for the insightful remarks from the Reviewers and Editor.

List of modifications of the text and figures that were required to conform to EMBO Journal formatting

A. We modified the article title so that the character count would be within a maximum range of 100.

Original title (1150 characters): Deadenylation modeling reveals complex relations between mRNA deadenylation and decay affected by growth conditions

Revised title (98 characters): mRNA deadenylation modeling at permissive and stress conditions reveals complex relations to decay

B. We modified the references to appendix material from Supplemental Figure (or Figure S[number]) to Expanded View Figure (EV[number]).

C. We created a Methods section in the main text, which details the techniques most relevant to the study as well as details of the modeling procedure. This section is supported with a extensive Appendix which lists non-essential methods and also disusses key quality control questions referencing to novel Appendix Figures 3 and 4 and novel Appendix Tables 5 and 6 along with revised Dataset Table 2.

D. We formatted the References and modified the style of citations in the main text from [Author et al., YEAR] to [Author et al, YEAR].

E. To comply with the requirement that a maximum of 5 Expanded View Figures can be shown we moved former Supplemental Figure 2 and 6 to the Appendix. Hereafter the Figures are referred to as Appendix Figure 1 and 2 respectively.

Response to Reviewer 1

In this manuscript, the authors develop a mathematical model to measure deadenylation rates based on nanopore direct RNA sequencing. The authors use as a base for their model data obtained after inhibition of nuclear mRNA export. The authors fit their model to the generated data to derive deadenylation rates and other parameters. Next, they combined the model with stress conditions and mutants for the mRNA degradation machinery to explore the relationship between mRNA stability and polyA length.

The relationship between mRNA stability and decay is a complex and interesting topic.

ANSWER: *We thank the Reviewer for the positive opinion about the relevance of our research.*

My main concern is that the readability of the paper decreases significantly from the description of the modeling strategy. And the follow up section where the derived model is combined with environmental and genetic perturbations (pages 8-9) is too compact and difficult to follow. To improve the readability and facilitate the evaluation of the authors work I have the following recommendations.

1. Add a conceptual interpretation of the parameters of the model. The authors describe it a bit better in the methods section, but not in the main text. I think they should devote the first lines of the section to explain what is X, what is the rate,... and state more clearly the assumptions. Then one can move some of the more technical aspects to the methods section where more detail can be provided.
2. I would recommend adding a pictogram depicting the biological phenomena that the model is trying to capture and how the parameters can be placed in that model. That would be a great panel for a main figure.

ANSWERS to questions 1 and 2: *We acknowledge that the main text lacked an adequate introduction to the properties of the gamma distribution and its parameters. We have revised the entire chapter describing the numerical model to rectify this oversight. Additionally, we have incorporated a diagram in Figure 4B (see below) to illustrate the conceptual meaning of the γ_{shape} and γ_{rate} parameters, which is supported by Figure EV5A. In brief, the γ_{rate} parameter is akin to the exponential function that runs along the right arm of the pA-tail distribution. It is the parameter that most robustly affects the distribution and it thus describes the balance between nuclear de novo mRNA adenylation and cytoplasmic deadenylation. The γ_{shape} does not have a direct meaning but by similarity to the Erlang distribution (a derivative of the classical gamma model that operates on integers) it designates the five critical events leading to full mRNA decay: removal of 3 Pab1 molecules, decapping and exonucleolysis. As such, the γ_{shape} value encapsulates the cytoplasmic mRNA decay pathway. To better explain our approach we sum up our modeling strategy in a novel panel 4K. Furthermore, to facilitate the reuse of our model by third parties we now provide the R code developed to estimate the γ_{rate} and γ_{shape} parameters along with a novel section 1.8 in the Appendix file entitled 'Calculation of the modified gamma parameters' that provides more detailed instructions.*

Figure EV2A

Novel Figure 4B

Expanded View 2A. Plot shows modified gamma distributions for γ_shape and γ_rate parameters within the experimental values. The green line shows a function that describes well a whole transcriptome pA-tail distribution in control cells ($\gamma_shape = 4$; $\gamma_rate = 0.05$). **Figure 4B.** Scheme showing the conceptual meaning of the modified gamma distribution parameters (see main text). In brief, the gamma_rate is akin to the exponential coefficient of the right arm of the pA-tail distribution, whereas the gamma_shape+1 can be interpreted as the number of critical events leading to complete mRNA decay.

3. On many occasions the authors explain that they fit to "the data". However, it is not obvious which data they use to fit unless the reader goes deep in the supplementary figures.

ANSWER: Indeed, the original manuscript did have some shortcomings in this regard. We have supplemented the manuscript with detailed information about the datasets to address this issue. We trust that these additions will enhance the manuscript's readability and clarity.

4. It is not clear to me the differences between the normal and the quantile model. Are quantiles of polyA lengths? Are the 2 methodologies independent, or is a simplification where instead of each nt of the polyA the authors use quantiles of the lengths? Again, a pictogram would facilitate readability.

ANSWER: We thank the Reviewer for bringing this issue to our attention. The manuscript provides two independent and non-interchangeable methods for the calculation of the deadenylation rate. The methods are intended for datasets differing in number of reads. The necessity for two methods stems from the prominent difference between the read coverage of the whole transcriptome (the sum of all mRNAs indiscriminately of type) and individual mRNA pA-tail length distributions, with the latter being often much more modest. The accurate numerical model is rooted in the modified gamma distribution and allows for precise estimation of the enzymatic deadenylation rate from the apparent transcriptomic one. However, this method heavily relies on estimation of terminal adenosine half-lives, which are well determined only in the whole-transcriptome distribution. For individual transcripts, the distribution of terminal adenosine is erratic due to lower read coverage, precluding the use of the modified gamma model. Therefore, to tackle this issue, we devised the quantile-based method, which does not

provide a precise enzymatic deadenylation rate but rather a means to distinguish how fast the pA-tail distribution changes for individual transcripts. In this method, the deadenylation process is described by changes to the value of intervals (quantiles) that represent the long-tailed extremity of the pA-tail distribution. Importantly, we show that on a transcriptomic scale, the quantile deadenylation rate reflects quite accurately the apparent deadenylation rate of the modified gamma distribution (compare Figures 4I to 5B below). Importantly, division of distributions into quantiles is a widely-used statistical method. A quantile (from 1 to 100) designates how many values in a distribution are below and above, with the 50th quantiles being equal to the distribution median. This was graphically shown early on in Fig. 2A and then simplified in Fig. 5A (off note, we revised the Fig. 5A to better distinguish between experimental data and the resulting modeling). We however acknowledge the shortcomings of our method description and we therefore heavily revised the main text to better explain both methodologies and the key differences. Importantly, those are also summed up the the novel panel 5C (below).

Novel Figure 5C

C Comparison of the modified gamma and quantile methods for estimation of deadenylation rate

Method:	Modified gamma	Quantile
distribution type:	whole transcriptome/ large groups	whole transcriptome/ large groups/ individual mRNAs
accuracy of deadenylation rate estimation:	high	low
type of deadenylation rate estimate:	apparent and enzymatic	apparent for groups/ differentiation between individual mRNAs rates

Figure 4I

Figure 5B

B The apparent adenosine half-life can be estimated from the quantile deadenylation rate

Revised Figure 5A

A The quantile deadenylation rate coefficient predicts the evolution of experimental quantile pA-tail lengths

Revised Figure 2A

A Experimental pA-tail length profile: as density plot

as violin plot

5. As the main deliverable from the paper is the models generated, the authors should provide a script (or at least a toy model) to be able to rerun the model. Ideally using as an input some of the tables the authors deposited in Mendeley.

ANSWER: *Indeed, we did not include the script with the numerical model in the initial version of the manuscript. The code calculates the modified gamma parameters (developed in basic R) is now deposited at Mendeley and provided as well in the novel appendix section 5.*

At present the code is adapted to the Mex67 dataset only. The output files of the code are: (1) a .csv table listing the γ_{shape} and γ_{rate} parameters calculated for the Mex67 datasets and (2) and a graphical file with a representation of the fitting procedure and deadenylation simulations (some of the graphs are displayed on Fig. 4C and EV2B). Both files are now available at Mendeley. The list of γ_{shape} and γ_{rate} parameters is also given as Appendix Table 5.

Additionally, in the Appendix, we now included a novel section 1.8 entitled 'Calculation of the modified gamma parameters' that provides guidelines for potential future users regarding the steps required to adapt the code for other datasets.

6. I found very difficult to understand the logic of the last section. I believe the interpretation from the authors is correct, however it should be expanded. For example, it is not obvious for a reader how to one should interpret genetic perturbations at single time points (Fig 6D-E). It is also difficult to put in context the experiments described in 6F-G and S7B-C. As those are key for the biological conclusions, they should be better explained.

ANSWER: *In response to the Reviewers' suggestions, we have extensively rewritten the final section of the results, amended graphical shortcomings in Fig. 6D-E, and included additional data for clarity (new Fig. 6H, EV5B-E).*

We have determined the precise decay and deadenylation rates based on our chase experiments with at least 16-time points performed under steady-state and stress conditions (heat or thiolutin) (Fig. 6B-C). Further analysis has shown quantitatively that deadenylation and decay can change in response to growth stimuli for individual mRNAs pointing to a functional relationship between both factors. However, two crucial questions remained unanswered:

- ! *Is deadenylation necessary for decapping and decay?*
- ! *Can other factors independently trigger these processes?*

Those questions could be answered with a simple YES or NO. Hence, we simplified our chase approach, concomitantly reducing the high price of our sequencing. Two time points were used: 0 and 13 minutes. 13 minutes was selected as best reflecting decay and deadenylation of RPG mRNAs under heat response. Importantly, a similar 'two point' approach was previously employed to estimate mRNA decay rates by the Cramer group (Miller et al., 2011 et al., 2012: DOI: 10.1101/gr.130161.111; Sun et al., 2013 DOI: 10.1016/j.molcel.2013.09.010) and in our previous works (Tudek et al., 2018: doi: 10.1016/j.celrep.2018.07.103.; Schmid et al., 2018:doi: 10.1016/j.celrep.2018.07.104.).

As a result, in the revised manuscript, we amended Figures 6D-E to clearly indicate that each dataset had its dedicated control, but now we also show some of the underlying raw data concerning the *ccr4Δ pan2Δ* data in Fig. EV5D.

This section is also supplemented with novel data that should facilitate understanding of the biological conclusions (for novel data presentation see also response to Reviewer 2 main point).

Efficient estimation of decay rate from control and single measurement

Non-essential suggestion:

I feel that this manuscript has an excess of figures. This makes difficult to identify the key supplementary figures bringing conceptual advancement from the more descriptive ones.

ANSWER: Indeed, there are a lot of figures in our manuscript, and after careful analysis we decided to remove some of them. Unfortunately, our possibilities are very limited. In the manuscript, we present data derived from 175 DRS sequencing of 45 different experimental variants. In addition, there are two modeling methods that require appropriate visualization. However, to meet the expectations of the Reviewer the former Supplemental Figures 2 and 6 have been moved to the Appendix as they were not essential to the main message of the paper. Furthermore, we deleted former panels S6F-G. We have also incorporated new schematic models in Figures 4B, 5C, 5F, and 6J to enhance the clarity and comprehension of the manuscript's main message.

Response to Reviewer 2

General summary

Here, the authors have established that the anchor-away system applied to depletion of nuclear Mex67 is an effective tool to study cytoplasmic mRNA deadenylation and degradation that correlated well with previous metabolic labelling studies. The study reports quantitative modelling of cytoplasmic mRNA deadenylation and degradation in yeast. Two numerical models for mRNA deadenylation are developed that fit the experimental data very closely; one is precise but limited to transcripts with high read scores, while the other is more broadly applicable but less accurate. A transcriptome-wide deadenylation rate is determined to be ~ 10As/min, with variation between individual transcripts that correlates with mRNA abundance, codon optimality and response to stress. As reported previously (Tudek et al., 2021), the data support the conclusion that high abundance mRNAs have short poly(A) tails and are deadenylated slowly, while low abundance transcripts have long poly(A) tails and are deadenylated more rapidly. The approach shows mRNAs to be sensitive to decapping when the poly(A) tail has been shortened to ~ 20 nucleotides. Pab1 depletion led to an increase in poly(A) tail length, with a greater effect on low abundance mRNAs. Deadenylation rates were shown to be constant for a given transcript, correlated inversely with mRNA abundance and were low for transcripts from house-keeping genes, and that these correlated with codon optimality. Deadenylation was shown to be accelerated upon heat stress and sensitive to depletion of Pan2 and Ccr4 in a synergistic manner.

The experimental data is of a very high technical quality. The development of accurate numerical models for mRNA deadenylation provide a basis for the future quantitative analyses of more detailed mechanistic aspects of mRNA degradation on a transcriptome-wide scale in yeast and other systems.

ANSWER: *We thank the reviewer for the positive opinion about our data.*

main points

The only main point that I would raise is that it is not clear to me why there is no data addressing the role of Caf1 in deadenylation. There are several reports supporting a contribution of Caf1 in mRNA deadenylation in yeast and mammalian cells (e.g. Tucker et al., 2001; Webster et al., 2018; Mostafa et al., 2020; Yi et al., 2018).

ANSWER: *Indeed, the CCR4-Not complex is equipped with two catalytic subunits, and our omission of Caf1 (Pop2 in yeast) was puzzling and needed an explanation. First of all, in *S. cerevisiae*, in contrast to humans, plants or even fission yeast Pop2 has a mutated active site, which leads to orders of magnitude lower catalytic activity (see, for instance, Jonstrup et al, 2007: DOI: 10.1093/nar/gkm178). More recently, it was suggested that *S. cerevisiae* Pop2 has a very unusual catalytic properties with an ability to generate 3' phosphorylated products (Ye et al, 2021: DOI: 10.1261/rna.078006.120). At the same time, our previous research (Tudek et al, 2021: DOI: 10.1038/s41467-021-25251-w) indicated that although in strains deleted for either CCR4 or PAN2 there is a similar mean increase in mRNA pA-tail length can be seen (7 to 7.5 As; Figure EV5B), Pan2 preferably targeted mRNAs of high abundance, whereas Ccr4 those of lower expression (Tudek et al., 2021: DOI: 10.1038/s41467-021-25251-w and revised Figure EV5C).*

To describe the role of the second deadenylase present in the CCR4-NOT complex, for the revised version of the manuscript, we conducted DRS of a yeast strain with a deletion of the *POP2* gene. We found that the lack of *POP2* had little effect on transcriptomic pA-tail lengths (1.9A increase on average; revised Figure EV5B), and in contrast to *Ccr4* or *Pan2*, its loss did not affect RPG mRNA decay dynamics (novel Figure EV5E and supporting panel below). However, even this modest increase in pA-tail lengths was reasonably well correlated with changes previously observed in *ccr4Δ* (novel Figure EV5C), suggesting that both deadenylases' activity is coordinated within the same complex. Moreover, we included a whole transcriptome pA-tail profile for a wild-type, *ccr4Δ*, *pan2Δ* and double *ccr4Δ pan2Δ* mutant (revised Figure 6H). This profile shows that in the double mutant strain, the pA-tails are, on average, 60 adenosines long, with the 95th quantile equal to 90 adenosines. None of the single mutants (*ccr4Δ* or *pan2Δ*) produced such elongated pA-tails. Importantly, the previously established lengths of newly synthesized in the nucleus mRNA pA-tail were shown to be 60 with a fail-safe mechanism limiting adenylation to 90 As (Turtola et al, 2021: doi: 10.1101/gad.348634.121). This strongly suggests that in a double *ccr4Δ pan2Δ* mutant, very little deadenylation occurs in the cytoplasm, but RPG mRNA decay still proceeds (Figure 6D-E and novel EV5D).

Novel Figure EV5B

Novel Figure EV5C

Novel Figure 6H

Revised Figure 6D-E

Support to Fig. EV5E:
[RPG decay at 37°C normalized to 25°C]

Novel panel to Figure EV5E

Figure EV5 Legends: **B.** Mean pA-tail lengths of individual mRNAs in wild-type cells were subtracted from the mean lengths observed in deadenylase mutants (*ccr4Δ* or *pan2Δ* or *pop2Δ*) and displayed in the form of a density plot. The mean change in each strain is marked as a vertical line and given in the figure legend. **C.** Scatterplots show the absolute change in mean pA-tail length in *ccr4Δ* - WT on x-axis compared to *pop2Δ* - WT (top graph) or *pan2Δ* - WT (bottom graph). Transcripts of high abundance or RPG mRNAs are highlighted in blue and red respectively. Spearman rho coefficients and the number of mRNAs compared (n) are listed in each panel legend. **D.** Plots show the raw pA-tail length density distribution of two example upper quantile in the wild-type strain heat stress chase compared to the control and 13 min heat shock of a double *pan2Δ ccr4Δ* mutant strain. The plots present raw data also shown on Figure 6E. The fitted lines are for orientation purposes only and link the local maxima either with local or linear regression. **E.** Abundance of selected RPG mRNAs normalized to 25S rRNA in control or *ski2Δ*, *pop2Δ*, *ccr4Δ*, *pan2Δ*, *lsm1Δ*, and *dhh1Δ* cells at 25 °C compared to 13 min heat shock at 37 °C determined using reverse transcription coupled to qPCR. Single dots show biological replicate values used to calculate the mean.

Figure 6 Legends: **D.** Time-dependent change in levels of RPG mRNAs in Mex67-depletion, thiolutin and heat stress time course normalized to control. Fitted lines show the group trend with a 95% confidence interval in gray. Point experiments show data normalized to corresponding controls obtained for 13 min heat stress of a double *ccr4Δ pan2Δ* mutant (black triangle) or Mex67-AA strain which was treated or not with rapamycin 3 min prior to heat stress for 13 min at 37 °C. For those the median is shown along with the data-point density contour. Experiment legend is listed next to panel 6E. **E.** For the same samples as in Fig. 6D the graphs shows the time-dependent change in pA-tail length value of the 95-90-85-80-75th quantiles normalized to the control sample. Normalization allows for direct comparison of all changes in the group, without the need to group transcripts by the control pA-tail length. **H.** Normalized pA-tail density plot showing the whole transcriptome adenylation profiles in wild-type cells compared to single *ccr4Δ* or *pan2Δ* mutants and a double *ccr4Δ pan2Δ* strain. The gray box shows the range of adenylation produced de novo in the nucleus by the polyA-polymerase Pap1 on pre-mRNAs (Turtola et al, 2021).

minor points

There is no Methods section in the submitted main text of the manuscript. The Methods section in the Supplemental data is overly long for the main text; a concise version should be included in the main text. This should include sufficient information for the reader to follow the mathematical modelling in particular.

ANSWER: Yes, indeed the Reviewer is right. In the original submission, the entire Methods section was provided in the Supplemental Data file. We apologize for this oversight. In the revised version we have extracted the most relevant parts and integrated them into the main text. The part referring to the numerical model is transferred in its entirety. It might seem long, but it's necessary for proper understanding of the modified gamma model.

Supplemental Tables 1 and 2 are not in the Supplemental data.

ANSWER: Supplemental Tables 1 and 2 were originally provided in the form of excel files, which were uploaded as Supplemental Datasets and thus did not appear in the main pdf-formatted Supplemental Data file. We apologize for the confusion. Those are large tables, and

in the revised version, we have again provided them as Datasets. We have carefully corrected the relevant parts of the text to ensure that those tables are referred to as Dataset Table 1 and 2. In contrast Tables that are included in the pdf Appendix file are now called Appendix Table 3, 4, 5 and 6 (Tables 5 and 6 are novel).

Is the increase in poly(A) tail length observed at the 30-minute time-point for the q0.95 fraction (Fig. 2A) due to hyperadenylation, as noted in Fig. S1K? This should be noted when describing this data, which comes before S1K is referenced.

ANSWER: Yes, the Reviewer is right. We have now included this information on the Figure 2A and similar panels in the EV Figures (by adding the text 'hyperadenylation window') and improved the figure legend additionally citing the Jensen et al, 2001 work (doi: 10.1016/s1097-2765(01)00232-5.) in order to guide the reader to the relevant literature. Off note, Figure 1B also highlights the utility of each chase datapoint for deadenylation and decay modeling.

Revised Figure 2A:

Error bars should be included for data across the three triplicates for each time-point shown in Fig 2E

ANSWER: The panel in question on Figure 2E showed median pA-tail values for each single data-point along with a trend line drawn using local regression in Rstudio (`geom_smooth`, `loess` function). As requested by the Reviewer we have now added a 95 % confidence interval to the trend line in a color that matches the class of mRNA analyzed. The confidence intervals are well separated and confirm that each class has a slightly different deadenylation rate. This clearly improves the readability of the graph and the conclusion. We thank the reviewer for the suggestion. The figure legend was also corrected to better describe the graph features. The corresponding former Figure 4F (now revised 4G), which shows median pA-tails for the gamma predicted distributions, was also changed accordingly.

Revised Figure 2E:

Revised Figure 4G:

Similarly, error bars should be shown for the average (?) poly(A) tail length indicated for control Dcp2-AID and Xrn1-AID by the vertical lines in Fig. 3B. Perhaps the poly(A) tail length before a rapid decrease in density is observed would be more appropriate for comparison than the average poly(A) tail length.

ANSWER: In the DRS data underlying the Figure 3B and similar ones replicates were pooled together, as was previously done in other works (Biliska et al., 2020: doi: 10.1038/s41467-020-15835-3). This was warranted by our internal quality control procedures, which we unwisely omitted in the original submission. To amend for this in the Appendix we prepared a new 2.5 page section no. 1.10 entitled 'Description of dataset quality control and considerations regarding the estimation of decay and quantile deadenylation coefficients' that is supported by novel Appendix Table 6 and Appendix Figures 3 and 4. The section specifies the basic quality control measures and explains in detail the quantile deadenylation and decay rate estimation.

Regarding specifically the Figure 3B; below we show violin plots for each replicate separately with the distribution parameters marked for comparison. Note that the distributions are not Gaussian and therefore a visual comparison is necessary in addition to analysis of simple metrics such as mean, median and variance. The related PCA plot further supports replicate reproducibility. These analyses were done for all plots presented in the paper and warranted pooling the replicates together. We strongly feel that such plots are much less straight forward compared to the original one and, respectfully, we would like to adhere to the previously proposed aesthetics.

Reviewer response Figure 1 A. Violin plots present global distribution of pA-tail length in biological replicates of Dcp2-AID, Xrn1-AID and WT-AID experiments. Red dashed line is a mean pA-tail length. **B.** Principal component analysis was done based on transcript abundance calculated based on DRS data for Dcp2-AID, Xrn1-AID and WT-AID replicates.

The authors should comment on why the poly(A) tail length of individual transcripts shows a sensitivity to Xrn1 digestion but there is no effect at a global level (Fig. S2G)? Is this because the transcripts (other than RPS13?) are of low abundance?

ANSWER: Yes, indeed the global profile after Xrn1 digestion in vitro is not markedly different from the control. This is expected because, in wild-type cells, there are very few uncapped mRNA species. Exonucleolysis closely follows decapping, facilitated by a physical interaction between the decapping enzyme or its co-factor and Xrn1, as shown in higher eukaryotes (Braun et al, 2021: DOI: 10.1038/nsmb.2413; Charenton et al, 2017: DOI: 10.1073/pnas.1711680114) and the high efficiency of Xrn1 catalysis (Athapattu et al, 2021: doi.org/10.1093/nar/gkab001). However, some small differences can be observed at the individual transcript level, as illustrated in separate panels in Appendix Figures 1I (former Figure S2). Those differences in pA-tail distribution are observed around the pA-tail profile peak. Since this peak fall for individual mRNAs anywhere between 18 to 30 As, at the scale of the whole transcriptome, which is a sum of all mRNAs, those differences tend to even out. We have now amended this section to better convey this message and included citations of the work by Braun et al (2012) and Charenton et al (2017).

The authors should describe how the estimates of transcription (plotted as blue lines in Fig. 4B) were determined - I couldn't see this described in the Methods section.

ANSWER: Indeed, the method to estimate new mRNA transcription was only briefly mentioned in the Figure legend but not in the methods section, which we have now corrected by adding a new Methods paragraph. We thank the Reviewer for spotting this issue. The main text was also corrected to specify better the purpose of producing a transcription estimate. The pA-tail mRNA profiles that we describe as estimates of new mRNA transcription (blue lines in Figures 4C and EV2B) were obtained by subtracting the modified gamma distribution from the experimental pA-tail profile, followed by log2 fitting. This procedure was performed for Mex67-AA distributions obtained for all time points but yielded only significant estimates of new mRNA production in control samples (compare control distributions in Fig. 4C with profiles obtained after Mex67 depletion in EV2B and see graphs deposited in Mendeley data: doi: 10.17632/2j3hh37zszs.1). This is consistent with previous observations showing that Mex67 depletion leads to

transcriptional arrest (Tudek et al., 2018; Schmid et al., 2018), so at each subsequent time point of the Mex67-AA chase experiment, the effect of the transcription factor on the shape of the pA tail distribution is diminishing. Moreover, we observed that our estimates of new pA-tail profiles clustered around pA-tail length 40-60 in control samples (revised Figure 4C), which was consistent with previously published works, determined the length of newly synthesized pA-tails in yeast (Turtola et al., 2021: DOI: 10.1101/gad.348634.121; Tudek et al., 2021: DOI: 10.1038/s41467-021-25251-w; Aibara et al., 2017: DOI: 10.1093/nar/gkw1224). In sum, this procedure was performed to ensure that the modified gamma equation correctly describes yeast pA-tail profiles, and any deviations from it are an offset of increased nuclear mRNA synthesis.

The section of text on p6 describing how the microscopic deadenylation rate is determined from the transcriptomic deadenylation rate could be clarified in simpler terms.

ANSWER: This section was extensively rewritten to improve clarity. It has been split into two sub-sections. The first provides two methods for estimation of the enzymatic deadenylation rate and is now supplemented with additional mathematical proof. The second section verifies this estimate by performing an *in silico* deadenylation simulation and showing the dimensionless scaling factor that designates consecutive deadenylation steps correlates in a linear manner with the experimental time (figures EV3D-E). For more clarity a novel Figure 4K is inserted that lists all the deadenylation rate modeling steps executed under the modified gamma model.

In brief, the ratio between both deadenylation rates can be estimated using two methods and is equal to either:

1. the slope value of the linear functions fitted into the pA-tail distributions displayed in a y-logged-scale (former Figure 4I now revised Figure 4J) or
2. $R_{apparent} = R_{enzymatic} \cdot (1 - e^{-\gamma_{rate}})$.

The second method is derived from the modified gamma model. In a mixed pA-tail length population (X_{N_A}) the microscopic (enzymatic) rate of removal of the terminal adenosine of a pA-tail of a given length N_A is seemingly diminished by the deadenylation of a polyadenosine chain that is one adenosine longer (X_{N_A+1} of N_A+1 , respectively). In other words, the loss of the total number of pA-tails of 39 adenosines (because of their deadenylation to 38 As) is apparently reduced by the products of decay of pA-tails of 40 adenosines to 39 As. The difference between the enzymatic (R) and apparent deadenylation rate (R_{app}) is encapsulated by the first order reaction formula, which is widely used in molecular biology:

$$-R_{app} \cdot X_{N_A} = dX_{N_A}/dt = -R \cdot X_{N_A} + R \cdot X_{N_A+1} = -R \cdot X_{N_A} \cdot (1 - X_{N_A+1}/X_{N_A}). \text{ (novel numerical proof)}$$

Within the range of pA-tail longer than 20As; that are bound by at least one Pab1 molecule the X_{N_A+1}/X_{N_A} ratio equals $e^{-\gamma_{rate}}$, and the apparent deadenylation rate can be simply expressed as:

$$R_{app} = R \cdot (1 - e^{-\gamma_{rate}}),$$

where γ_{rate} is a parameter of the modified gamma distribution estimated for the combined RNA population (Figure EV3A). Since for short experimental times, the γ_{rate} values were estimated to be ~ 0.1 (Figure EV3A), the microscopic deadenylation rate is therefore ~ 10 times higher, ~ 10.9 A/min (confidence interval: 8.4-15.4).

At the end of the manuscript there is some data addressing mRNA turnover in strains mutant for decapping factors. It's not clear why mutants of the SKI complex have not been addressed.

ANSWER: The Reviewer suggested an important control that we did not perform. The SKI complex is responsible for removing mRNAs that are difficult to translate (Tomecki et al, 2023: DOI: 10.1002/wrna.1795). It is possible that under stress conditions, RPG mRNAs could display translation defects that would lead to activation of the SKI complex and the exosome. Therefore, following the Reviewer's remark, we analyzed RPG decay during heat stress in a *ski2Δ* mutant but found that those mRNAs underwent the same decay dynamics as in the wild-type strain (revised panel on Figure EV5E inserted below). We thus conclude that the SKI complex does not play a role in RPG mRNA decay under heat stress conditions, which we now discuss in the main text. Since RPG mRNA decay could be rescued by depletion of Dcp2 or deletion of its co-factors DHH1 and LSM1 (Figures EV5E and EV5F → see also response to the next question) we conclude that RPG mRNAs are decayed via the decapping-Xrn1 pathway, as is the case at steady-state.

Novel Figure EV5E panel 1:

At the end of the Discussion section, it is stressed that the analyses reveal mechanisms that allow decapping in the absence of deadenylation under stress conditions. Does this relate to the degradation observed in the *ccr4Δ pan2Δ* mutant when there is little deadenylation? If that is the case, the degradation observed has not been shown to be dependent upon decapping.

ANSWER: We did verify that RPG mRNA decay was fully dependent on decapping, but the legend of the relevant graph was mislabeled. We are very grateful for spotting this issue. We also verified other panels for any additional mistakes. The revised Figure EV5F (below) shows an analysis of RPG mRNA levels in control and heat stress conditions in a Mex67-AA Dcp2-AID strain. This strain allows for blocking export with the addition of rapamycin or blocking decapping following the addition of auxin to the media (the main decapping enzyme is tagged with an Auxin Inducible Degron tag, and the efficiency of its depletion is verified in Appendix Figure 1A). The analysis shows that following Dcp2 depletion RPG mRNA decay under heat stress is strongly impaired. Furthermore, we have also shown that the deletion of decapping co-factors DHH1 and LSM1 resulted in a partial rescue of RPG mRNA levels under heat shock, as

shown in Figure EV5E (also below). Regardless of the graph mislabeling, those data were insufficiently discussed in the main text, and the Reviewer was right to point this out. We have now extensively rewritten this section.

Fig. EV5F:

Fig. EV5E panel 2:

Response to Reviewer 3

This manuscript from Czarnocka-Cieciura and colleagues explores the quantitative landscape of deadenylation and decay in budding yeast. With the advent of third-generation direct RNA long-read sequencing technology, the ability to measure poly(A) tail lengths (easily) has opened the world of deadenylation dynamics, making this manuscript timely and innovative. Another key innovation is the use of a Mex67-anchor allele, which allows extremely rapid shut-off of nuclear export and fast degradation of nuclear transcripts - together this tool enables accurate measurements of cytoplasmic RNA fates. There are many strengths to this paper. First, the authors generate quantitative models for global and gene-specific deadenylation rates. Second, the results highlight the perils in using steady-state poly(A) tail lengths due to complex relationships between deadenylation, decapping, and export processes - this manuscript, as well as other recent papers, reconciles the surprising conundrum that more unstable transcripts have longer (not shorter) poly(A)-tail lengths at steady state. Third, the data provide strong evidence for the protective role of Pab1p in blocking decapping. Although I cannot support publication in its current form due to a few major weaknesses, once these have been dealt with, this manuscript will be an important contribution to the field.

ANSWER: *We thank the reviewer for the overall positive opinion about our manuscript*

Major comments:

1. It would be useful to discuss more quality-control metrics for the original datasets. E.g., how similar were the half-lives and tail length (dynamics) between replicates? How were replicates combined? What was the minimum read number for each gene? How many genes? Adding some of this information up-front will help orient the reader for the more sophisticated analyses.

ANSWER: *We agree that the analysis of the data coherence and quality was not sufficiently tackled in the original version of the manuscript, as it was only represented by Supplemental Tables 1 and 2 (now named Dataset Tables 1 and 2). Therefore, we now included a novel 2.5 page section in the manuscript Appendix that describes the data quality and considerations regarding the quantile deadenylation and decay rate modeling. The section 1.10 is entitled 'Description of dataset quality control and considerations regarding the estimation of decay and quantile deadenylation coefficients' and refers to the two novel Appendix Figures 3 and 4 and to the new Appendix Table 6. This large chapter and supporting elements show the number of reads used for the subsequent analyses with division into mRNAs and other subclasses, including RPG mRNAs, and the correlation between replicate features such as read count and mean pA-tail length. The novel section also visually presents the raw data for the chase experiments and discusses the details of decay and quantile deadenylation rate estimation.*

2. The authors convincingly highlight the perils of relying purely on steady-state poly(A) tail lengths to inform an understanding of dynamics, and yet this is their strategy for a variety of experiments, including Xrn1, Dcp1, and Pab1 depletion. In the case of Pab1, early reports from Caponigro and Parker suggest that decapping occurs on long-tailed RNA species. Thus, there are alternative explanations that the authors do not address to explain the poly(A)-tail length changes upon Pab1 depletion. In particular, looking at deadenylation rates in this line would substantially strengthen this section and this paper.

ANSWER: *We appreciate the Reviewer's comment on this matter. While we acknowledge that our treatment of the decapping issue in the Pab1 strain was somewhat limited, this was due to*

specific technical constraints rather than an oversight. Unfortunately, the depletion kinetics of Pab1 using the auxin-inducible degron system were very slow, with 40 and 20 % of protein remaining intact in the cells after 1 and 2 hours of depletion time, respectively (see Reviewer Response Figure 1A and 1B below). We are certain that near complete removal of Pab1 is possible as the strain is largely non-viable on auxin, as is expected since Pab1 is an essential protein (Reviewer Response Figure 1C). However, the time required to deplete Pab1 is many times longer than the time required to remove Mex67 from the nucleus (2.5min) and degrade most of cellular mRNAs (average mRNA half-life is 9min, see Figure 1), which precludes the use of this strain for deadenylation modeling. Despite this, the obtained data clearly point towards a function of Pab1 in promoting deadenylation, which we now explain in more detail. The Pab1 role in promoting deadenylation is made clear upon analysis of single mRNA examples grouped by decreasing half-life (Reviewer Response Figure 1D split into novel Fig. 3C and novel Appendix Figure 1L). The shorter the mRNA half-life the more prominent is the accumulation of a distinct, long pA-tailed mRNA fraction. The accumulating pA-tail lengths corresponded in length to those that are newly-made in the nucleus by Pap1 (60As on average with an adenylation fail-safe at 90 As, Turtola et al, 2021: DOI: 10.1038/s41467-021-25251-w). Since during Pab1-depletion, transcription is not inhibited and budding yeast do not possess cytoplasmic adenylases, those accumulating mRNAs can only be issued from new transcription. This clearly suggests that Pab1 is indeed required for deadenylation. Since all mRNAs have increased mean pA-tail lengths we conclude that the phenotype is universal but only easily showed on mRNAs that are rapidly degraded and therefore have no large transcript pool produced prior to (the inefficient) Pab1-depletion.

Moreover, in our work, we do not deny the occurrence of decapping on mRNAs with long pA tails as postulated by Caponigro and Parker. Furthermore, we do show it happens using an orthogonal approach. In the last section of the manuscript (data related to revised Figures 6D-E and novel Figure 6H and EV5) we show that a near complete impairment of deadenylation in a double *pan2Δ ccr4Δ* strain does not block decapping and decay. In this extensively revised section, we now show that Ccr4 and Pan2 are the two main budding yeast deadenylases (revised Figure 6H -> Reviewer Response Figure 1E), with the contribution of Pop2 being minor at best (see novel Figures EV5B,C,E). The pA-tail distributions of cells deleted for PAN2 and CCR4 genes have a mean of 60As with the 95th distribution quantile being equal to 90 As (revised Figure 6H and EV5D). This indicates that in this double *pan2Δ ccr4Δ* mutant mRNAs newly produced in the nucleus are barely deadenylated. We further show that despite deadenylation blockage, during heat stress, decay of RPG mRNAs proceeds (Figures 6D-E -> Reviewer Response Figure 1F, look at the black triangle relative to control). In contrast, loss of Dcp2 or its co-factors inhibits or significantly delays RPG mRNA decay, showing that during heat stress those transcripts are, in fact, decapped prior to decay (see revised Figures EV5E and EV5F). Therefore, our analyses support the early work by Caponigro and Parker (1995), especially since they were performed in different genetic backgrounds.

We now included the reference to the pioneering work by Caponigro and Parker (1995).

Reviewer 3 response Figure 1. **A.** Western blot analysis of Pab1 depletion using the auxin inducible degron system. Samples were taken for the control and 1 or 2 hours after addition of 1 mM auxin sodium salt to the media. Note that lanes 1-6 are shown in Appendix Figure 1. The blot included here also shows additional lanes that were used for Western blot quantifications. Those are serial dilution of the control sample 4. Those dilutions were used to estimate the level of Pab1-AID remaining after 1 and 2 hours auxin treatment (next panel B). **B.** Graph shows an estimation of the Pab1-AID protein levels remaining after 1 or 2 hours after addition of auxin to the media. **C.** Serial dilution spot test that shows the growth of the double Mex67-AA and Pab1-AID strain on auxin or rapamycin. Since both proteins are essential, as expected, cells are inhibited in growth in both conditions. **D.** Series of graphs show the pA-tail length distribution change for individual mRNAs following Pab1-depletion for 1 or 2 hours. Graphs are ranked according to mRNA half-life from the most to the least stable. **E.** Novel Figure 6H showing the pA-tail length whole transcriptome distribution of WT, *pan2Δ*, *ccr4Δ* and *pan2Δ ccr4Δ* strains. In gray is highlighted the range of length of pA-tails newly synthesized in the nucleus by Pap1. **F.** Figures 6D and 6E which show time-dependent changes in the mRNA abundance or quantile pA-tail length normalized to respective controls for the three chase experiments and for several point heat stress experiments performed in a *pan2Δ ccr4Δ* or under export block.

3. It is unclear of the extent to which the authors directly looked at how much decapping rates vary between different mRNAs. Critically, in the mammalian paper from Eisen et al., one of the main conclusions is that more rapid deadenylation coincides with more rapid decapping. There is no discussion of this idea in the text - and it is unclear whether this possibility was explored in the modeling. If not, that is a critical point. If the authors obtained similar decapping rates, then that is an important result.

ANSWER: We agree that an accurate estimate of the decapping rate for transcripts would be an interesting extension of our deadenylation studies. However, we did observe that uncapped mRNAs were scarce in the wild-type RNA sample (Appendix Figure 1F-H). This suggested that following decapping Xrn1-mediated decay occurs almost instantaneously. Our assumption is supported by published data that show a physical link between the decapping enzyme or its cofactor and Xrn1 in the higher eukaryote system (Braun et al., 2021: DOI: 10.1038/nsmb.2413; Charenton et al., 2017: DOI: 10.1073/pnas.1711680114). Recent studies suggest that Xrn1 enzyme can display a decay rate of 26 ± 5 nt per second (Athapattu et al, 2021: doi.org/10.1093/nar/gkab001), which would imply that an average yeast mRNA of around 1000 nt would be degraded in under 1 minute. Much faster than the average mRNA half-life of 9 min. Consequently, in the modified gamma model, we expressed this strong link between decapping and decay by the $\tanh(0.096 * pA)$ correction to the classical equation and concluded that estimates of mRNA half-lives should be a good reflection of the decapping rates or at least be a very good proxy.

Furthermore, in the manuscript, we calculated half-lives of individual mRNAs and found that they spanned from minutes to dozens of minutes with the transcriptomic median equal to 9 min (data related to Figure 1, partially shown below in Reviewer 3 response Figure 2 → Figure 1D histogram). Importantly, our half-lives correlated strongly with previously published data obtained in budding yeast using orthogonal methods (Reviewer 3 response Figure 2 → Figure 1E-F). When we compared the half-life estimate with the deadenylation rate calculated for each transcript using the quantile method, we noticed a strong positive correlation (Reviewer 3 response Figure 2 → Figure 5D.-E.), with slowly deadenylated mRNAs displaying long half-lives and rapidly deadenylated mRNAs undergoing fast degradation. Moreover, we saw that both factors can be similar within functional mRNA groups as previously postulated for half-lives (Fig.

5H; Wang et al, 2002: doi:10.1073/pnas.092538799). We can therefore surmise that in yeast, as in higher eukaryotes, there is a direct relationship between decapping-decay and deadenylation rates. To better highlight these observations, we have now inserted a mini-model panel of Figure 5F and improved the main text, formulating this conclusion more straightforwardly.

Finally, in the last extensively revised section of the manuscript, we further tested the causality of the deadenylation-decay relationship under stress conditions. This was possible since a group of highly abundant 129 RPG mRNAs displayed very similar half-lives and deadenylation rates at steady-state and were transcriptionally silenced during heat stress (Vinayachandran et al, 2018). In the revised version, we better show that RPG mRNA decay during heat stress is dependent on decapping, similar to the steady-state condition. To this end, we show that depletion of the main decapping factor Dcp2 or deletion of its co-factors DHH1 or LSM1 results in rescue of RPG mRNA decay, whereas loss of SKI2, which is involved in the 3'-5' decay pathway has no effect (revised Figure EV5E and EV5F). With this crucial control solidified, we showed that during the brief response to heat stress, deadenylation and decay of RPG mRNAs was accelerated, which further supported the functional link between both factors. However, we also demonstrated that a near complete block in deadenylation, which could be obtained in a double *pan2Δ ccr4Δ* mutant, only slowed down RPG decay but did not block it (revised Figure 6D-E, novel 6H and EV5D). This indicates that deadenylation, though acting as a potent stimulator, is not a prerequisite for decay. Consequently, decapping can occur on long-tailed mRNAs. We had also shown that RPG mRNAs, with the exception of RPL4A, differ from other transcripts as most of them require ongoing nuclear export to induce decay (Fig. 6D-E, EV5F-G). This indicated that, aside from deadenylation, there are other triggers to mRNA decapping and decay. To better convey those complex conclusions, we have now inserted a model in Figure 6 (Reviewer 3 response Figure 2→ Figure J).

Off note, the different deadenylation rates of individual transcripts can be reconciled with the modified gamma model that calculated a constant deadenylation rate of 10 A/min. Our deadenylation simulation from a sharp ~70 A distribution (akin to a transcription burst, Figure EV5H, also shown below) shows a gradual increase in the distribution variance with deadenylation time. This can only be a sign of the differential deadenylation rate of individual molecules (mRNAs) within this averaged constant speed.

Link between decapping and deadenylation rate

Figure 1 → decapping and mRNA decay rates vary between transcripts

Figure 5 → decapping and quantile deadenylation rates correlate

Figure 6 → decapping and quantile deadenylation rates change in response to heat stress

Reviewer 3 response Figure 2 Figure 1D mRNA and ncRNA half-life distribution calculated from the Mex67-depletion time **Figure 1F**. Correlation of half-life values in Mex67-depleted sample and half-life values from Miller et al, (2011). **Figure 5C**. Histogram shows distribution of single mRNA terminal adenosine half-lives for the entire coding transcriptome (light-red bars) or mRNAs of high abundance (gray bars). Vertical lines designate the value of the group median. **Figure 5E.-F**. Comparison of the decay and quantile deadenylation rates (D.) or log2-scaled half-life of mRNA to that of its terminal adenosine (E.). Density plot refers to all mRNAs, whereas mRNAs with at least 70 reads in replicate A are shown as single dots with a blue regression line. Spearman rho correlations were calculated separately for each set. **Figure 5G**. Scheme highlighting the strong, presumably causative, link between deadenylation and decapping that is evidenced by data presented in Figures 5E.-F. **Figure 6J**. Scheme showing the role of deadenylation in mRNA decapping and decay. For most of the transcriptome, deadenylation is a rate-limiting factor that dictates the onset of decapping. RPG mRNAs are a special group, for which deadenylation can accelerate decay in conditions such as heat stress, but ultimately the decapping and decay is dependent on an unknown factor which is dependent on nuclear export. **Figure EV5H** In silico deadenylation modeling from a sharp pA-tail distribution (red line). The consecutive deadenylation steps are shown with a collection of blue-green lines.

4. Finally, this paper would be substantially strengthened by repeating the deadenylation dynamics experiments and modeling in a line lacking a trans-factor like Puf3. This experiment would provide quantitative information on how such a trans-factor (that has been studied in vitro) affects deadenylation and/or decapping rates.

ANSWER: We agree that investigating the *in vivo* role of Ccr4-NOT cofactors in deadenylation is an interesting question that has been the focus of numerous studies, primarily through CRAC/RIP and differential expression analyses. These investigations have identified potential targets of Puf proteins; however, a comprehensive analysis of the *in vivo* impact of Puf proteins on deadenylation remains incomplete. In our opinion, this topic merits a dedicated study rather than being included in the current submission.

Nonetheless, to assess the utility of the *puf3Δ* background in our studies, we generated a DRS dataset, which revealed negligible differences in the pA-tail length distributions for the whole transcriptome between WT and *puf3Δ* (Reviewer 3 response Figure 3A), which was likely due to the low share of Puf3-targets Lapointe et al (2018; DOI:10.1016/j.cels.2017.11.012 - 4% of all mRNAs summing up to 8.5% of the coding transcriptome mass).

Surprisingly, for the set of 233 Puf3-bound mRNAs determined previously by Lapointe et al (2018), the effect of *puf3Δ* on pA-tail distribution was also minor and represented pA-tail shortening rather than expected elongation (Reviewer 3 response Figure 3B). Although the Passmore group demonstrated that *in vitro* Puf3 accelerates deadenylation of a substrate containing its binding sites, *in vivo* studies conducted on the most studied Puf3-bound transcript, COX17 (Lee et al, 2010:doi:10.1016/j.jmb.2010.04.034), suggested pleiotropic events. Those were due to the indirect effects of Puf3 deletion on the Pan2/3 deadenylase activity. Indeed, our DRS data for COX17 replicates the Northern blot analysis previously published by Lee et al (2010) (compare Reviewer 3 response Figure 3C and 3D). In sum, unlike the *in vitro* system, *in vivo* conditions present complications, even in conditions of pulse-chase analyses implemented by Lee et al (2010) because both Ccr4-NOT and Pan2/3 deadenylases are recruited to substrates independently of Puf3. This redundancy makes it challenging to isolate *in vivo* the specific stimulatory effect of Puf3 from the overlapping catalytic activities of

Ccr4-NOT and Pan2/3 (Webster et al., 2018; DOI:10.1016/j.molcel.2018.05.033; Schäfer et al., 2019; DOI:10.1016/j.cell.2019.04.013).

In summary, the regulation of deadenylation by Puf proteins is complex and extends beyond the recruitment of the Ccr4-NOT complex to Puf-recognized mRNAs. As a consequence, more focused, dedicated and complex studies should be designed. Importantly, our study presents a good preparation for such a work as it comprehensively shows the redundancy between Ccr4-NOT and Pan2/3 complexes on a large group of mRNAs (see response to Reviewer 2 main point).

Reviewer 3 response Figure 3. A. Whole-transcriptomic pA-tail distribution of a wild-type and *puf3Δ* strain. Dashed lines indicate the median poly(A) tail lengths: *puf3Δ* - 31, wild-type - 30. **B.** pA-tail distribution of potential Puf3 targeted transcripts (233 mRNAs identified by Lapointe et al 2010) for a wild-type and *puf3Δ* strain. Dashed lines indicate the median poly(A) tail lengths: *puf3Δ* - 32, wild-type - 34. **C.** COX17 mRNA pA-tail distribution in a wild-type and *puf3Δ* strain. **D.** Northern blot by Lee et al (2010) assessing COX17 mRNA pA-tail lengths after pulse-chase in WT and *puf3Δ* strain.

Minor comments:

1. In displaying the modeling and other results, sometimes it is not clear from the text or figure whether all mRNAs are being modeled (and, if so, is this a weighted average of them or is just based on raw reads?), etc. More clear labeling within the figures and text would help with clarity.

ANSWER: *We have now carefully improved the Figure descriptions. The modified gamma modeling was performed on a sum of reads mapping to each category (all mRNA, high, low abundance, RPG and non-RPG, single mRNA examples).*

Data availability statement

1. Basecalled nanopore sequencing data: Gene Expression Omnibus GSE272785
(<https://www.ncbi.nlm.nih.gov/geo/query/acc.cgi?acc=GSE272785> , reviewer access token: mnidkcwolfqvngf)
2. Raw nanopore sequencing data: European Nucleotide Archive (please consult the Dataset Table 2 for sample accession numbers (as ENA does not provide a reviewer access option; data will become public upon acceptance of the manuscript).
3. Source data underlying each figure panel, the R code to calculate γ_{rate} and γ_{shape} parameters are deposited at Mendeley Data, doi: 10.17632/2j3hh37zss.1, <https://data.mendeley.com/preview/2j3hh37zss?a=b2968161-df0a-41ab-8af6-b5ddfe546e54> and at the EMBO J submission system. Additionally, the R code is also listed in the Appendix section 5.

Dear Dr Tudek,

Thank you for submitting a revised version of your manuscript. Your study has now been seen by all original referees, who find that their previous concerns have been addressed and now recommend publication of the manuscript. There remain only a few mainly editorial points that have to be addressed before I can extend formal acceptance of the manuscript:

- As we are switching from a free-text author contribution statement towards a more formal statement based on Contributor Role Taxonomy (CRediT) terms, please remove the present Author Contribution section and instead specify each author's contribution(s) directly in the Author Information page of our submission system during upload of the final manuscript. See <https://casrai.org/credit/> for more information.

- Please adjust the in-text callouts for individual figures and figure panels: e.g. Appendix tables 4 and 5 appear to be missing

- Please rename your datasets (both in the file and in the text) to EXPANDED VIEW DATASETS (call-out: "Dataset EV1/2"), and the spreadsheets each need a separate "Legend" tab containing dataset title and legend information, that should be removed from the Table of Contents in Appendix PDF

- APPENDIX 1 FILE WITH ToC: nomenclature should be Appendix Figure S1-S4 and Appendix Table S1-S4 (the tables start from Appendix Table 3) throughout the Appendix file and in callouts in the ms file

- Please provide suggestions for a short 'blurb' text prefacing and summing up the conceptual aspect of the study in two sentences (max. 250 characters), followed by 3-5 one-sentence 'bullet points' with brief factual statements of key results of the paper; they will form the basis of an editor-written 'Synopsis' accompanying the online version of the article. Please also provide an altered synopsis image, making sure that the aspect ratio conforms to our website's format - it should be exactly 550 pixels wide and between 300-600 pixels high.

- Please adjust the order of the manuscript sections: Title page with complete author information, Abstract, Keywords, Introduction, Results, Discussion, Methods, Data Availability Section, Acknowledgements, Disclosure and Competing Interests Statement, References, Main figure legends, Tables, Expanded Figure Legends.

In addition our data editors have raised a number of points regarding the figures and figure legends.

1. Please add figure titles for figures EV 1, 2, 3, 4, 5

2. Please state exact p values in the legends of figures 1f; 5e-f, i; EV 1c-d, f-g; EV 4b-c.

3. Please indicate the statistical test used for data analysis in the legends of figures 5e-f, i; EV 4b-c.

4. Please note that the box plots need to be defined in terms of minima, maxima, centre, bounds of box and whiskers, and percentile in the legend of figure EV 1a.

5. Please note that information related to n is missing in the legends of figures 2a; 4d-e; EV 1a, h-i.

6. Please note that the error bars are not defined in the legends of figures 4d-e; EV 2f-g.

7. Please note that the measure of center for the error bars needs to be defined in the legend of figure 4i.

- In our standard source data check, we have noted unexplained numerical duplications in the source data. I have attached the corresponding files with the detected duplications labelled in color. Please take a look and correct as needed. A brief explanation would be very helpful.

With best regards,

Cornelius Schneider

Cornelius Schneider, PhD
Editor | The EMBO Journal
c.schneider@embojournal.org

Referee #1:

The authors have addressed all my concerns. Although it is still a complex manuscript to follow, the authors have significantly improved its readability. I have no further concerns and support its publication.

Referee #2:

The authors have extensively revised and improved the submitted manuscript and have responded appropriately to the comments that I made on the first submission. I support publication of the revised manuscript.

Referee #3:

This manuscript is substantially improved over the first submission. I have two minor points. First, the modeling sections, although clearer, are still challenging to read. Second, in the discussion, a larger integration of the results into the field's literature on differential decapping / deadenylation rates, especially findings from mammalian cells (e.g., work from Narry Kim and David Bartel), would be very helpful for the reader. I support publication of the manuscript.

September 2024

Agnieszka Czarnocka-Cieciura^{1,#}, Jarosław Poznański^{2,#}, Matti Turtola³, Rafał Tomecki^{2,4},
Paweł S. Krawczyk¹, Seweryn Mroczek^{4,1}, Wiktoria Orzeł¹, Upasana Saha⁵, Torben Heick
Jensen⁵, Andrzej Dziembowski^{1,4*}, Agnieszka Tudek^{2,*}

¹ International Institute of Molecular and Cell Biology, Księcia Trojdena 4, 02-109 Warsaw, Poland
² Institute of Biochemistry and Biophysics, Polish Academy of Sciences, Adolfa Pawińskiego 5A, 02-106 Warsaw, Poland
³ Department of Life Technologies, University of Turku, Biocity, Tykistökatu 6, 205240 Turku, Finland
⁴ University of Warsaw, Faculty of Biology, Miecznikowa 1, 02-089 Warsaw, Poland
⁵ Aarhus University, Department of Molecular Biology and Genetics – Universitetsbyen 81, 8000 Aarhus, Denmark

those authors contributed equally

* correspondence should be addressed to: atudek@ibb.waw.pl or adziembowski@iimcb.gov.pl

Czarnocka-Cieciura et al point-by-point response following 2nd revision and editorial evaluation

- As we are switching from a free-text author contribution statement towards a more formal statement based on Contributor Role Taxonomy (CRediT) terms, please remove the present Author Contribution section and instead specify each author's contribution(s) directly in the Author Information page of our submission system during upload of the final manuscript. See <https://casrai.org/credit/> for more information.

Answer: we removed the Author Contribution section and specified the contribution type according to CRediT in the Author information page of the submission system according to the following scheme:

Agnieszka Czarnocka-Cieciura: Data Curation, Investigation, Validation, Visualization, Writing – Review & Editing;

Jarosław Poznański: Conceptualization, Data Curation, Formal Analysis, Software, Visualization, Methodology;

Matti Turtola: Investigation, Validation, Resources, Funding Acquisition;

Rafał Tomecki: Investigation, Validation, Funding Acquisition;

Paweł S. Krawczyk: Software, Data Curation, Supervision;

Seweryn Mroczek: Funding Acquisition, Methodology, Supervision;

Wiktoria Orzeł: Investigation;

Upasana Saha: Resources;

Torben Heick Jensen: Supervision;

Andrzej Dziembowski: Conceptualization, Supervision, Funding Acquisition, Investigation, Methodology, Writing – Review & Editing;

Agnieszka Tudek: Conceptualization, Investigation, Formal Analysis, Funding Acquisition, Methodology, Resources, Supervision, Validation, Visualization, Writing – Original Draft, Writing – Review & Editing.

- Please adjust the in-text callouts for individual figures and figure panels: e.g. Appendix tables 4 and 5 appear to be missing

Answer: Former Appendix Tables 3-6 were re-named Appendix Tables S1-S4 and are now all cited in the main text Methods section.

- Please rename your datasets (both in the file and in the text) to EXPANDED VIEW DATASETS (call-out: "Dataset EV1/2"), and the spreadsheets each need a separate "Legend" tab containing

dataset title and legend information, that should be removed from the Table of Contents in Appendix PDF

Answer: Dataset Tables 1 and 2 were re-named to Expanded View Dataset 1 and 2 in the main text and in the Appendix section. In the first tab ("Legend" → see screen shot in response to last question - page 7) of Dataset EV1 and Dataset EV2 we now provide the title and a legend describing each column in each sheet. For Dataset EV1 we also added novel columns to facilitate data sorting in the Mex67-AA datasets (column C and D indicates if the mRNAs belongs to the RPG group and whether its transcription is repressed during heat stress). This releases the reader from the necessity of merging the table with the Vinayachandran et al, 2018 dataset). We also added missing annotation columns for the heat stress chase data.

- APPENDIX 1 FILE WITH ToC: nomenclature should be Appendix Figure S1-S4 and Appendix Table S1-S4 (the tables start from Appendix Table 3) throughout the Appendix file and in callouts in the ms file

Answer: The Appendix Tables and Figures were re-named throughout the main text and the Appendix file as required.

- Please provide suggestions for a short 'blurb' text prefacing and summing up the conceptual aspect of the study in two sentences (max. 250 characters), followed by 3-5 one-sentence 'bullet points' with brief factual statements of key results of the paper; they will form the basis of an editor-written 'Synopsis' accompanying the online version of the article. Please also provide an altered synopsis image, making sure that the aspect ratio conforms to our website's format - it should be exactly 550 pixels wide and between 300-600 pixels high.

Answer: As synopsis and bullet-point we propose the following text (the synopsis figure is submitted separately):

'Blurb':

Czarnocka-Cieciura et al using large datasets modeled mRNA deadenylation in yeast at permissive and stress conditions showing a positive deadenylation-decay correlation. Curiously, ribosomal protein mRNAs can be decayed independent of deadenylation.

Bullet points:

- *The yeast in vivo deadenylation reaction was described by a modified gamma model that calculated the transcriptomic enzymatic deadenylation rate of 10 A/min*
- *The deadenylation rates were inferred per transcript from changes to pA-tail distribution quantile values*
- *In budding yeast the deadenylation and decay rates correlate positively at steady-state and change at stress conditions, suggesting a functional link*
- *mRNAs issued from ribosomal protein-coding genes (RPGs) are an exception as they display decay mechanisms independent of deadenylation*

We also propose the following graphical abstract, which was submitted separately:

- Please adjust the order of the manuscript sections: Title page with complete author information, Abstract, Keywords, Introduction, Results, Discussion, Methods, Data Availability Section, Acknowledgements, Disclosure and Competing Interests Statement, References, Main figure legends, Tables, Expanded Figure Legends.

Answer: The order of the main manuscript sections was adjusted. The paper contains no Table section, as all out tables are part of the Appendix and two other files are submitted as Expanded View Datasets.

In addition our data editors have raised a number of points regarding the figures and figure legends.

1. Please add figure titles for figures EV 1, 2, 3, 4, 5

Answer: Figure titles were added:

Expanded View 1. Following nuclear decay of pre-mRNAs in export-block conditions the cytoplasmic deadenylation and decay of mRNAs is revealed.

Expanded View 2. A modified gamma distribution accurately describes experimental yeast pA-tail distributions.

Expanded View 3. The deadenylation process can be reconstituted in silico using the modified gamma distribution.

Expanded View 4. A simplified method based on changes to distribution quantile values can measure deadenylation rates.

Expanded View 5. RPG and non-RPG mRNAs have altered deadenylation and decay rates during response to heat stress.

2. Please state exact p values in the legends of figures 1f; 5e-f, i; EV 1c-d, f-g; EV 4b-c.

Answer: The exact p-values are now listed directly on the Figure panels. Here as an example we show the two rightmost panels of Fig. 2I (added p-value is indicated by a red arrow).

3. Please indicate the statistical test used for data analysis in the legends of figures 5e-f, i; EV 4b-c.

Answer: Throughout the manuscript the Spearman correlation coefficient was calculated and listed along its significance test results. We now indicate this more clearly in the novel Appendix section 1.9 on page 5:

Quote: 'Computation of correlation coefficients using Rstudio. Correlation coefficients presented in this study were computed using rstatix package (0.7.2) in Rstudio version 2024.042 build 764. The function used was cor_test performed using Spearman's correlation test with a 0.95 confidence interval and a two-sided correlation significance test.'

4. Please note that the box plots need to be defined in terms of minima, maxima, centre, bounds of box and whiskers, and percentile in the legend of figure EV 1a.

Answer: The Figure EV1 legend now explains the significance of the boxplot elements: 'The central line is the median. The box edges are the 25th and 75th percentiles. Whiskers extend to 1.5 times the IQR (Inter Quartile Range).'

5. Please note that information related to n is missing in the legends of figures 2a; 4d-e; EV 1a, h-i.

Answer: Where missing the n number is now listed directly on the main Figure and EV Figure panels. If the n number is the same as in any preceding Figure then that is now specified in the figure legend (e.g. Fig. 4A and 4H display the same data as Fig. 2A, but in a different conceptual context. In their legends we thus inserted the following sentence: 'The number of reads summing up to form the distribution is indicated in Fig. 2A.'). If need be additional explanation is also given in the legend, as is the case for Fig. EV1A. Here, as an example we show Fig. 2A left panel (the n number now added is indicated with a red arrow).

6. Please note that the error bars are not defined in the legends of figures 4d-e; EV 2f-g.

Answer: The figure legend was fixed (quote): '(...) The parameters are given as full colored dots supplemented with vertical standard error bars. (...)'

7. Please note that the measure of center for the error bars needs to be defined in the legend of figure 4i.

Answer: The figure legend was fixed (quote):

'As red triangles with vertical error bars are plotted terminal adenosine half-lives calculated independently for each position relative to their rank in the pA-tail, calculated from distributions adjusted for mRNA decapping (Figure EV3B). A continuous exponential function is fitted with a 0.95 confidence interval.'

- In our standard source data check, we have noted unexplained numerical duplications in the source data. I have attached the corresponding files with the detected duplications labelled in color. Please take a look and correct as needed. A brief explanation would be very helpful.

Answer:

File: GAMMA_tanh_res.csv/.ods – the columns designating the γ _shape and γ _rate parameters were duplicated and both named equally 'Estimate'. We have indeed overlooked this issue and we thus removed the duplicated columns. Furthermore, we clarified the headers of the following two sets of columns 'Std, Error', 't value', 'Pr(>|t|)' by adding 'shape_' or 'rate_' prefix as appropriate. We added a missing name for column D ('mRNA_or_transcript_group'). The file is now named GAMMA_tanh_res_corr.csv/.ods and deposited at Mendeley and in the Figure source files.

File MTH: We examined the MTH.csv/.ods files and found no issues requiring table modification, but we understand why some data appeared 'duplicated'. To facilitate navigation through the file we produced a novel 'how_to_read_the_MTH_file.txt' file and deposited it at Mendeley and in the revised Figure source folders.

Briefly, MTH files show our raw reverse-transcription qPCR data and group them by experiment and biological replicate. The sample data and sorting columns are listed in rows and grouped in columns by primer pair and RNA fraction examined (e.g. 'mRNA XX -total' or mRNA XX pA⁺). Some biological replicates (entire chase experiments) were run twice for quality control, producing two columns (the second technical replicate is marked with the 'tech2' suffix). We averaged the technical replicates and listed this mean in a third column containing the suffix 'mean'. Thus those biological replicates that lacked technical repeats appear in the 'mean' column as duplications of the preceding column (the only replicate available).

Specifically, the 'RPL36A total mean' in column R is the mean of data in column N (,RPL36A total') and P (,RPL36A total tech2'), while data in column S (,RPL36A poly(A)+ mean') is the average of columns O (,RPL36A poly(A)+') and Q (,RPL36A poly(A)+ tech2'). The ,Mex67-AA 25°C' experiment contained three biological replicates (see column B) but replicate 1 of the RPL36A poly(A)+ series had no technical replicate thus in column S (,RPL36A poly(A)+ mean') it appears as a data duplication. Please also note that the other issues in the table concerned identical situations with PMA1 mRNA. PMA1 (column V:AA) along with PAB1 (columns T:U) and TDH3 (columns F:G) are data that we did not show in the original submission. Nonetheless we'd like to release them.

Since the idea is to show raw data at Mendeley we think that there is no need to remove the technical replicate data.

HS_global_e_model.csv: This table along with two others is part of the Expanded View Dataset 1. The Dataset EV1 is now revised and contains a legend tab that facilitates navigation through the file. We also added more columns informing about the transcript annotation and biotype (as mentioned above). This complex file contains raw information about the chase control samples: raw and normalized count number and the mean, median and quantile pA-tail values. Regarding those columns some duplications might appear by chance as the values are rooted in datasets of large size (e.g. two mRNAs can have the same median pA-tail length or raw number of counts just by chance).

Most importantly, however, the table lists the mRNA decay rate, mRNA half-life, the mean and median deadenylation rate and terminal adenosine half-life. It also presents the intermediate values that allowed us to compute the deadenylation rate (example below). Most of the 'duplications' in this file were detected between the column called 'HS_e_deadenylation_rate_q0.75q0.95_median_median' and either of the following columns that were used to compute it:

- HS_e_deadenylation_q0.95_rate_median
- HS_e_deadenylation_q0.90_rate_median
- HS_e_deadenylation_q0.85_rate_median
- HS_e_deadenylation_q0.80_rate_median
- HS_e_deadenylation_q0.75_rate_median

The median is given as the middle number of an odd number of values or the average of an even number of observations. **Since there were 5 observations given, our median is mostly a 'duplication' of one of them. We clarified this in the novel Dataset EV1 legend tab and improved the relevant text of the Appendix section 1.10***. As the reader could be interested in using either of the five upper quantile deadenylation rates separately we would like to keep those numbers for information and future use. Each is given with the corresponding mean and standard deviation.

* Appendix section 1.10 revised quote:

'After establishing the uniformity of the chase replicates, we conducted a bulk estimation of decay and deadenylation coefficients from all replicates. In case of the deadenylation rate, for each quantile separately (95th, 90th, 85th, 80th, 75th, 50th, 15th, 10th and 5th) the mean, median and standard deviation of the deadenylation rate was calculated. Next, using the 95th-75th quantiles the mean and medians were calculated from respective intermediate values, and listed along with a standard deviation in Expanded View Dataset 1.'

The Figure below shows the novel legend tab for one of the chase datasets:

B	C	D	E	F	G	H
Mex67AA_25C_chase	column:	column name:	column content description:			
A	transcript	transcript order number	transcript order number			
B	RPG	the full transcript annotation in our reference transcriptome (e.g. for mRNAs the transcript name is a fusion of: SystematicName_StandardName)	does the mRNA belong to RPG (ribosomal protein gene) family? The column allows for sorting by 'YES' or 'NO'			
D	is gene repressed during heat shock		the column allows to sort when the gene is transcriptionally downregulated during response to heat stress according to Vinayachandran et al (2018)			
E	symbol		the standard name (e.g. TDH3)			
F	isoType		type of RNA: 'CDS' = mRNA and various non-coding RNAs (e.g. CUTs, XUTs and SLUTs)			
G	min0_counts_norm_repA		lists normalized count number for each transcripts in replicate A control			
H	min0_counts_norm_repB		lists raw count number for each transcripts in replicate A control			
I	min0_counts_norm_repB		lists normalized count number for each transcripts in replicate B control			
J	min0_counts_norm_repB		lists raw count number for each transcripts in replicate B control			
K	min0_counts_norm_repH		lists normalized count number for each transcripts in replicate H control			
L	min0_counts_norm_repH		lists raw count number for each transcripts in replicate H control			
M	e_decay_mean		lists the mean decay rate for each transcript			
N	e_decay_sd		lists the standard deviation of the mean (columnM) and median (columnN) decay rate estimates for each transcript			
O	e_half_life_mean		lists each mRNA half-life estimate calculated from the mean decay rate (columnM)			
P	e_half_life_mean		lists each mRNA half-life estimate calculated from the median decay rate (columnN)			
Q	e_half_life_mean		lists the mean of the deadenylation rate mean estimates calculated from the 95 th , 90 th , 85 th , 80 th and 75 th quantile (given in columns: X, AA, AD, AG, AJ)			
R	terminal_60A_adenosine_half_life_min		lists the standard deviation of the deadenylation rate median estimates calculated from the 95 th , 90 th , 85 th , 80 th and 75 th quantile (given in columns: X, AA, AD, AG, AJ)			
S	terminal_60A_adenosine_half_life_min		lists the median of the deadenylation rate median estimates calculated from the 95 th , 90 th , 85 th , 80 th and 75 th quantile (given in columns: Y, AB, AE, AH, AK)			
T	e_deadenylation_q75_q95_mean_mean		lists the standard deviation of the deadenylation rate median estimates calculated from the 95 th , 90 th , 85 th , 80 th and 75 th quantile (given in columns: Y, AB, AE, AH, AK)			
U	e_deadenylation_q75_q95_mean_sd		lists the mean deadenylation rate estimates calculated from the 95 th quantile			
V	e_deadenylation_q75_q95_median_mean		lists the standard deviation for the mean and median deadenylation rate estimates calculated from the 95 th quantile			
W	e_deadenylation_q75_q95_median_sd		lists the mean deadenylation rate estimates calculated from the 95 th quantile			
X	e_deadenylation_q95_mean		lists the standard deviation for the mean and median deadenylation rate estimates calculated from the 95 th quantile			
Y	e_deadenylation_q95_mean		lists the mean deadenylation rate estimates calculated from the 95 th quantile			
Z	e_deadenylation_q95_sd		lists the standard deviation for the mean and median deadenylation rate estimates calculated from the 95 th quantile			
AA	e_deadenylation_q90_mean		lists the mean deadenylation rate estimates calculated from the 90 th quantile			
AB	e_deadenylation_q90_mean		lists the standard deviation for the mean and median deadenylation rate estimates calculated from the 90 th quantile			
AC	e_deadenylation_q90_sd		lists the mean deadenylation rate estimates calculated from the 90 th quantile			
AD	e_deadenylation_q85_mean		lists the standard deviation for the mean and median deadenylation rate estimates calculated from the 85 th quantile			
AE	e_deadenylation_q85_mean		lists the mean deadenylation rate estimates calculated from the 85 th quantile			
AF	e_deadenylation_q85_sd		lists the standard deviation for the mean and median deadenylation rate estimates calculated from the 85 th quantile			
AG	e_deadenylation_q80_mean		lists the mean deadenylation rate estimates calculated from the 80 th quantile			
AH	e_deadenylation_q80_mean		lists the standard deviation for the mean and median deadenylation rate estimates calculated from the 80 th quantile			
AI	e_deadenylation_q80_sd		lists the mean deadenylation rate estimates calculated from the 80 th quantile			
AJ	e_deadenylation_q75_mean		lists the standard deviation for the mean and median deadenylation rate estimates calculated from the 75 th quantile			
AK	e_deadenylation_q75_mean		lists the mean deadenylation rate estimates calculated from the 75 th quantile			
AL	e_deadenylation_q75_sd		lists the standard deviation for the mean and median deadenylation rate estimates calculated from the 75 th quantile			
AM	e_deadenylation_q50_mean		lists the mean deadenylation rate estimates calculated from the 50 th quantile			
AN	e_deadenylation_q50_mean		lists the standard deviation for the mean and median deadenylation rate estimates calculated from the 50 th quantile			
AO	e_deadenylation_q50_sd		lists the mean deadenylation rate estimates calculated from the 50 th quantile			
AP	e_deadenylation_q15_mean		lists the standard deviation for the mean and median deadenylation rate estimates calculated from the 15 th quantile			
AQ	e_deadenylation_q15_mean		lists the mean deadenylation rate estimates calculated from the 15 th quantile			
AR	e_deadenylation_q15_sd		lists the standard deviation for the mean and median deadenylation rate estimates calculated from the 15 th quantile			
AS	e_deadenylation_q10_mean		lists the mean deadenylation rate estimates calculated from the 10 th quantile			
AT	e_deadenylation_q10_mean		lists the standard deviation for the mean and median deadenylation rate estimates calculated from the 10 th quantile			
AU	e_deadenylation_q10_sd		lists the mean deadenylation rate estimates calculated from the 10 th quantile			
AV	e_deadenylation_q05_mean		lists the standard deviation for the mean and median deadenylation rate estimates calculated from the 5 th quantile			
AW	e_deadenylation_q05_mean		lists the mean deadenylation rate estimates calculated from the 5 th quantile			
AX	e_deadenylation_q05_sd		lists the standard deviation for the mean and median deadenylation rate estimates calculated from the 5 th quantile			
AY	e_deadenylation_q05_sd		lists the mean of the mean pA-tail lengths in the control samples			
AZ	e_deadenylation_q05_sd		lists the median of the median pA-tail lengths in the control samples			
BA	e_deadenylation_q05_sd		lists the mean value of the mode in the control samples			
BB	e_deadenylation_q05_sd		lists the mean value of the 95 th quantile pA-tail length for control samples			
LEGEND	Mex67AA_25C_chase	Heat stress chase	Thiolutin_25C_chase			

the final estimate is the mean of the mean estimates listed in the table

more estimates are listed, which were not used in the study

Off note, in the original submission the thiolutin deadenylation rate was calculated as the mean of quantiles 90, 85, 80 and 75. For uniformity with the other datasets we now added the 95th quantile to the mean/median. Importantly, the former estimate correlates strongly with the novel one with a 0.988 Spearman rho value (panel A in Figure below shows as scatterplot the new and the old terminal adenosine half-lives grouping along the grey diagonal). The 95th quantile was initially excluded as it slightly increased the estimate standard deviation (1.5-fold on average). However, the standard deviation is still very low (pink series inn panel B in Figure below: revised Appendix Figure S4D last panel). As a result slight corrections were made to Figure 6C (concerning only the green 'thiolutin' series). Importantly, those corrections had no impact on the biological and methodological conclusion, which were mostly drawn from raw data presented in Fig. 6D.-E.

Response to Reviewers

We would like to thank all the Reviewers for the insightful comments, the positive evaluation of our work and for the time devoted to the revision process.

Referee #1:

The authors have addressed all my concerns. Although it is still a complex manuscript to follow, the authors have significantly improved its readability. I have no further concerns and support its publication.

Referee #2:

The authors have extensively revised and improved the submitted manuscript and have responded appropriately to the comments that I made on the first submission. I support publication of the revised manuscript.

Referee #3:

This manuscript is substantially improved over the first submission. I have two minor points. First, the modeling sections, although clearer, are still challenging to read. Second, in the discussion, a larger integration of the results into the field's literature on differential decapping / deadenylation rates, especially findings from mammalian cells (e.g., work from Narry Kim and David Bartel), would be very helpful for the reader. I support publication of the manuscript.

***Answer:** We indeed mostly cited the Eisen et al (2020) work from Bartels' groups as it provided deadenylation modeling in human cells lines and was conceptually closest to our project. We have also cited the fine work by Yi et al, 2018 from the Narry Kim group. However, the choice of works based in the human system is deliberately limited. This is because the deadenylation-decay process is very different in human compared to yeast. Notably, the cytoplasmic adenylases are a huge game-changer that require the generation of a much more complex numerical model. Furthermore, there is a shift in the importance of CCR4-NOT and PAN2/3 deadenylase complexes. We strongly believe that a dedicated comparative work would be more appropriate to put this differences forward.*

Data availability:

We removed the Reviewer tokens from the main publication file. To access the raw data deposited at Mendeley please log into your Mendeley account and use the link:

<https://data.mendeley.com/preview/2j3hh37zszs?a=b2968161-df0a-41ab-8af6-b5ddfe546e54>

The GEO accession Reviewer token is available at Gene Expression Omnibus GSE272785

(<https://www.ncbi.nlm.nih.gov/geo/query/acc.cgi?acc=GSE272785>, reviewer access token: mnidkcwolfoqvnngf)

Dear Dr. Tudek,

I am pleased to inform you that your manuscript has been accepted for publication in the EMBO Journal.

Yours sincerely,

Cornelius Schneider, PhD
Editor
The EMBO Journal
c.schneider@embojournal.org
